# Understanding meteorological and physio-geographical controls of variability of flood event classes in headstream catchments of China

Yongyong Zhang[1], Yongqiang Zhang[1], Xiaoyan Zhai[2], Jun Xia[3,1], Qiuhong Tang[1], Wei Wang[1], Jian Wu[1], Xiaoyu Niu[1], Bing Han[1]

[1]Key Laboratory of Water Cycle and Related Land Surface Processes, Institute of Geographic Sciences and Natural Resources Research, Chinese Academy of Sciences, Beijing, 100101, China.
[2]China Institute of Water Resources and Hydropower Research, Beijing, 100038, China.
[3]State Key Laboratory of Water Resources and Hydropower Engineering Science, Wuhan University, Wuhan, 430072, China.

*Correspondence to*: Yongyong Zhang (zhangyy003@igsnrr.ac.cn)

**Abstract.** Classification is beneficial for understanding flood variabilities and their formation mechanisms from massive flood event samples for both flood scientific research and management purposes. Our study investigates comprehensive manageable flood event classes from 1446 unregulated flood events in 68 headstream catchments of China using the hierarchical and partitional clustering methods. Control mechanisms of meteorological and physio-geographical factors (e.g., meteorology, land cover and catchment attributes) on spatial and temporal variabilities of individual flood event classes are explored using constrained rank analysis and Monte Carlo permutation test. We identify five robust flood event classes, i.e., moderately, highly, and slightly fast floods, as well as moderately and highly slow floods, which accounts for 24.0%, 21.2%, 25.9%, 13.5% and 15.4% of total events, respectively. All the classes are evenly distributed in the whole period, but the spatial distributions are quite distinct. The fast flood classes are mainly in the southern China, and the slow flood classes are mainly in the northern China and the transition region between southern and northern China. The meteorological category plays a dominant role in flood event variabilities, followed by catchment attributes and land covers. Precipitation factors, such as volume and intensity, and drought index during the events are the significant control factors. Our study provides insights into flood event variabilities and aids in flood prediction and control.

## 1 Introduction

Flood events usually show tremendous spatial and temporal variabilities in behavior due to heterogeneities in meteorological and underlying surface conditions over large basins or entire regions (e.g., county, continent and world) (Berger and Entekhabi, 2001). Existing studies provide insights on impacts of changes in meteorological or underlying surface conditions on specific flood metrics (e.g., magnitude, peak, timing or seasonality) and their changes using trend separation method, correlation testing, mathematical modelling, and so on (Berghuijs *et al*., 2016; Tarasova *et al*., 2018; Liu *et al*., 2020; Wang *et al*., 2024). However, all of these studies are implemented at event scale or in catchments with certain landscapes and climates, which are insufficient for the comprehensive flood change investigation and generalized results (Tarasova *et al*., 2019; Zhang *et al*., 2020). Flood

event similarity analysis is beneficial to investigate comprehensive dynamic characteristics of flood events in space and time by grouping massive heterogenous events into some manageable classes with significantly statistical differences of flood responses (e.g., great or small floods, fast or slow floods, rain or snowmelt floods) (Brunner, 2018). Flood event class

determines hydrological response characteristics, longitudinal and lateral transfers of energy and material, and structures and functions of riverine ecosystems (Arthington *et al*., 2006; Poff and Zimmerman, 2010). The class also directly determines flood disaster losses for human society and affects the strategy formulations of flood control and management (Hirabayashi *et al*., 2013; Jongman *et al*., 2015). Hence, for both flood scientific research and management purposes, it is fundamentally important to identify the flood event classes and their formation mechanisms (Sikorska *et al*., 2015).


Inductive and deductive approaches are reported for the flood event similarity classification according to the clustering objectives (Olden *et al*., 2012). The inductive approach directly focuses on the shape similarity of flood events by clustering the response characteristics extracted from the flood event hydrographs. The response characteristics include magnitude, frequency, duration, timing and seasonality, variability metrics, which are considered as the critical components to characterize

the entire range of flood events (Poff *et al*., 1997; Kuentz *et al*., 2017;Zhang et al., 2020). The reported flood event classes are the fast events with steep rising and falling limbs, the slow events with both elongated rising and falling limbs, the sharp or fast flood event, the flash flood (Kuentz *et al*., 2017; Brunner *et al*., 2018; Zhai *et al*., 2021; Zhang *et al*., 2020). The deductive approach mainly focuses on the similarity of environmental factors which control flood events, such as meteorological variables (e.g., storm intensity, duration and snowmelt) and physio-geographical conditions (e.g., soil moisture, land cover and

topography) (Merz and Blöschl, 2003; Ali *et al*., 2012; Brunner *et al*., 2018; Zhang *et al*., 2022). The reported flood event classes are the long-rain floods, short-rain floods, flash floods, rain-on-snow floods, and snowmelt floods (Merz and Blöschl, 2003; Sikorska *et al*., 2015; Brunner *et al*., 2018; Zhang *et al*., 2022). However, the control relationships of environmental factors on flood event shapes are not well defined so that the identified classes are not exactly helpful to investigate the flood change patterns at event scale. Therefore, it is a challenge to better understand the formation mechanisms of individual flood

event classes.

The main procedure of existing flood event classification is to cluster the similarity of flood event attributes (e.g., flood response characteristics or control factors) across the spatial and temporal scales. According to the classification procedure, there are two widely-adopted approaches, namely the tree clustering methods (e.g., decision tree, regression tree, fuzzy tree and random forest) (Sikorska *et al*., 2015; Brunner *et al*., 2017) and the non-tree clustering methods (e.g., single linkage,

complete linkage, average linkage, centroid linkage, ward linkage, *k*-mean, *k*-medoids) (Zhang *et al*., 2020; Zhai *et al*., 2021). The tree clustering methods are implemented to binarily split all the flood events successively into smaller classes of similar flood events according to the thresholds of flood response metrics until obtaining final classes (Sikorska *et al*., 2015; Brunner *et al*., 2017). The classification results could be applicable to other basins and the flood response characteristics of different

studies would be directly comparable if the same thresholds are adopted. However, these methods assume that the boundaries

of flood response metrics in different classes are clear and the thresholds of flood response metrics should be predefined and should not overlap among different classes (Olden *et al*., 2012; Sikorska *et al*., 2015; Zhai *et al*., 2021). Additionally, the classification is very sensitive to the thresholds, whose small changes would cause different flood event classes (Olden *et al*., 2012; Sikorska *et al*., 2015). Therefore, it will be difficult to define the thresholds clearly to get robust classification performance. The non-tree clustering methods are implemented to directly split all the flood events according to different division rules of the comprehensive similarity measures of flood event shapes or metrics (Olden *et al*., 2012; Zhang *et al*., 2020). The class boundaries of flood response metrics are vague, and the flood event classes are mainly based on the class membership degree deduced from sufficient of heterogeneous flood events (Sikorska *et al*., 2015). The flood response characteristics of individual classes were usually qualitatively described to distinguish the differences among classes (Olden *et al*., 2012; Tarasova *et al*., 2019; Zhang *et al*., 2020). Therefore, the classification results obtained from different flood event samples are still difficult to quantitatively compare even though the flood response characteristics or hydrographs in the certain class are similar (e.g., high or low, fast or slow floods) (Zhang *et al*., 2024). The determinations of clustering method and final cluster number are subjective in most existing studies, and the assessment of clustering performance is usually unavailable (Olden *et al*., 2012; Sikorska *et al*., 2015; Brunner *et al*., 2017). Therefore, robustness of flood event classification should be further explored.

The main aim of this study is to investigate the flood event similarity and the control mechanisms of meteorological and physio-geographical factors in space and time at class scale across China. Over one thousand unregulated flood events at 68 heterogeneous catchments with wider meteorological and physio-geographical conditions are selected for our study. The specific objectives are as follows:

(i) to determine the optimal flood event classes by comparing multiple classification performance criteria of both the hierarchical and partitional clustering methods;

(ii) to identify the main flood response characteristics of individual classes and their spatial and temporal variabilities;

(iii) to quantify the effects of meteorological and physio-geographical factors on the variabilities of individual flood event classes.

This study provides more comprehensive insights into meteorological and physio-geographical controls of variabilities of flood event classes, and provides the mechanism supports for predicting flood event classes.

## 2 Study area and data sources

According to the Köppen-Geiger climate classification (Peel *et al*., 2007), China has diverse climate types, including alpine tundra climate (ET for Köppen-Geiger codes), tropical climate (A), arid, steppe and cold climate (BSk), arid, desert and cold (BWk), cold without dry season (Df), cold with dry winter (Dw), temperate without dry season (Cf) and temperate with dry

winter (Cw). Most Köppen-Geiger climate types in China (i.e., A, Dw, Cf and Cw) are controlled by the southeast and southwest monsoons in the summer with temperate and humid climates and the northwestern and northeastern monsoons in the winter with cold and dry climates. In these monsoon controlled climate types, the mean annual precipitation was 365–2654 mm with a mean of 1184 mm, of which over 65% fell between May and September according to the gauged daily precipitation observations from 2001 to 2020 in these regions. This led to frequent flooding and thus the region in the monsoon controlled climate types is usually considered as the flood-prone area of China (China Institute of Water Resources and Hydropower Research and Research Center on Flood and Drought Disaster Prevention and Reduction, the Ministry of Water Resources, 2021). In the last decade, flooding occurred in 455 rivers annually, which affected 822 million people and averaged over 10 billion US dollars (Ministry of Water Resources of the People's Republic of China, 2020a).

Sixty-eight headstream stations spread across the flood-prone areas of upper major river basins in China were selected with catchment areas ranging from 21 $km^2$ to 4830 $km^2$, which were in all the monsoon controlled climate types of China, except tropical climate in the islands (i.e., A) (Figure 1). Most catchments had large forest coverage, with mean area percentages of 67.0%, particularly in the Yangtze (69.9%) and Pearl (68.7%) River Basins. A total of 1446 unregulated flood events with hourly time steps were collected from the Hydrological Yearbooks of the Songliao, Yellow, Huaihe, Yangtze, Southeast and Pearl River Basins over the period 1993–2015. The event was extracted following the Standard of Ministry of Water Resources of the People's Republic of China, i.e., Code for hydrologic data processing (SL/T 247–2020) (Ministry of Water Resources of the People's Republic of China, 2020b). The extracted flood events at the individual stations usually had the maximum flood peak or flood volume, isolated flood peak, continuous flood peaks, or flood peak after prolonged drought during the high and normal flow years (Ministry of Water Resources of the People's Republic of China, 2020b). There were 53 events at four stations, 104 events at four stations, 215 events at 13 stations, 844 events at 38 stations, 90 events at five stations, and 140 events at four stations in the upper tributaries of the Songliao River Basin (i.e., Songhua and Wusuli Rivers), Yellow River Basin (i.e., Huangshui, Jinghe and Yiluo Rivers), Huaihe River Basin (i.e., Northern and Southern tributaries), Yangtze River Basin (i.e., Hanjiang, Wujiang, Dongtinglake, Poyanglake, and lower Yangtze River), Southeast River Basin (i.e., Qiantang and Jinjiang Rivers) and Pearl River Basin (i.e., Beijing, Xijiang and Dongjiang Rivers), respectively. No less than 10 flood events were collected for every station to ensure the representativeness. The densities of flood events and gauges in the Southern China (i.e., Huaihe, Yangtze, Southeast and Pearl River Basins) were 1.25–11.01 times and 2.94–9.15 times greater than those in the Northern China (i.e., Songliao and Yellow River Basins) because of the higher occurrences of flood events (Table S1 in the Supplement) (China Institute of Water Resources and Hydropower Research and Research Center on Flood and Drought Disaster Prevention and Reduction, the Ministry of Water Resources, 2021).

Meteorological, catchment and land cover data sources were collected together to calculate the potential meteorological and physio-geographical control factors and quantify their contributions on the spatial and temporal variabilities of flood event classes. The meteorological data sources were the synchronous hourly precipitation events extracted from the Hydrological

Yearbooks, and the daily precipitation, maximum and minimum temperature observations from 1993 to 2015 at the meteorological stations within or around the catchments downloaded from the China Meteorological Data Sharing Service System. All the meteorological stations in the buffer zone with a radius of 100 km of every catchment centers were selected.

The station number was 466 in total and no less than eight stations for each catchment. The daily meteorological variables were interpolated to the catchment by the inverse distance weighting method, which is one of commonly-used meteorological interpolation methods (Ahrens, 2006; Tan *et al*., 2021). The geographic information system (GIS) data contained the digital elevation model, and the land cover data series in six periods (i.e., 1990, 1995, 2000, 2005, 2010 and 2015) whose spatial resolution is 30 m×30 m. The GIS data were downloaded from the Data Center of Resources and Environmental Science,

Chinese Academy of Sciences, and were adopted to extract catchment attributes and area percentages of individual land cover types. All these data sources for control factor calculations had been widely used to represent the meteorological and underlying surface conditions in China for hydrometeorogical change detection and causal analysis, hydrological modelling, and so on (Zhang *et al*., 2020; Du *et al*., 2022; Zhang *et al*., 2024).

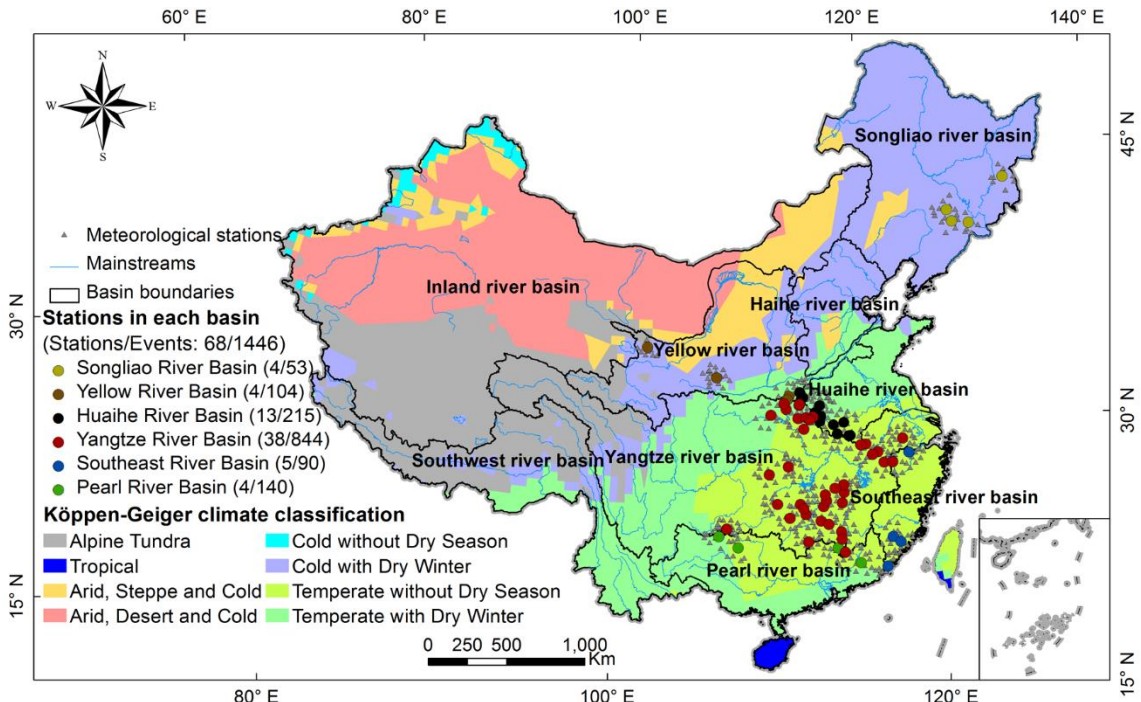

**Figure 1. Spatial distributions of all the selected flood events and their corresponding climate types**

## 3 Methods

### 3.1 Flood response metrics

The flood classification in our study mainly focuses on the detailed response characteristics of flood hydrographs by the inductive approach. The magnitude, variability, timing, duration, and rate of changes are widely-accepted as the main five components to characterize the entire flood events (Arthington *et al.*, 2006; Kennard *et al.*, 2010; Poff *et al.*, 2007; Zhang *et al.*, 2012) and thus are also adopted to characterize the detailed flood responses in our study. Additionally, flood peak number is one of the most important metrics for flood control (Aristeidis *et al.*, 2010; Rustomji *et al.*, 2009). Therefore, nine metrics are used to fully characterize the response of flood events (Table 1). Particularly, $T_{bgn}$ is characterized using the circular statistical approach which translates the calendar date into the polar coordinates on the circumference of a circle, and is beneficial to distinguish the seasonal pattern (Fisher, 1993; Dhakal *et al.*, 2015).

**Table 1. Metrics used to characterize flood responses in our study**

| Components | Metrics | Abbreviations | Units | Equations | References |
|---|---|---|---|---|---|
| Magnitude | Total flood volume | $R$ | mm·day⁻¹ | $R = 86.4 \cdot \sum_{t=TF_{bgn}}^{TF_{end}} Q_t / A$ | Fisher, 1993; Black and Werritty, 1997; Poff *et al.*, 2007; Villarini, 2016; Hall and Blöschl, 2018; Zhang *et al.*, 2020 |
| | Maximum flood peak | $Q_{pk}$ | mm·day⁻¹ | $Q_{pk} = \max(86.4 \cdot Q_t) / A$ | |
| Variability | Coefficient of variation | $CV$ | - | $CV = \sigma / Q_{av}$ | |
| Timing | Ratio of beginning date of flood event in the calendar year using circular statistics | $T_{bgn}$ | radian | $T_{bgn} = 2\pi \cdot TF_{bgn} / TD$ | |
| | Ratio of occurrence time of maximum flood peak to flood duration | $T_{pk}$ | % | $T_{pk} = TF_{pk} / T_{drn} \cdot 100$ | |
| Duration | Duration of flood event | $T_{drn}$ | h | $T_{drn} = 24 \cdot (TF_{end} - TF_{bgn} + 1)$ | |
| Rate of changes | Mean rate of positive changes | $RQ_r$ | h⁻¹ | $RQ_r = \dfrac{(Q_{pk} - Q_{bgn}) / Q_{av}}{(TF_{pk} - TF_{bgn} + 1) \cdot 24}$ | |
| | Mean rate of negative changes | $RQ_d$ | h⁻¹ | $RQ_d = \dfrac{(Q_{pk} - Q_{end}) / Q_{av}}{(TF_{end} - TF_{pk} + 1) \cdot 24}$ | |
| Number | Number of peaks during the event | $N_{pk}$ | - | | Aristeidis *et al.*, 2010; Zhai *et al.*, 2021 |

Note: $Q_t$ is the flood magnitude on day $t$ (m³·s⁻¹); $Q_{av}$ is the mean flood magnitude (m³·s⁻¹); $Q_{bgn}$ and $Q_{end}$ are flood magnitudes at the beginning and end of event (m³·s⁻¹), respectively; $\sigma$ is the standard deviation of flood magnitude (m³·s⁻¹); $TD$ is the total days of the calendar year (day), i.e., 365 for common year or 366 for leap year; $TF_{bgn}$ and $TF_{end}$ are the beginning and end dates

of flood events; $TF_{pk}$ is the occurrence date of maximum flood peak; $A$ is the catchment area (km$^2$) ; 86.4 is the unit conversion factor from m$^3$·s$^{-1}$·km$^{-2}$ to mm.

## 3.2 Flood event classification

High dimensionality and multicollinearity exist among flood response metrics and affect the flood event classification when a large number of metrics are considered (Olden *et al*., 2012; Zhang *et al*., 2012). Here, principal component analysis is used to transform the high dimensional metrics into a few principal components (*PCA*) based on the orthogonal transform. If the cumulative variance is over 85% of the total explained variances of all the flood response metrics, the first *m PCA*s are selected for classification. The main flood response metrics in the individual *PCA*s were determined according to the load coefficient

matrix. If the load coefficient is over 0.45, the corresponding flood response metric are considered to be highly correlated with the *PCA*.

Subsequently, both the hierarchical (Ward's) and partitional (*k*-medoids) clustering methods are used to cluster flood events based on the similarity of the selected *PCA*s. Euclidean distance is the distance measure. Twenty-two criteria are used to assess

the classification performance and determine the best number of clusters, i.e., KL, CH, Hartigan, CCC, Scott, Marriot, TrCovW, TraceW, Friedman, Silhouette, Ratkowsky, Ball, Ptbiserial, Dunn, Rubin, Cindex, DB, Duda, Pseudot2, McClain, SDindex and SDbw (Table S2 in the Supplement) (Charrad et al., 2014). The greater values of the first fourteen indexes (i.e., KL to Dunn) or the smaller values of the rest eight criteria (i.e., Rubin to SDbw) indicate the better classification. If the best criteria number is the largest in a certain cluster number, the cluster number is optimal and the corresponding clustering method is also

selected. The implementations of all the multivariable statistical analyses are given in the Appendix A.

## 3.3 Control mechanisms of meteorological and physio-geographical factors on the variabilities of flood event classes

### 3.3.1 Meteorological and physio-geographical factors

The meteorological (e.g., precipitation intensity, timing and duration, evapotranspiration volume) and physio-geographical

factors (e.g., land covers and catchment attributes) directly affect the flood generation and routing processes, which thus cause the diversity of flood event shapes (Ali *et al*., 2012; Brunner *et al*., 2018; Merz and Blöschl, 2003; Zhang *et al*., 2022). The potential control factors are selected as many as possible to investigate the control mechanisms on the variability of flood event classes according to the existing studies. There are 34 meteorological, catchment and land cover factors selected in all the catchments (Table 2). In the meteorological factor category, 17 factors related to precipitation, potential evapotranspiration

and drought index are selected, including the amounts, intensities and timing factors during flood events, in the antecedent period and at annual scale. All the precipitation factors during the flood events are extracted using the hourly precipitation

observations. The precipitation factors at daily or annual scale are extracted using the daily precipitation observations. The potential evapotranspiration at daily or annual scale is estimated using the Hargreaves method (Hargreaves and Samani, 1982), and the drought index is the ratio of potential evapotranspiration to precipitation. All these factors mainly affect the flood yield

processes (Merz and Blöschl, 2003; Aristeidis *et al*., 2010; Zhang *et al*., 2022).

In the physio-geographical factor category, 10 catchment attributes are selected, including catchment location, area, elevation and slope, river density and slope, and seven land cover factors for the six land cover periods are selected, including the area fractions of paddy, dryland, forest, grassland, water, urban and rural area to the total catchment, respectively. All these physio-

geographical factors are extracted using the `Hydrology` and `Zonal` functions of the Spatial Analyst Tools in the ArcGIS Desktop (version 10.0), and mainly affect the flood yield and routing processes (Ali *et al*., 2012; Kuentz *et al*., 2017; Zhai *et al*., 2021).

**Table 2. Meteorological and physio-geographical factors in our study**

| Factor categories | Factors | Data sources | Flood event effects |
|---|---|---|---|
| Meteorology | Precipitation | • pcp_ant: cumulative amount in the antecedent seven days (mm);<br>• pcp_dur:total amount during the flood event (mm);<br>• pcp_av: mean amount during the flood event (mm hr$^{-1}$);<br>• pcp_max: maximum intensity during the flood event (mm hr$^{-1}$);<br>• pcp_max: maximum intensity during the flood event (mm hr$^{-1}$);<br>• pcp_Tbeg: precipitation timing;<br>• pcp_Tdur: precipitation duration (days);<br>• pcp_ann: annual mean amount (mm);<br>• pcp_year: amount in the year when the flood event happens (mm) | Hourly precipitation in hydrological yearbooks; daily precipitation at 466 meteorological stations | Flood yield process |
| | Potential evapotranspiration | • pet_ant: cumulative amount in the antecedent seven days (mm);<br>• pet_dur: total amount during the flood event (mm)<br>• pet_max: maximum intensity during the flood event (mm hr$^{-1}$)<br>• pet_ann: annual mean amount (mm);<br>• pet_year: amount in the year when the flood event happens (mm) | Daily maximum and minimum temperature at 466 meteorological stations | Flood yield process |
| | Drought index | • SPEI_ant: mean value in the antecedent seven days;<br>• SPEI_dur: mean value during the flood event ;<br>• SPEI_ann: annual mean value;<br>• SPEI_year: mean value in the year when the flood event happens | Daily maximum and minimum temperature at 466 meteorological stations | Flood yield process |

| | | | | |
|---|---|---|---|---|
| | | • Longitude: longitude of catchment center<br>• Latitude: latitude of catchment center | Global positioning system | Meteorological conditions |
| Physio-geography | Catchment attributes | • Slope: catchment slope (%);<br>• Area: catchment area ($km^2$);<br>• Length: catchment slope length (km);<br>• Elevation: average elevation of catchment (m);<br>• MaxiElev: maximum elevation of catchment (m); | Digital elevation model (size: 30 m×30 m) | Flood yield and overland routing processes |
| | | • Rivden: river density ($km/km^2$);<br>• RivSlope: river slope (%);<br>• Rwd: ratio of river width to depth (m/m); | Digital elevation model (size: 30 m×30 m) | Flood routing processes in river system |
| | Land covers | • Rpaddy: area fraction of paddy to catchment (%);<br>• Rdryland: area fraction of dryland to catchment (%);<br>• Rforest: area fraction of forest to catchment (%);<br>• Rgrass: area fraction of grass to catchment (%);<br>• Rwater: area fraction of water to catchment (%);<br>• Rurban: area fraction of urban to catchment (%);<br>• Rrural: area fraction of unused land to catchment (%) | Land covers in 1990, 1995, 2000, 2005, 2010 and 2015 (size: 30 m×30 m) | Flood yield and overland routing processes |


### 3.3.2 Effect quantifications of meteorological and physio-geographical factors

The constrained rank analysis is adopted to quantify the direct and interactive effects of multiple control factor categories on spatial and temporal variabilities of individual flood event classes for both the distributed and lumped analyses. The widely adopted methods of constrained rank analysis are the Redundancy Analysis (RDA) and the Canonical Correlation Analysis

(CCA). The RAD is a linear model and the CCA is a unimodal model, both of which are the extended methods of principal component analysis interactive with regression analysis. These methods have strong advantages to solve multiple linear regressions and interactions between dependent and independent variable matrixes which are transformed into a few independent composite factors (ter Braak, 1986; Legendre and Anderson, 1999), and are beneficial to quantify the effects of independent variable matrix on dependent variable matrix and to find the most important factors. Both methods have been

commonly used in testing the multispecies response to environmental variables in the biological or ecological sciences (Legendre and Anderson, 1999), effects of physio-geographical factors and human activities on diffuse nutrient losses or water quality (Zhang *et al.*, 2016; Shi *et al.*, 2017), and so on.

The selection of CCA and RDA is based on the first axis length of detrended correspondence analysis. The CCA is proposed

when the first axis length is greater than 4.0, while the RDA is proposed when the first axis length is less than 3.0. Otherwise, both CCA and RDA are proposed (ter Braak, 1986; Zhang *et al.*, 2020). Additionally, because of multiple control factor categories considered, two constrained rank analyses are implemented, namely entire and partial analyses. The entire analysis is implemented by involving all the control factors as the independent variable matrix, and the variance percentage explained by independent variable matrix to the total variance of dependent variable matrix is considered as the entire contribution of all

the control factors or categories on total variabilities of flood event classes. The partial analyses of individual control factor categories are also implemented by involving a certain control factor category as the independent matrix and the effects of other control factor categories are held constant. The percentage of constrained variance is considered as the individual contribution of involved control factor category. The meteorological, land cover and catchment categories are adopted for the analysis individually, and their individual contributions are determined. If the sum of all the individual contributions is less than the entire contribution of all the factors, the interactive effects exist among the control factors and the difference between the summed and entire contributions is the interactive contribution (Legendre and Anderson, 1999; Zhang *et al.*, 2016).

Furthermore, the Monte Carlo permutation test is adopted to test the statistical significance of control factors, and obtain the correlation coefficients (*r*) between flood response matrix and control factor matrix in the individual catchments (i.e., distributed analysis) and the entire region (i.e., lumped analysis), respectively. All the meteorological and physio-geographical factors are included for the lumped analysis, while the catchment attributes are excluded for the distributed analysis because they are not dynamic in the individual catchments. The significant statistical interval is set as 95%, i.e., $p=0.05$.

## 4 Results

### 4.1 Flood event classification

By the tests of independence and linear correlation for all the flood response metrics, $T_{bgn}$ is independent from $R$, $RQ_r$, $RQ_d$ and $N_{pk}$; $Q_{pk}$ is independent from $T_{pk}$; and $N_{pk}$ is independent from $RQ_r$ and $RQ_d$. Expect these independent metrics, all the other metrics have linear correlations with each other (Table S3 in the Supplement). By the principal component analysis, five independent *PCA*s are found with the total cumulative variance of 85.7%, all of which are selected in our study (Table 3). The first *PCA* is related with magnitude, variability and rates of changes with the explained variances of 33.3%. The second *PCA* is related with magnitude, variability and peak number with the explained variances of 17.0%. The third–fifth *PCA*s are mainly related with flood event duration, beginning time of flood event and flood peak timing with the explained variances of 16.0%, 10.8% and 8.6%, respectively. Furthermore, the optimal classification of all the 1446 flood events are determined by comparing the classification performance between the hierarchical and *k*-medoids clustering methods. The five clusters using the *k*-medoids clustering method are optimum for further analysis in our study (Figure B1 in the Appendix B).

**Table 3. Loads coefficients of flood response metrics in the selected PCAs and their explained variances**

| Components | Variances (%) | Main hydrological metrics and their coefficients | Hydrological meanings |
|---|---|---|---|
| *PCA*1 | 33.3 | $Q_{pk}$ (0.97), $R$ (0.61), $RQ_r$ (0.84) and $RQ_d$ (0.84) | Flood magnitude and rates of changes |
| *PCA*2 | 17.0 | $R$ (0.51), $CV$ (-0.47), $T_{pk}$ (0.56) and $N_{pk}$ (0.77) | Flood magnitude, variability and peak number |
| *PCA*3 | 16.0 | $T_{drn}$ (0.84) | Flood event duration |

| | | | |
|---|---|---|---|
| PCA4 | 10.8 | $T_{bgn}$ (0.92) | Beginning time of flood event |
| PCA5 | 8.6 | $T_{pk}$ (0.64) | Flood peak timing |

## 4.2 Flood response characteristics of different classes

The value ranges of flood response metrics in different classes are presented in Figure 2 and Table S4 in the Supplement. For the magnitude metrics, the distributions of both total flood volume ($R$) and maximum flood peak ($Q_{pk}$) are the same among different classes. That is to say, the metric values are the largest in Class 3, followed by Classes 5, 2, 1 and 4. For the variability metrics ($CV$), the events are the most variable in Class 5, and are slightly variable in the other Classes with the mean $CV$ being less than 1.0. For the timing and duration metrics (i.e., $T_{bgn}$, $T_{drn}$ and $T_{pk}$), 73.2% of flood events in Class 1 occur before the wet season (i.e., January–May), and 58.5%, 67.7% and 57.0% of flood events in Classes 2, 3 and 5 occur in the earlier wet season (i.e., June–July), and 52.8% of flood events in Class 4 occur in the latter wet season (i.e., August–September). The mean duration ($T_{drn}$) is the longest in Class 5, followed by Classes 3 and 1. The mean $T_{drn}$ values in Classes 4 and 2 are the shortest. The timings of maximum flood peaks ($T_{pk}$) are usually the largest in Class 2 with the mean of 50.6%±10.3%, which means that the flood peaks mainly occur in the middle or late stages of flood events. The flood peaks usually occur in the early stage of flood events in the other classes (i.e., Classes 1, 2, 4 and 5). Particularly in Class 3, the mean $T_{pk}$ value is only 23.7%±13.6%.

For the rates of changes, $RQ_r$ in most classes are much greater than $RQ_d$ because the flood peaks usually occur in the early stage of flood events, except Class 2. The largest values of both $RQ_r$ and $RQ_d$ are in Class 3 because of the greatest flood peak. The smallest $RQ_r$ values are mainly in Classes 2 because of the late occurrences of flood peaks, while the smallest $RQ_d$ values are mainly in Class 5 because of the long durations of flood recession. For the flood peak number ($N_{pk}$), 71.2%, 69.9%, 76.5% and 77.1% of flood events has one flood peaks in Classes 1, 2, 4 and 5, respectively, and multiple flood peaks (i.e., two–four) exist in 94.4% of total flood events in Class 3, accounting for 33.8% (two peaks), 48.7% (three peaks) and 11.8% (four peaks), respectively.

According to the metric distributions (Figure 2), and hydrographs and duration frequencies (Figure 3) of individual flood event classes, we can conclude that Class 1 is for moderately fast flood events occurring before the wet season, characterized by a single peak and moderate duration, referred to as the "moderately fast flood event class". Class 2 represents highly fast flood events with a single peak in the late stage and short duration, denoted as the "highly fast flood event class". Class 3 exhibits highly slow flood events during the latter part of the wet season, featuring multiple peaks and long duration, known as the "highly slow and multipeak flood event class". Class 4 reflects slightly fast flood events occurring in the latter wet season with a single peak and short duration, named the "slightly fast flood event class". Lastly, Class 5 displays moderately slow flood events with a single peak and long durations, designated as the "moderately slow flood event class".

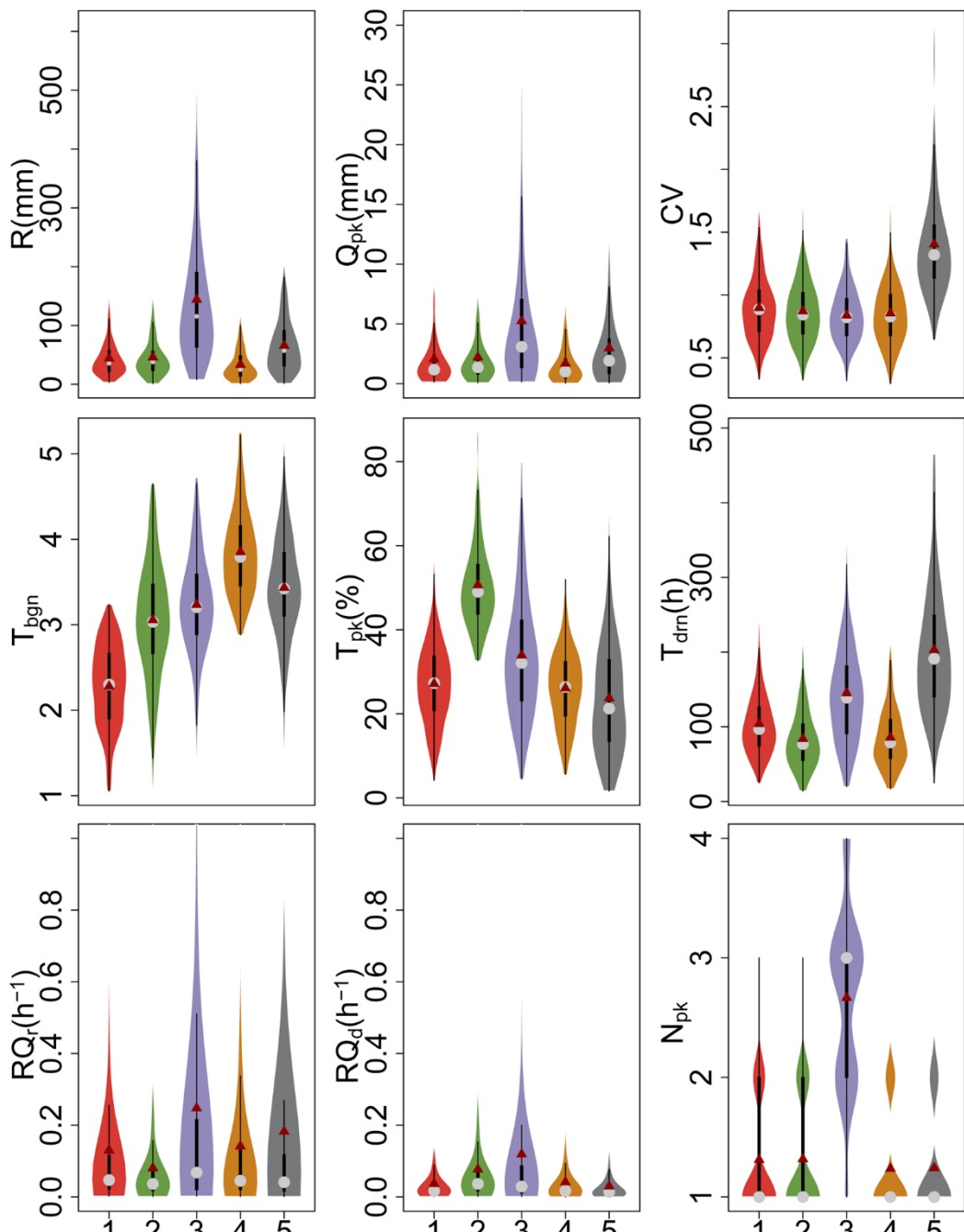

**Figure 2. Variations of flood response metrics among Classes 1–5. The solid darkred dot and gray dot define the mean and 50th percentile values, respectively. Each black box means the 25th and 75th percentile values, and the vertical line defines the minimum and maximum values without outliers. The violin shape means the frequency distribution of flood response metric.**

285

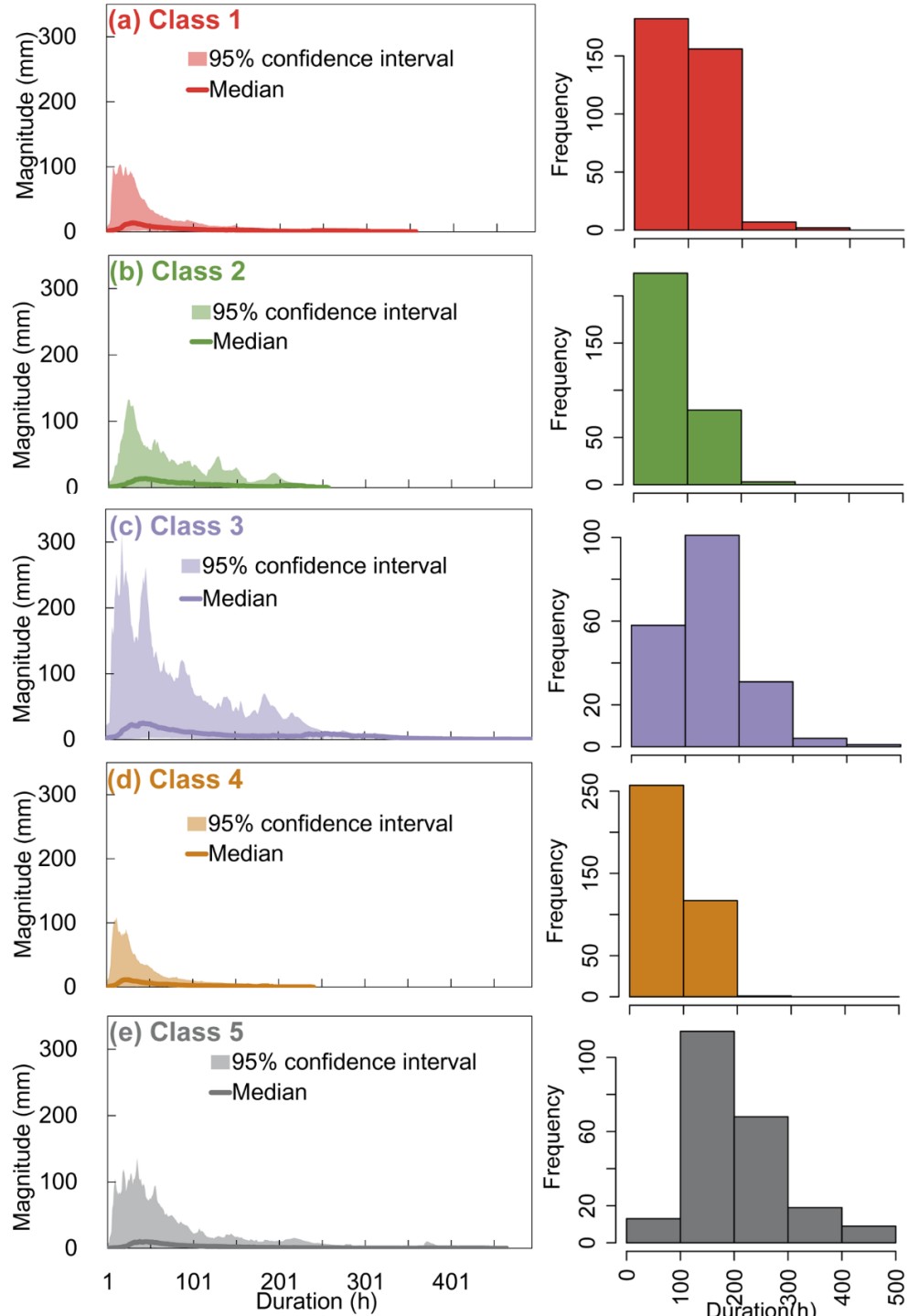

Figure 3. Flood event distributions in the 95% confidence interval and their median, and their duration frequencies of Classes 1–5 (a–e)

**4.3 Spatial and temporal distributions of flood event classes**

The spatial distributions of individual classes are showed in Figures 4 and S1, and Table S5 in the Supplement. The moderately fast flood event class (i.e., Class 1) is mainly in the upper Dongjiang River of the Pearl River Basin, Poyanglake and Dongtinglake tributaries of Yangtze River Basin, accounting for 37.1% (52/140) and 29.7% (251/844) of total events in the main river basins, respectively. Specifically, Class 1 is dominant at the Yanling (54.5%, 18/33) and Tongtang (50.0%, 14/28) stations in the Dongtinglake tributaries, the Shanggao (52.6%, 10/19) station in the Poyanglake tributaries, and the Hezikou (47.2%, 42/89) station in the Dongjiang River. The highly fast flood event class (i.e., Class 2) is mainly in the upper Beijing River of the Pearl River Basin, and Dongtinglake tributaries of Yangtze River Basin, accounting for 31.4% (44/140) and 22.5% (190/844) of total events in the main river basins, respectively. Class 2 is particularly dominant at the Xiaogulu (80.0%, 24/30) station in the Beijiang River, and the Tangdukou (57.6%, 19/33) station in the Dongtinglake tributaries. The highly slow and multipeak flood event class (i.e., Class 3) is mainly in the upper Jinjiang, Qiantang and Minjiang Rivers in the Southeast River Basin, accounting for 42.2% (38/90) of total events, particularly at the Longshan (69.6%, 16/23) station in the Jinjiang River. The slightly fast flood event class (i.e., Class 4) is mainly in the upper Huangshui, Jinghe and Yiluo Rivers of the Yellow River Basin, and upper Songhua and Wusuli Rivers of the Songliao River Basins, accounting for 64.4% (67/104) and 60.4% (32/53) of total events in the main river basins, respectively. This class is dominant at the Qiaotou (77.3%, 17/22) station in the Huangshui River, the Huating (63.6%, 7/11) station in the Jinghe River and the Luanchuan (69.2%, 27/39) station in the Yiluo River, the Jingyu (69.2%, 9/13) and Dongfeng (64.3%, 9/14) stations in the Songhua River, and the Muling (58.3%, 7/12) station in the Wusuli River. The moderately slow flood event class (i.e., Class 5) is mainly in the southern tributaries of Huaihe River Basin, accounting for 47.4% (102/215) of total events, particularly at the Beimiaoji (100%, 12/12) and Qilin (70.0%, 7/10) stations. Therefore, the Classes 1 to 3 are mainly in the Temperate without Dry Season climate region in southern China (Figure 1), the Class 4 is mainly in the Cold with Dry Winter climate region in northern China, and the Class 5 is mainly in the transition region between Temperate without Dry Season climate and Cold with Dry Winter climate.

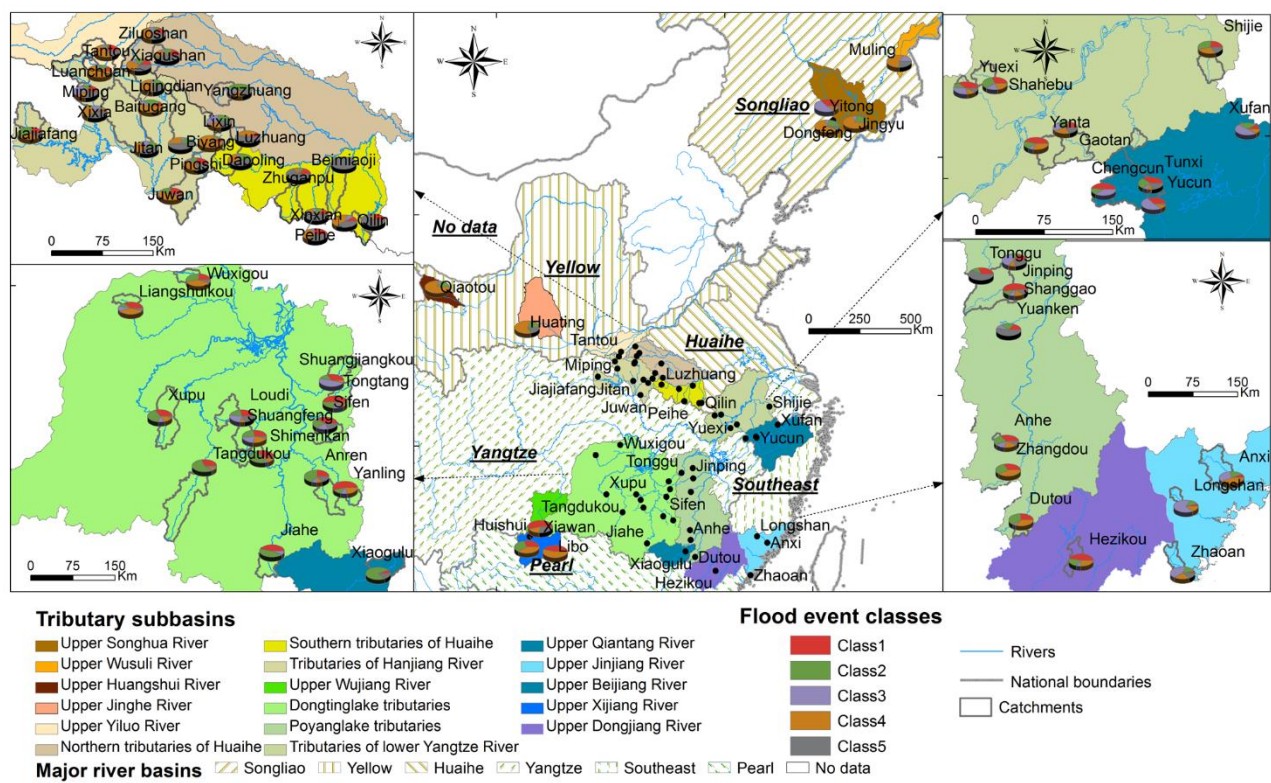

**Figure 4. Spatial variabilities of individual flood event classes at headstream stations of major river basins**

According to the interannual distributions of individual classes (Figure 5a), all the classes are evenly distributed, whose annual
mean percentages are 24.0±5.9%, 21.2±6.4%, 13.5±7.7%, 25.9±6.2%, and 15.4±12.5%, respectively. However, the interannual
distributions of individual classes are quite distinct at different stations, particularly in the upper Songhua and Wusuli Rivers
of Songliao River Basin. At the headstream stations of Songliao River Basin (Figure 5b), the Class 4 is dominant with the
annual mean percentage of 26.1±38.3% (*n*=32) though flood events are missed in several years due to the dry period. The
dominance of Class 4 is the most considerable in 1996, 1998, 2002 and 2009 at the Muling station in the upper Wusuli River.
At the headstream stations of Yellow River Basin (Figure 5c), the Class 4 is also dominant across the whole period with the
annual mean percentage of 58.1±33.9% (*n*=67), particularly in 1994–1996, 1999 and 2007. The dominance of Class 4 is the
most considerable in 1993–1995 and 2001–2004 at the Huating station in the upper Jinghe River. At the headstream stations
of Huaihe River Basin (Figure 5d), the Class 5 gradually prevail with the annual mean percentage of 41.5±23.7% (*n*=102),
particularly after 2007, whose percentage reaches 63.2±15.8% (*n*=79). The dominance of Class 5 is the most considerable in
2007-2014 at the Beimiaoji station in the southern tributaries. The event numbers of both Classes 1 and 2 gradually decrease,
accounting for 33.1±24.4% (*n*=11) and 8.7±7.1% (*n*=5) of annual flood events in the period of 1993–1999 and 2011–2015 for
the Class 1, respectively, and 20.3±20.9% (*n*=9) and 2.7±1.3% (*n*=1) in the period of 1993-1999 and 2011-2015 for the Class

2, respectively. The decrease in Classes 1 and 2 are remarkable at the Peihe station in the southern tributaries and the Ziluoshan station in the northern tributaries, respectively. The explanations are that the total precipitation amount and duration probably

increase due to the climate change (Dong *et al.*, 2011; Jin *et al.*, 2024). At the headstream stations of Yangtze River Basin (Figure 5e), the Classes 1, 2 and 4 are dominant, accounting for 29.3±9.6% ($n$=251), 23.0±11.5% ($n$=197) and 21.1±7.0% ($n$=181) of annual mean flood events, respectively. Although the interannual changes of event numbers of Classes 1 ($n$=1–21), 2 ($n$=1–14) and 4 ($n$=1–16) are considerable, those of class percentages are relatively uniform except 2015. The class dominance is the most considerable in 1993, 1995–1997 and 1998 at the Yanling station in the Dongtinglake tributaries for

Class 1, in 1993, 1994 and 1997 at the Dutou station in the Poyanglake tributaries for Class 2, in 1998, 2000, 2001, 2004, 2005, 2007, and 2010–2013 at the Biyang station in the tributaries of Hanjiang River for Class 4, respectively. At the headstream stations of Southeast River Basin (Figure 5f), the Class 3 gradually prevail after 2000 with the annual mean percentage of 46.2±32.5% ($n$=39), which is remarkable at the Longshan station in the upper Jinjiang River. At the headstream stations of Pearl River Basin (Figure 5g), the Class 1 is dominant with the annual mean percentage of 36.0±24.0% ($n$=52), but gradually

shifts to Class 2 which accounts for 30.0±25.2% of annual mean flood events ($n$=40), particularly after 2008. The class dominance is the most considerable from 1993 to 2007 at the Hezikou station in the upper Dongjiang River for Class 1, and in 1993, 1994, 1996, 2005, 2006, and 2009–2011 at the Xiaogulu station in the upper Beijiang River for Class 2, respectively.

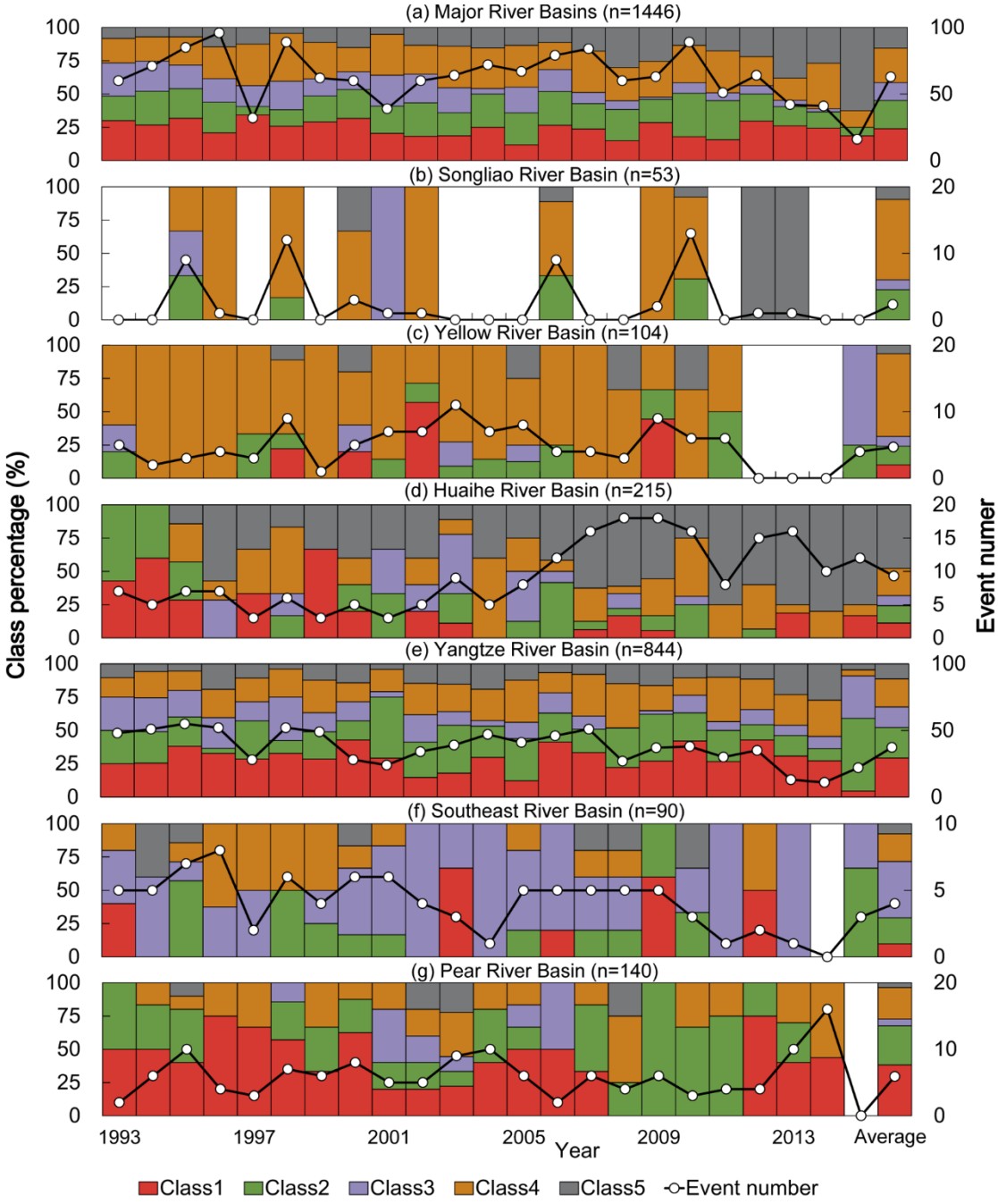

**Figure 5. Interannual variabilities of individual flood event classes (total events and their percentages) at headstream stations of major river basins**

**4.4 Control mechanisms of meteorological and physio-geographical factors**

**4.4.1 Control factors and their contributions for the distributed analysis**

According to the Monte Carlo permutation test between flood response matrix and control factor matrix in the individual catchments of Class 1, the total and mean precipitations, and drought index during the event ($r_{\text{pcp\_dur}}$=0.65–0.99, $n$=14; $r_{\text{pcp\_av}}$=0.70–0.97, $n$=7; $r_{\text{SPEI\_dur}}$=0.52–0.97, $n$=7) are the major control factors in 44.7% (17/38), 20% (1/5) and 25% (1/4) of total catchments of the Yangtze, Southeast and Pearl River Basins, respectively (Figure 6 and Table 4). The contributions of control factors are statistically significant only in the Liangshuikou catchment of the Yangtze River Basin and Hezikou catchment of the Pearl River Basin. In the Liangshuikou catchment, 96.3% of temporal differences are explained, in which the meteorological and land cover categories explain 92.5% and 3.8%, respectively. In the Hezikou catchment, 66.7% of temporal differences are explained, in which the meteorological category and the interactive impact explain 49.4% and 17.3%, respectively. The major control factors and their contributions for the Classes 2–5 are also presented in Text S1 and Figures S2–5 of the Supplement. For all the classes, only the factors in the meteorological category are statistically significant, particularly the precipitation amount and intensity, and drought index during the events. The most control factors with statistical significances are in Class 1, followed by Classes 4, 5, 3 and 2. These control factors for individual classes are detected mainly in the catchments of Yangtze (Class 1), Yellow and Pearl (Class 4), Huaihe (Class 5), Southeast (Class 3) and Pearl (Class 2) River Basins, respectively. The explanations are that the precipitation amount and potential evapotranspiration during the event usually show remarkable differences among different events, which directly determine the spatial and temporal heterogeneities of flood generation process, and consequently flood event hydrograph, but the land covers usually show slow changes in the headstream catchments due to slight disturbances of human activities and climate changes.

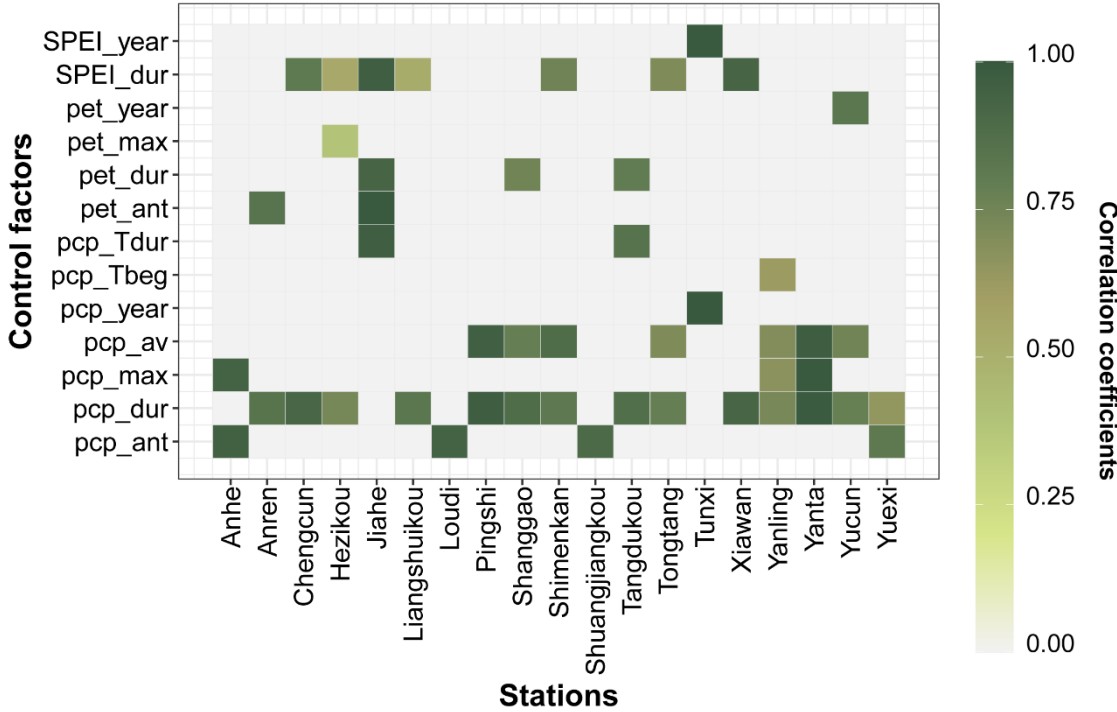

**Figure 6. Significant control factors and their correlation coefficients for the temporal variabilities of flood event Class 1 in the individual catchments. The gray color means the control factor without statistical significance.**

Note: Anhe, Anren, Chengcun, Jiahe, Liangshuikou, Loudi, Pingshi, Shanggao, Shimenkan, Shuangjiangkou, Tangdukou, Tongtang, Xiawan, Yanling, Yanta, Yucun and Yuexi catchments are from the Yangtze River Basin; Tunxi catchment is from Southeast River Basin; Hezikou catchment is from Pearl River Basin.

**Table 4. Effect contributions of control factor categories on the temporal variabilities of flood event classes in the individual catchments**

| Classes | Catchments | River Basins | Meteorology | Land cover | Interaction | Total |
|---------|------------|--------------|-------------|------------|-------------|-------|
| Class1 | Hezikou | Pearl | 49.4% | 0.0% | 17.3% | 66.7% |
| | Liangshuikou | Yangtze | 92.4% | 3.8% | 0.1% | 96.3% |
| Class2 | Shimenkan | Yangtze | 87.1% | 0.0% | 3.6% | 90.7% |
| | Tangdukou | Yangtze | 95.9% | 0.0% | 0.0% | 95.9% |
| | Xiaogulu | Pearl | 71.9% | 0.0% | 24.9% | 96.8% |
| Class3 | - | - | - | - | - | - |
| Class4 | Hezikou | Pearl | 82.1% | 0.0% | 16.0% | 98.1% |
| | Liangshuikou | Yangtze | 76.8% | 0.0% | 10.2% | 87.0% |
| Class5 | - | - | - | - | - | - |

380

### 4.4.2 Control factors and their contributions for the lumped analysis

The Monte Carlo permutation tests across the entire study area suggest that the meteorological category is also the most important (Figure 7), particularly the precipitation amount and intensity (i.e., pcp_ant, pcp_dur, pcp_max, pcp_av, pcp_Tbeg,

and pcp_Tdur), and the drought index during the events (SPEI_dur) with the correlation coefficients of 0.33–0.74, 0.20–0.38 and 0.29–0.41, respectively. The significant factor number in the catchment attribute category is less, which are mainly the mean catchment length (Length), river density (Rivden) and ratio of river width to depth (RivSlope) with the correlation coefficients of 0.18–0.32, 0.15–0.24 and 0.21–0.30, respectively. In the land cover category, only the grassland area ratio (Rgrass) is significant in the Class 1 with the correlation coefficient of 0.21.

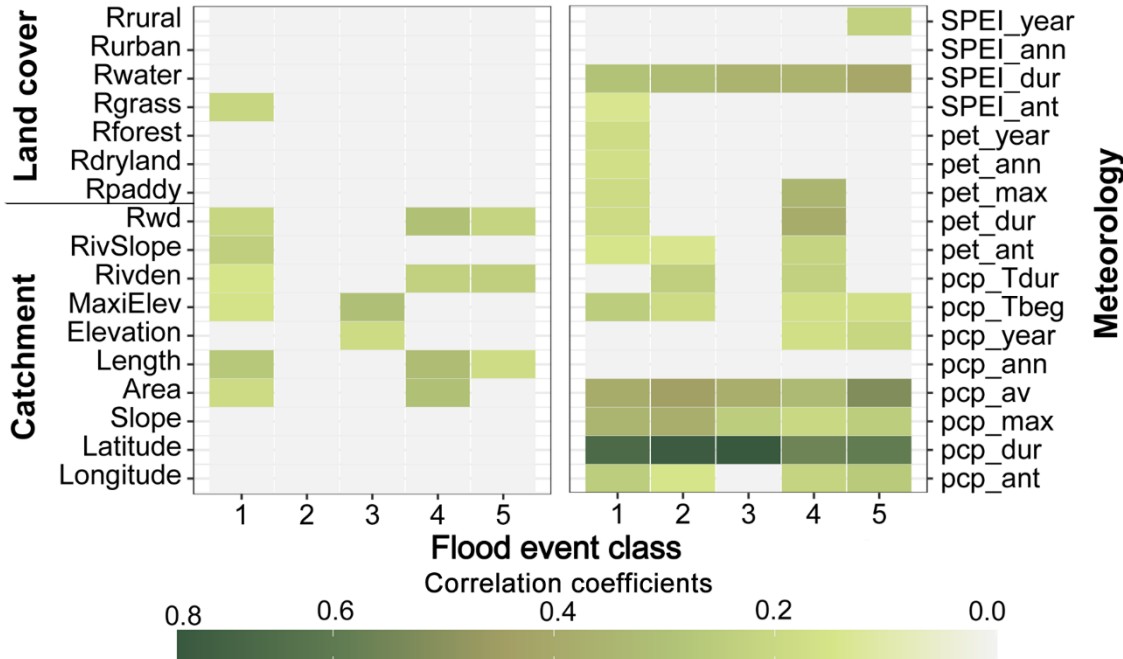

**Figure 7. Significant control factors and their correlation coefficients for the variabilities of individual flood event classes (i.e., Classes 1–5). The gray color means the control factor without statistical significance.**

In the Class 1, the significant control factors are the precipitation, potential evapotranspiration and drought index in the antecedent seven days (i.e., pcp_ant, pet_ant and SPEI_ant) and during the events (i.e., pcp_dur, pcp_av, pcp_max, pcp_Tbeg, pet_dur, pet_max and SPEI_dur), and the potential evapotranspiration at the annual scale (i.e., pet_ann and pet_year) in the meteorological category, the area (Area), mean length (Length), maximum elevation (MaxiElev), river density (Rivden) and slope (RivSlope) and ratio of river width to depth (Rwd) in the catchment attribute category, and the grassland area ratio (Rglass) in the land cover category. There are 72.7% of total spatial and temporal variabilities of flood events explained by all the control factor categories, in which 43.9% of total variabilities are explained by the meteorological category (particularly the factors during the events), followed by the interactive impact (22.7%), catchment attribute category (4.2%) and land cover category (1.5%), respectively (Figure 8a).

The significant control factors of Class 2 are mainly in the meteorological factor category, including precipitation and potential evapotranspiration in the antecedent seven days (i.e., pcp_ant and pet_ant), precipitation and drought index during the flood events (i.e., pcp_dur, pcp_av, pcp_max, pcp_Tbeg, pcp_Tdur and SPEI_dur). In the Class 3, the significant control factors are mainly the precipitation and drought index during the flood events (i.e., pcp_dur, pcp_av, pcp_max and SPEI_dur) and catchment elevation (i.e., Elevation and MaxiElev). In the Classes 4 and 5, most of the meteorological and catchment factors are significant. The specific factors are the precipitation and potential evapotranspiration in the antecedent seven days and during the events (i.e., pcp_ant, pcp_dur, pcp_av, pcp_max, pcp_Tbeg, pcp_Tdur, pet_ant, pet_dur and pet_max), drought index during the events (i.e., SPEI_dur) and precipitation at the annual scale (i.e., pcp_year) for the meteorological factor category, and the catchment area (Area), mean length (Length), river density (Rivden) and ratio of river width to depth (Rwd) in the catchment attribute category for the Class 4, and precipitation factors (i.e., pcp_ant, pcp_dur, pcp_av, pcp_max, pcp_Tbeg and pcp_year), drought index during the events and at the annual scale (i.e., SPEI_dur and SPEI_year) for the meteorological factor category, and the catchment mean length (Length), river density (Rivden) and ratio of river width to depth (Rwd) in the catchment attribute category for the Class 5. For the entire contributions of all the control factors or categories, 73.3%, 85.4%, 65.9% and 65.7% of total spatial and temporal variabilities of flood events are significantly explained in the Classes 2–5, respectively (Figure 8b–e). For the individual contributions, the meteorological factor category explains the largest variabilities (i.e., 36.5%–50.5%), followed by the catchment attribute category (i.e., 5.1%–6.1%), and the land cover category explains the least variabilities, i.e., 0.0–2.4%. The interactive impacts of all the control factor categories also explain 17.5%–33.0% of total variabilities, particularly in the Class 3.

Therefore, the total variabilities of flood events in the Class 1 are mainly controlled by the total precipitation amount and its intensity during the events which determine the magnitudes of total flood yield and flood peak, the catchment slope length and river slope which affect the flood routing processes, e.g., total duration of flood event and occurrence time of flood peak. The total variabilities in the Class 2 are also mainly controlled by the total precipitation amount and its intensity during the events. The total variabilities in the Class 3 are mainly controlled by the total precipitation amount, its intensity and the drought index during the events which determine the total magnitudes and occurrence time of flood yield, and the catchment elevation which determine the flood routing time. The total variabilities in the Class 4 are mainly controlled by the total precipitation amount, potential evapotranspiration and the drought index during the events which determine the total magnitude and occurrence time of flood yield, and evapotranspiration, as well as the catchment area, slope and river morphology which determine the flood routing time and river storage capacity. The total variabilities in the Class 5 are mainly controlled by the total precipitation amount and the drought index during the events which determine the total magnitudes and occurrence time of flood yield, as well as the river density which determine the flood routing time in the river system.

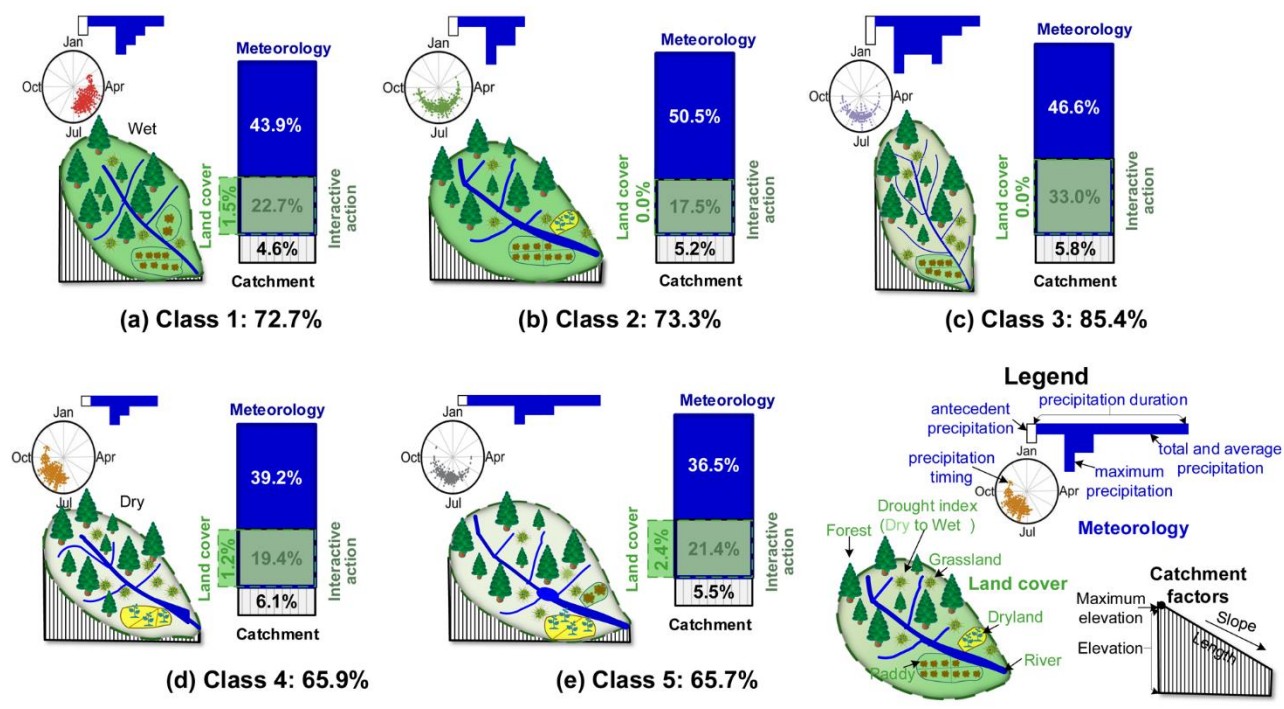

**Figure 8. Entire, individual and interactive contributions of control factor categories on the spatial and temporal variabilities of flood event classes 1–5 (a–e)**

### 4.4.3 Control mechanisms in the individual flood event classes

In both the individual catchments and the entire region, the dominant control factors of all the flood event classes are the total and mean precipitation volumes, the maximum precipitation intensity, the drought index and the precipitation timing during the events, the precipitation in the antecedent days in the meteorological category (Figures 9 and S6 in the Supplement). Therefore, the flood events in Class 1 are mainly caused by the rainfall with low volume and intensity before the wet season in the wet, steep and low-latitude catchments. The events in Class 2 are mainly caused by the short rainfall with high mean intensity in the wet low-latitude catchments. The events in Class 3 are mainly caused by the long rainfall with high volume and intensity in the small catchments of high altitude and low latitude. The events in Class 4 are mainly caused by the short rainfall with low volume and intensity in the latter wet season in the dry, steep and small catchments of high altitude and latitude. The events in Class 5 are mainly caused by the long rainfall with high volume and low mean intensity in the dry, gentle and large mid-latitude catchments.

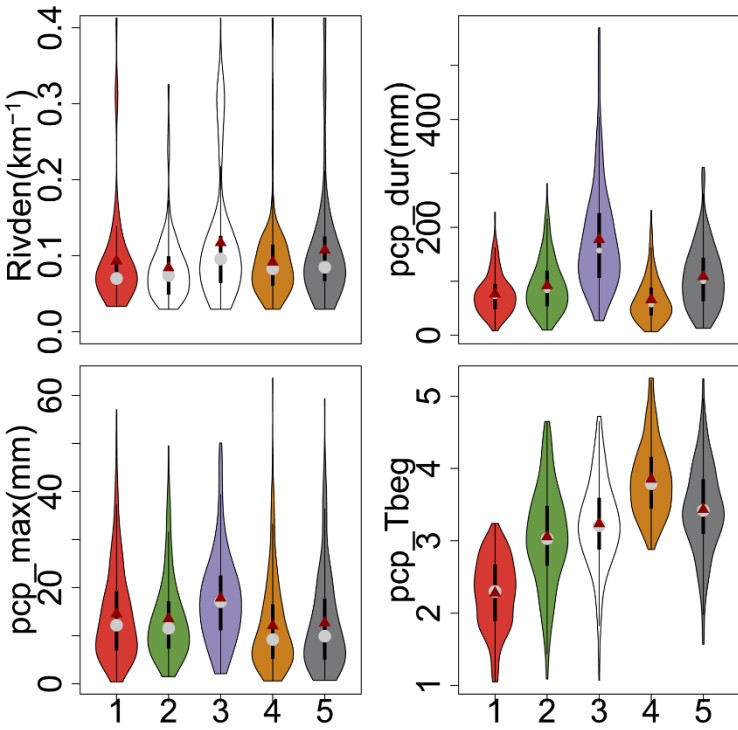


**Figure 9. Variations of four critical control factors among Classes 1–5. The solid darkred dot and gray dot define the mean and 50th percentile values, respectively. Each black box means the 25th and 75th percentile values, and the vertical line defines the minimum and maximum values without outliers. The violin shape means the frequency distribution of control factor, and the unfilled shape means the control factor without statistical significance.**

## 5. Discussion


Flood classification has strong advantages in systematically identifying manageable classes from a large number of historical flood events based on the similarity of flood response characteristics (Arthington *et al.*, 2006; Kuentz *et al.*, 2017; Poff *et al.*, 2007; Sikorska *et al.*, 2015; Sivakumar *et al.*, 2015). Flood events in the same class are widely accepted to have similar hydrological responses caused by similar meteorological or underlying surface conditions (Sikorska *et al.*, 2015). Therefore,

it is more efficient to investigate flood event changes and their cause mechanisms in a comprehensive manner than individual event analyses (Zhang *et al.*, 2012). It is expected to provide more useful flood response characteristics for flood disaster management purposes (e.g., early warning and quick design of flood control plans) and provide deep insights to investigate riverine ecological and environmental response mechanisms.

In our study, the flood event classes are identified based on the entire flood response characteristics, which cover not only the flood magnitude metrics (e.g., large, moderate and small floods) but also the event shape metrics (e.g., fast or slow floods).

Therefore, our study captures more detailed response dynamics of flood events than the predefined classes reported by several existing studies, such as flash floods, short-rain floods, rain-on-snow floods or snowmelt floods (Brunner *et al.*, 2018; Merz and Blöschl, 2003; Sikorska *et al.*, 2015). The specific values and boundaries of flood response metrics of individual classes

were difficult to quantitatively compare with most existing studies because the adopted classification methods were usually different. However, the flood event classes with similar hydrographs or response mechanisms were also found in the existing studies. Classes 1 and 2 are mainly in the southern China, particularly in the Pearl and Yangtze River Basins, which are controlled by the temperate climate without a dry season. Storms with high intensities and short durations before the wet season in the southern China are likely to cause flood events with great magnitudes and variabilities (Class 1) or fast flood events with

a high single peak and short durations (Class 2) (Gao *et al.*, 2018). The flood response characteristics in these two classes are similar to the flash floods and short-rain floods in Austria (Merz and Blöschl 2003), and fast events in Switzerland (Brunner *et al.*, 2018) and China (Zhai *et al.*, 2021). Class 3 is mainly in the Southeast River Basin controlled by the tropical cyclone climate. Severe storms with high intensities and durations are likely to cause high slow flood events with multiple peaks (Class 3) (Yin *et al.*, 2010; Zhang *et al.*, 2020). The flood response characteristics are similar to the high unit peak flood in the west

coast of the USA (Saharia *et al.*, 2017) because both the response characteristics were mainly controlled by subtropical or tropical storms near the ocean in the Cf climate type. They are also similar to the slow events in China (Zhai *et al.*, 2021) because the rates of positive changes are 0.01–0.94 $h^{-1}$ in our study and 0.04–1.78 $h^{-1}$ in China (Zhai *et al.*, 2021), and the rates of negative changes are 0.01–0.33 $h^{-1}$ in our study and 0.02–0.25 $h^{-1}$ in China (Zhai *et al.*, 2021). Class 4 is mainly in the northern China controlled by the cold climate with dry winters. The heavy storms ahead of westerlies trough mainly occur in

the latter wet season in this region, which usually have low intensities and short durations (Gao *et al.*, 2018). Thus, they are likely to cause the small fast flood events (Class 4), whose mean flood peak magnitude and coefficients of variation are 0.47 $m^3/s/km^2$ and 0.86, respectively. The similar flood events are also reported, e.g., the low flashiness floods with the mean flood peak magnitude of 0.20–0.25 $m^3/s/km^2$ and the mean coefficients of variation of approximate 0.90 in the northern part of central–eastern Europe (Kuentz *et al.*, 2017), which is also controlled by the similar climate type (i.e., Df). Class 5 is mainly

in the south–north climate zone of China (i.e., Huaihe River Basin), which has the dual climate characteristics of both south and north monsoons. Storms characterized by a long period of continuous rainy meteorological with high frequency and low intensities (e.g., Meiyu rainfalls) in the earlier wet season are likely to cause moderate slow flood events with long durations (Gao et al., 2018; Sampe and Xie, 2010). The flood response characteristics are similar to the intermediate flood events in China (Zhai *et al.*, 2021). For example, the coefficients of variation are 0.65–3.15 in our study and 0.78–3.07 in China (Zhai

*et al.*, 2021). The rates of positive and negative changes are 0.02–8.00 $h^{-1}$ and 0.01–0.64 $h^{-1}$ in our study, respectively, while those reported in Zhai *et al.* (2021) were 0.36–4.90 $h^{-1}$ and 0.09–0.46 $h^{-1}$ in China, respectively. Therefore, the classification is helpful to deeply investigate the control mechanisms of flood events, which is easy to transfer to predict flood events with similar control factors (Sikorska *et al.*, 2015).

The meteorological, land cover and catchment attribute categories are mainly reported to affect the flood generation and routing processes, and could be widely-accepted as the critical control factors of spatial and temporal differences of flood event classes (Ali *et al*., 2012; Brunner *et al*., 2018; Merz and Blöschl, 2003; Zhang *et al*., 2022). Our results also find that the meteorological factor category is dominant, which explain 49.4–95.9% and 36.5–50.5% of the flood event differences in individual classes at catchment scale and in the entire region, respectively. Similar results were reported in Kuentz *et al*. (2017), which are that the
climatic variables (e.g., precipitation, temperature and drought index) play the most important role for 75% of total flow signatures and catchment attributes (e.g., area, elevation, slope and river density) are more important for the flood flashiness. The main significant meteorological factors are the precipitation volume, intensity and the drought index during the events. The main explanation is that the precipitation and drought index during the flood events directly affect the hydrograph through flood generation, e.g., total volume and peak, variability, duration, rate of changes and peak number (Merz and Blöschl, 2003;
Aristeidis *et al*., 2010). Additionally, these control factors in the antecedent days directly affect the antecedent soil moisture, which determine the initial losses of precipitation and the runoff generation timing during the flood events (Hall and Blöschl, 2018; Xu *et al*., 2023). The contribution of meteorological factor category is the largest in the Class 2, particularly in the Tangdukou catchment of Yangtze River Basin because the flood events in this class usually show quick responses to the precipitation, while the contribution is the lowest in the Class 5 because the river density and river morphology play important
roles in the flood storage capacity and routing time in the river system.

Secondly, the catchment attributes (e.g., geographical location and topography) mainly affect the hydrograph patterns through flood routing (Berger and Entekhabi, 2001; Ali *et al*., 2012), and the identified factors in our study are the catchment area and length, river density and ratio of river width to depth. For example, a catchment with longer routing length, larger routing area,
river density and ratio of river width to depth usually has larger flood regulation and storage capacity, and thus generates the slow flood events, while a catchment with shorter routing length, smaller routing area, river density and ratio of river width to depth usually has weaker flood regulation and storage capacity, and thus generates the fast flood events (Zhang *et al*., 2020). However, the comprehensive contributions of catchment attributes are not considerable, i.e., only 0.0–6.1% in the entire region because the catchment attributes do not always well match the flood event responses (Kuentz *et al*., 2017; Ali *et al*., 2012).
The contributions of catchment attribute category in the slow flood event classes (e.g., Classes 3 and 5) are usually larger than those in the fast flood event classes (e.g., Classes 1, 2 and 4) because the catchment attribute factors are significantly correlated with the flood response metrics in the Classes 3 and 5, particularly the catchment maximum elevation and river density. Furthermore, the location, annual precipitation, potential evapotranspiration and drought index mainly affect the overall catchment hydrological conditions (Berger *et al*., 2001; Kennard *et al*., 2010). Finally, the land covers mainly determine the
precipitation intercept and retention processes, which directly affect the flood variability and rate of changes (Kuentz *et al*., 2017; Merz *et al*., 2020). For example, catchments with greater vegetation covers (e.g., forest, grassland) usually generate the slow flood events, while catchments with weaker vegetation covers (e.g., rural and urban lands) usually generate the fast flood events (Kuentz *et al*., 2017; Zhai *et al*., 2021). However, all the catchments selected in our study are mainly in the river source

regions with good vegetation coverages with mean area percentages of 67.0% for forest and 6.6% for grassland. The spatial
and temporal differences of land covers are not remarkable so that it only explains 3.8% and 1.5% of the flood event differences
in Class 1 at the Liangshuikou catchment of Yangtze River Basin and in the entire region, respectively.

Our study provided an approach to investigate some manageable flood event classes from massive events at large scale and to
quantify the meteorological and physio-geographical controls of spatial and temporal variabilities of flood event classes. The
approach could be applied easily to other regions or countries if a great number of flood events were collected. All the selected
flood events were sufficient to represent the flood response characteristics of headstream catchments in main river basins of
China. Thus, our classification results and the control mechanisms of variability of flood event classes would be applied in
other regions with similar climate types. However, several works should be paid attention for further improvements of our
study. Firstly, total flood event number is the main restricted factor for the classification performance, the flood event class
representativeness and their control mechanisms at catchment scale (Merz and Blöschl, 2003; Olden *et al.*, 2012; Sikorska *et
al.*, 2015; Tarasova *et al.*, 2020). It could be overcome effectively by adopting large flood event numbers of individual classes
(i.e., approximately 10% of total events at least in our study) for the classification (Zhang *et al.*, 2020). However, not all the
control mechanisms of flood event classes were well explained because of the insufficient flood events, which were mainly in
the Songliao and Yellow River Basins, or most catchments expect the Shimenkan, Liangshuikou and Tangdukou catchments
of the Yangtze River Basin, Xiaogulu and Hezikou catchments of the Pearl River Basin. The representatives of individual
classes should be further investigated particularly in the basins with low densities of flood events. Secondly, the class
boundaries of most flood response metrics were not clear using the inductive classification approaches (Parajka *et al.*, 2005;
Sikorska *et al.*, 2015), e.g., the flood magnitude, rates of positive and negative changes in our study. Although the predefined
the sharp thresholds of all the flood response metrics are beneficial to clearly separate the flood events using the classification
tree methods (e.g., decision tree, crisp tree), the predefinition is still challenging (Sikorska *et al.*, 2015; Brunner *et al.*, 2017;
Tarasova *et al.*, 2020). Finally, the control mechanism deduction was mainly based on the statistical detection of control factors
and their contributions. The interactive impacts of different control factor categories were still difficult to be clearly explained
using the adopted statistical analysis method (i.e., the constrained rank analysis in our study).

## 6. Conclusions

In our study, the main flood event classes characterized by multiple flood response metrics are identified in 68 headstream
catchments using the hierarchical and partitional clustering methods. The control mechanisms of different flood event classes
are investigated using the constrained rank analysis and Monte Carlo permutation test. Results are summarized as follows: the
partitional clustering method (i.e., *k*-medoids) performs better than the hierarchical method, and the optimal five flood event
classes are identified which are the moderately fast flood event class (Class 1), the highly fast flood event class (Class 2), the
highly slow and multipeak flood event class (Class 3), the slightly fast flood event class (Class 4) and the moderately slow

flood event class (Class 5). Most of the flood event differences among individual classes are explained by the meteorological, land cover and catchment attribute factors. The flood event differences in Class 3 (85.4%) are well explained, followed by Classes 2 (73.3%), 1 (72.7%), 4 (65.9%) and 5 (65.7%). The meteorological category is the most significant among all the control factors, particularly the precipitation factors (e.g., volume, intensity) and drought index during the flood events.


This study preliminarily investigates the flood event classes in space and time at some headstream stations of China, which is beneficial to explore the comprehensive formation mechanisms of flood events and the critical control factors, and provides the scientific foundation for flood event prediction and control. In future, more unimpaired flood events could be collected to strengthen the representativeness of flood event classes, and to further support the control mechanism analysis of flood classes

at individual catchments. The interactive impacts of control factor categories could also be further decomposed into the impacts of individual factors using the hydrological model with strong physical mechanism.

**Appendix A:**

All the multivariable statistical analyses are implemented using R software (version 3.1.1) (R Development Core Team, 2010), involving the `aov`, `cor` and `princomp` functions in stats Package (version 4.1.3) for independence test, linear correlation test and principal component analysis, respectively (Mardia *et al*., 1979), the `hcluster` function in amap Package (version

test and principal component analysis, respectively (Mardia *et al*., 1979), the `hcluster` function in amap Package (version 0.8-18) for hierarchical cluster analysis (Antoine and Sylvain, 2006), the `clara` function in cluster Package (version 2.1.3) for *k*-medoids cluster analysis (Kaufman and Rousseeuw, 1990), the `NbClust` function in NbClust Package (version 3.0.1) for the optimal class number determination and classification performance assessment (Charrad *et al*., 2014). The Monte Carlo permutation test are implemented using the `envfit`, `decorana`, `rda`, `cca`, `permutest` functions in the vegan Package

(version 2.5-7) of R software (version 3.1.1) (ter Braak, 1986; R Development Core Team, 2010).

**Appendix B:**

The optimal classification method and cluster number are determined by comparing the classification performance between the hierarchical and *k*-medoids clustering methods among individual cluster numbers. Figure B1 shows that the optimal criteria number is the largest when the cluster number is five (i.e., 22.7% of total) for the *k*-medoids clustering method. The optimal

criteria are the CCC, TrCovW, Silhouette, Ratkowsky and PtBiserial with the values of -2.98, $1.39\times10^{15}$, $4.12\times10^{6}$, 0.20, 0.29 and 0.39, respectively. Therefore, the five clusters using the *k*-medoids clustering method are optimum for further analysis in our study. The flood event numbers in the individual classes are 347, 306, 195, 375 and 223, accounting for 24.0%, 21.2%, 13.5%, 25.9% and 15.4% of total events, respectively.

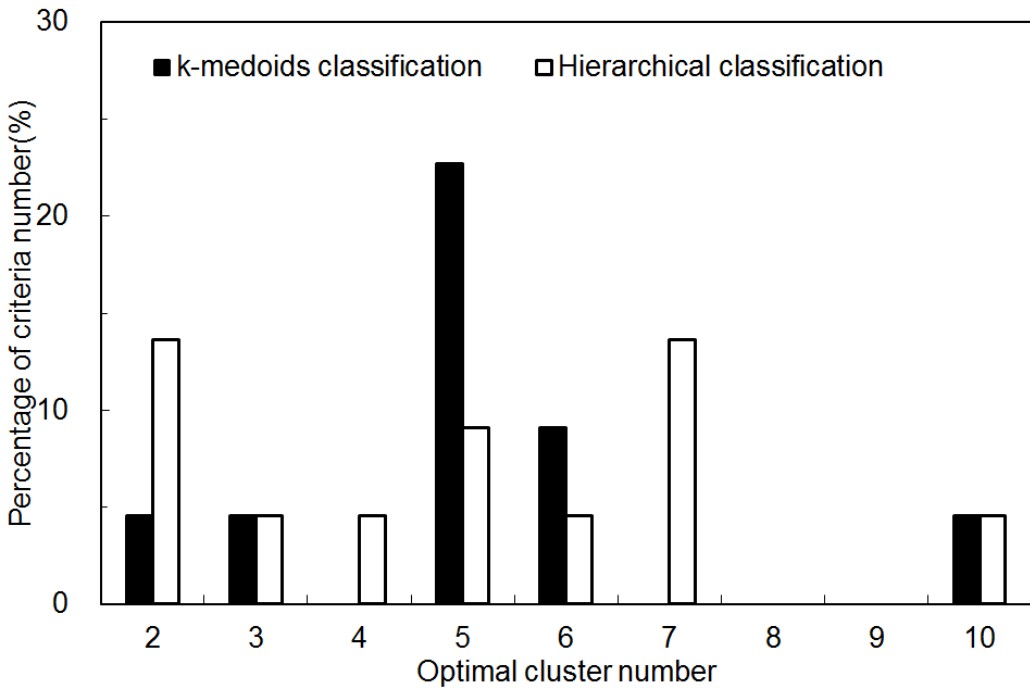


**Figure B1. Classification performance comparisons between the hierarchical and *k*-medoids methods among individual optimal cluster numbers**

**Acknowledgements**

This study was supported by the National Natural Science Foundation of China (No. 42171047), the National Key Research and Development Program of China (No. 2021YFC3201102) and the CAS-CSIRO Partnership Joint Project of 2024 (No. 177GJHZ2023097MI). We extend our gratitude to the editors and anonymous reviewers for their insightful feedback and constructive comments.

**Author contribution**

**Yongyong Zhang**: Conceptualization, Methodology, Formal Analysis, Writing-Original draft preparation, Writing-Reviewing and Editing, Funding acquisition; **Yongqiang Zhang**: Conceptualization, Writing-Reviewing and Editing; **Xiaoyan Zhai:** Data curation, Formal Analysis, Writing-Reviewing and Editing, Funding acquisition; **Jun Xia:** Conceptualization, Writing-Reviewing and Editing; **Qiuhong Tang:** Conceptualization, Writing-Reviewing and Editing; **Wei Wang:** Data Processing, Formal Analysis; **Jian Wu:** Formal Analysis; **Xiaoyu Niu:** Formal Analysis; **Bing Han:** Formal Analysis

## Code/Data availability

The geographic information system data sources are obtained from the Data Center of Resources and Environmental Science, Chinese Academy of Sciences (http://www.resdc.cn/). The historical flood events and synchronous precipitation are collected from the Hydrological Yearbooks of the Songliao, Yellow, Huaihe, Yangtze, Southeast and Pearl River Basins which are published by the Ministry of Water Resources of the People's Republic of China (http://xxfb.mwr.cn/sq_djdh.html). The daily precipitation and temperature observations are collected from the China Meteorological Data Sharing Service System (http://cdc.cma.gov.cn/home.do).

## Competing interests

The authors declare no conflicts of interest relevant to this study.

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
