# Peer review of "Understanding meteorological and physio-geographical controls of variability of flood event classes in headstream catchments of China"

_Hydrology and Earth System Sciences, 2024_

## Author Comment (AC1)

**RC1: 'Comment on hess-2024-126', Anonymous Referee #1, 09 Jun 2024**

The paper provides a comprehensive analysis of three primary classifications for a catchment: meteorological, attributes, and response. By correlating this information, the paper identifies characteristic classes of flood responses. The main findings show that meteorological data has a much greater impact on flood response compared to land cover and catchment attributes. However, certain catchment attributes were also found to be correlated with the response.

**Response:** Thank you very much for your careful review and constructive comments. We revised this manuscript substantially and provided point-by-point responses to all the comments and suggestions of reviewers accordingly.

 Here are my main concerns about this paper:

1.  The results don't contribute new knowledge about the streamflow-generating process. It's well known that streamflow is mainly controlled by factors such as precipitation, intensity, duration, and its distribution. A similar analysis using the rational method could yield the same results as presented in this paper.

**Response:** We appreciate your critical comments. In our study, the main motivations are to investigate some manageable flood event classes from massive events across China with statistical significance and to quantify the meteorological and physio-geographical controls of spatial and temporal variabilities of these flood event classes using the clustering method, constrained rank analysis and Monte Carlo permutation test. We agreed that this study did not contribute new mechanisms about the streamflow-generating process because the investigation was quite difficult from massive heterogenous flood events in space and time at large scale. However, existing

studies usually focused on impacts of changes in meteorological or underlying surface conditions on specific flood metrics (e.g., magnitude, peak and timings) and their changes using trend separation method, correlation testing, mathematical modelling, and so on (Berghuijs *et al*., 2016; Tarasova *et al*., 2018; Liu *et al*., 2020). All of these studies were implemented at event scale or in catchments with certain landscapes and climates, which were insufficient for the comprehensive flood change investigation and generalized results (Tarasova *et al*., 2019; Zhang *et al*., 2020). Therefore, we explored the control mechanisms of meteorological and physio-geographical factors on spatial and temporal variabilities of flood events at class scale across China. The primary meteorological and physio-geographical control factors were identified for different flood event classes clustered from over one thousand flood events, and their contributions of the class variabilities were quantified for individual classes. All of these analyses were implemented in more heterogeneous catchments with wider meteorological and physio-geographical conditions and flood events, and provided more comprehensive insights into meteorological and physio-geographical controls of variabilities of flood event classes in China.

To make the novelty and contributions of our studies clearer, we made several revisions. The manuscript was revised as follows: *"Our study investigates comprehensive manageable flood event classes from 1446 unregulated flood events in 68 headstream catchments in China using the hierarchical and partitional clustering methods. Control mechanisms of meteorological and physio-geographical factors (e.g., meteorology, land cover and catchment attributes) on spatial and temporal variabilities of individual flood event classes are explored using constrained rank analysis and Monte Carlo permutation test."*

*"Existing studies provide insights on impacts of changes in meteorological or underlying surface conditions on specific flood metrics (e.g., magnitude, peak and timings) and their changes using trend*

*separation method, correlation testing, mathematical modelling, and so on (Berghuijs et al., 2016; Tarasova et al., 2018; Liu et al., 2020; Wang et al., 2024). However, all of these studies are implemented at event scale or in catchments with certain landscapes and climates, which are insufficient for the comprehensive flood change investigation and generalized results (Tarasova et al., 2019; Zhang et al., 2020)."*

*"Over one thousand unregulated flood events at 68 heterogeneous catchments with wider meteorological and physio-geographical conditions are selected for our study."*

*"This study provides more comprehensive insights into meteorological and physio-geographical controls of variabilities of flood event classes at large scale, and provides the mechanism supports for predicting flood event classes."*

*References*

*Berghuijs, W. R., Woods R. A., Hutton C. J., and Sivapalan M.: Dominant flood generating mechanisms across the United States, Geophys. Res. Letters, 43, 4382–4390, http://doi.org/10.1002/2016GL068070, 2016.*

*Liu, J.Y., Feng, S.Y., Gu, X.H., Zhang, Y.Q., Beck, H.E., Zhang, J.W., and Yan, S.: Global changes in floods and their drivers, J. Hydrol., 614, 128553. https://doi.org/10.1016/j.jhydrol.2022.128553, 2020*

*Tarasova, L., Basso, S., Zink, M. and Merz, R.: Exploring controls on rainfall-runoff events: 1. Time series-based event separation and temporal dynamics of event runoff response in Germany, Water Resour. Res. 54, 7711–7732, https://doi.org/10.1029/2018WR022587, 2018.*

*Wang, H., Liu, J.G., Klaar, M., Chen, A.F., Gudmundsson, L., and Holden, J.: Anthropogenic climate change has influenced global river flow seasonality, Science, 383(6686), 1009-1014, https://doi.org/10.1126/science.adi9501, 2024.*

2.    While the classification found in the paper may have value for local or basin analysis, most of the results cannot be applied to other regions or countries. The attempt to connect with other countries in the discussion is qualitative and not valid for

comparison without quantitative analysis. What is considered high or low, fast or slow in one country could be entirely different in another.

**Response:** Thank you very much for your constructive comments.

**For the applicability of our study,** we provided an approach to investigate some manageable flood event classes from massive events at large scale and to quantify the meteorological and physio-geographical controls of spatial and temporal variabilities of flood event classes. The approach could also be applied easily to other regions or countries if a great number of flood events were collected. The main motivations and implications of this study were clarified as follows: *"This study provides more comprehensive insights into meteorological and physio-geographical controls of variabilities of flood event classes at large scale, and provides the mechanism supports for predicting flood event classes."* in the introduction section, and *"Our study provided an approach to investigate some manageable flood event classes from massive events at large scale and to quantify the meteorological and physio-geographical controls of spatial and temporal variabilities of flood event classes. The approach could be applied easily to other regions or countries if a great number of flood events were collected." i*n the discussion section.

**For the comparability of our study,** we agreed that the results were difficult to quantitatively compare with most existing studies because the adopted classification methods and boundaries of individual classes were usually different. The widely-adopted classification method categories were presented in the revision to explain the comparability of classification results. *"According to the classification procedure, there are two widely-adopted approaches, namely the tree clustering methods (e.g., decision tree, regression tree, fuzzy tree and random forest) (Sikorska et al., 2015; Brunner et al., 2017) and the non-tree clustering methods (e.g., single linkage, complete linkage, average linkage, centroid linkage, ward linkage, k-mean, k-medoids) (Zhang et al., 2020; Zhai et al., 2021). The tree clustering methods as the hard clustering methods, are implemented to binarily split all the flood events successively into smaller classes of similar*

*flood events according to the thresholds of flood response metrics until obtaining final classes (Sikorska et al., 2015; Brunner et al., 2017). The classification results could be applicable to other basins and the flood response characteristics of different studies would be directly comparable if the same thresholds are adopted. However, these methods assume that the boundaries of flood response metrics in different classes are clear and the thresholds of flood response metrics should be predefined and should not overlap among different classes (Olden et al., 2012; Sikorska et al., 2015; Zhai et al., 2021). Additionally, the classification is very sensitive to the thresholds, whose small changes would cause different flood event classes (Olden et al., 2012; Sikorska et al., 2015). Therefore, it will be difficult to define the thresholds clearly to get robust classification performance. The non-tree clustering methods as the soft clustering methods, are implemented to directly split all the flood events according to different division rules of the comprehensive similarity measures of flood event shapes or metrics (Olden et al., 2012; Zhang et al., 2020). The class boundaries of flood response metrics are not clear, which are mainly based on sufficient of heterogeneous flood events (Sikorska et al., 2015). The flood response characteristics of individual classes were usually qualitatively described to distinguish the differences among classes (Olden et al., 2012; Tarasova et al., 2019; Zhang et al., 2020). Therefore, the classification results obtained from different flood event samples are still difficult to quantitatively compare even though the flood response characteristics or hydrographs in the certain class are similar (e.g., high or low, fast or slow floods) (Zhang et al., 2024). However, these methods were widely-used due to their ease of use (Olden et al., 2012; Tarasova et al., 2019; Zhang et al., 2020)."*

**We also discussed the reliability of our classifications in China and tried to make quantitative comparisons with the existing studies of other regions.** In our study, a total of 1446 unregulated flood events in 68 headstream catchments were selected for classification. All the catchments were mainly spread across the flood-prone areas and in all the monsoon controlled climate types of China, except tropical climate in the islands (i.e., A). The selected flood events were sufficient to represent the flood response characteristics of headstream catchments in main river basins of China. Thus, our classification results and the control mechanisms of variability of flood event

classes would be applied in other regions with similar climate types. The revisions were given as follows: "*thus the region in the monsoon controlled climate types is usually considered as the flood-prone area of China (China Institute of Water Resources and Hydropower Research and Research Center on Flood and Drought Disaster Prevention and Reduction, the Ministry of Water Resources, 2021).*" and "*Sixty-eight headstream stations spread across the flood-prone areas were selected with catchment areas ranging from 21 $km^2$ to 4830 $km^2$, which were in all the monsoon controlled climate types of China, except tropical climate in the islands (i.e., A).*" in the section of study area and data sources, and "*All the selected flood events were sufficient to represent the flood response characteristics of headstream catchments in main river basins of China. Thus, our classification results and the control mechanisms of variability of flood event classes would be applied in other regions with similar climate types.*" in the discussion section.

The values of some critical metrics of individual classes were also quantitatively compared with those of existing studies in the discussion section. The revisions were given as follows: "*The specific values and boundaries of flood response metrics of individual classes were difficult to quantitatively compare with most existing studies because the adopted classification methods were usually different. However, the flood event classes with similar hydrographs or response mechanisms were also found in the existing studies. …… The flood response characteristics in these two classes are similar to the flash floods and short-rain floods in Austria (Merz and Blöschl 2003), and fast events in Switzerland (Brunner et al., 2018) and China (Zhai et al., 2021).*"

"*The flood response characteristics are similar to the high unit peak flood in the west coast of the USA (Saharia et al., 2017) because both the response characteristics were mainly controlled by subtropical or tropical storms near the ocean in the Cf climate type. They are also similar to the slow events in China (Zhai et al., 2021) because the rates of positive changes are 0.01–0.94 $h^{-1}$ in our study, and 0.04–1.78 $h^{-1}$ in China (Zhai et al., 2021), and the rates of negative changes are 0.01–0.33 $h^{-1}$ in our study and 0.02–0.25 $h^{-1}$ in China (Zhai et al., 2021).*"

"*The similar flood events are also reported, e.g., the low flashiness floods with the mean flood peak magnitude of 0.20–0.25 $m^3/s/km^2$ and the mean coefficients of variation of approximate 0.90 in the northern part of central–eastern Europe (Kuentz et al., 2017), which is also controlled by the similar climate type (i.e., Df).*"

*"The flood response characteristics are similar to the intermediate flood events in China (Zhai et al., 2021). For example, the coefficients of variation are 0.65–3.15 in our study and 0.78–3.07 in China (Zhai et al., 2021). The rates of positive and negative changes are 0.02–8.00 $h^{-1}$ and 0.01–0.64 $h^{-1}$ in our study, respectively, while those reported in Zhai et al. (2021) were 0.36–4.90 $h^{-1}$ and 0.09–0.46 $h^{-1}$ in China, respectively."*


**Response:** Your suggestion has been adopted. We summarized our results in a higher-level way and moved some detailed information into the supplementary tables (Tables S4 and S5). The examples were as follows:

*"Table S4. Average, standard deviation, median, maximum and minimum of flood response metrics in different classes*

| Characteristic value | Class | $R(mm \cdot day^{-1})$ | $Q_{pk}(mm \cdot day^{-1})$ | CV | $T_{bgn}$ | $T_{pk}(\%)$ | $T_{drn}(h)$ | $RQ_r(h^{-1})$ | $RQ_d(h^{-1})$ | $N_{pk}$ |
|---|---|---|---|---|---|---|---|---|---|---|
| Average± Standard Deviation | 1 | 43.97±29.94 | 2.04±2.51 | 0.90±0.26 | 2.28±0.49 | 27.14±9.60 | 103.92±43.39 | 0.13±0.32 | 0.04±0.07 | 1.31±0.51 |
| | 2 | 45.81±34.01 | 2.21±2.52 | 0.87±0.25 | 3.06±0.69 | 50.64±10.28 | 83.82±41.20 | 0.08±0.14 | 0.08±0.12 | 1.32±0.50 |
| | 3 | 143.97±108.33 | 5.23±6.04 | 0.84±0.22 | 3.24±0.61 | 33.90±15.02 | 145.26±68.99 | 0.25±0.62 | 0.12±0.28 | 2.67±0.76 |
| | 4 | 33.31±26.64 | 1.69±2.11 | 0.86±0.26 | 3.85±0.51 | 26.11±9.09 | 85.73±39.97 | 0.14±0.30 | 0.04±0.08 | 1.24±0.43 |
| | 5 | 65.79±43.80 | 2.98±3.68 | 1.40±0.43 | 3.43±0.61 | 23.74±13.60 | 202.88±85.42 | 0.18±0.62 | 0.03±0.04 | 1.24±0.46 |
| Median | 1 | 35.63 | 1.17 | 0.89 | 2.30 | 27.27 | 97.01 | 0.05 | 0.02 | 1.00 |
| | 2 | 37.84 | 1.36 | 0.84 | 3.03 | 49.04 | 76.99 | 0.04 | 0.04 | 1.00 |
| | 3 | 115.53 | 3.09 | 0.82 | 3.21 | 32.09 | 139.01 | 0.07 | 0.03 | 3.00 |
| | 4 | 25.09 | 1.00 | 0.83 | 3.79 | 26.39 | 79.01 | 0.05 | 0.02 | 1.00 |
| | 5 | 57.11 | 1.92 | 1.32 | 3.42 | 21.26 | 190.99 | 0.04 | 0.01 | 1.00 |
| Maximum | 1 | 171.48 | 22.92 | 1.97 | 3.24 | 57.14 | 357.00 | 4.58 | 0.74 | 3.00 |
| | 2 | 194.87 | 19.84 | 1.81 | 4.65 | 86.96 | 256.99 | 1.24 | 1.06 | 3.00 |
| | 3 | 610.70 | 34.79 | 1.45 | 4.72 | 79.91 | 493.99 | 6.89 | 2.45 | 4.00 |
| | 4 | 174.43 | 21.02 | 2.12 | 5.25 | 55.67 | 241.01 | 3.50 | 0.91 | 3.00 |
| | 5 | 201.00 | 27.18 | 3.15 | 5.24 | 81.56 | 465.00 | 6.76 | 0.31 | 3.00 |
| Minimum | 1 | 3.22 | 0.13 | 0.33 | 1.05 | 4.17 | 25.01 | 0.00 | 0.00 | 1.00 |
| | 2 | 1.11 | 0.07 | 0.32 | 1.09 | 32.65 | 13.99 | 0.00 | 0.00 | 1.00 |
| | 3 | 7.79 | 0.14 | 0.32 | 1.07 | 4.47 | 19.99 | 0.00 | 0.00 | 1.00 |
| | 4 | 1.17 | 0.04 | 0.29 | 2.88 | 5.56 | 16.99 | 0.00 | 0.00 | 1.00 |
| | 5 | 1.54 | 0.07 | 0.65 | 1.57 | 1.61 | 25.01 | 0.00 | 0.00 | 1.00 |

**Table S5. Flood event number and their percentages of individual classes in all the selected catchments**

| Basins | Stations | Abbreviations | Flood event number of class | | | | | | Percentage(%) | | | | |
|---|---|---|---|---|---|---|---|---|---|---|---|---|---|
| | | | 1 | 2 | 3 | 4 | 5 | Total | 1 | 2 | 3 | 4 | 5 |
| Songliao | Dongfeng | DF | 0 | 3 | 1 | 9 | 1 | 14 | 0.0 | 21.4 | 7.1 | 64.3 | 7.1 |
| | Jingyu | JY | 0 | 3 | 1 | 9 | 0 | 13 | 0.0 | 23.1 | 7.7 | 69.2 | 0.0 |
| | Muling | ML | 0 | 0 | 2 | 7 | 3 | 12 | 0.0 | 0.0 | 16.7 | 58.3 | 25.0 |
| | Yitong | YT | 0 | 6 | 0 | 7 | 1 | 14 | 0.0 | 42.9 | 0.0 | 50.0 | 7.1 |
| | Total | | 0 | 12 | 4 | 32 | 5 | 53 | 0.0 | 22.6 | 7.5 | 60.4 | 9.4 |
| Yellow | Huating | HT | 0 | 2 | 0 | 7 | 2 | 11 | 0.0 | 18.2 | 0.0 | 63.6 | 18.2 |
| | Luanchuan | LC | 4 | 6 | 2 | 27 | 0 | 39 | 10.3 | 15.4 | 5.1 | 69.2 | 0.0 |
| | Qiaotou | QT | 0 | 4 | 1 | 17 | 0 | 22 | 0.0 | 18.2 | 4.5 | 77.3 | 0.0 |
| | Tantou | TT | 7 | 2 | 2 | 16 | 5 | 32 | 21.9 | 6.3 | 6.3 | 50.0 | 15.6 |
| | Total | | 11 | 14 | 5 | 67 | 7 | 104 | 10.6 | 13.5 | 4.8 | 64.4 | 6.7 |
| Huaihe | Beimiaoji | BM | 0 | 0 | 0 | 0 | 12 | 12 | 0.0 | 0.0 | 0.0 | 0.0 | 100.0 |
| | Dapoling | DP | 0 | 6 | 1 | 5 | 9 | 21 | 0.0 | 28.6 | 4.8 | 23.8 | 42.9 |
| | Huangnizhuang | HN | 1 | 0 | 1 | 4 | 4 | 10 | 10.0 | 0.0 | 10.0 | 40.0 | 40.0 |
| | Lixin | LX | 0 | 5 | 5 | 4 | 4 | 18 | 0.0 | 27.8 | 27.8 | 22.2 | 22.2 |
| | Luzhuang | LZ | 1 | 0 | 0 | 4 | 6 | 11 | 9.1 | 0.0 | 0.0 | 36.4 | 54.5 |
| | Peihe | PH | 5 | 0 | 1 | 5 | 7 | 18 | 27.8 | 0.0 | 5.6 | 27.8 | 38.9 |
| | Qilin | QL | 2 | 0 | 0 | 1 | 7 | 10 | 20.0 | 0.0 | 0.0 | 10.0 | 70.0 |
| | Xiagushan | XG | 3 | 3 | 1 | 3 | 9 | 19 | 15.8 | 15.8 | 5.3 | 15.8 | 47.4 |
| | Xinxian | XX | 3 | 3 | 2 | 2 | 14 | 24 | 12.5 | 12.5 | 8.3 | 8.3 | 58.3 |
| | Yangzhuang | YZ | 0 | 5 | 1 | 2 | 2 | 10 | 0.0 | 50.0 | 10.0 | 20.0 | 20.0 |
| | Zhongtang | ZT | 2 | 3 | 1 | 4 | 5 | 15 | 13.3 | 20.0 | 6.7 | 26.7 | 33.3 |
| | Zhuganpu | ZG | 4 | 2 | 1 | 2 | 17 | 26 | 15.4 | 7.7 | 3.8 | 7.7 | 65.4 |
| | Ziluoshan | ZL | 3 | 2 | 2 | 8 | 6 | 21 | 14.3 | 9.5 | 9.5 | 38.1 | 28.6 |
| | Total | | 24 | 29 | 16 | 44 | 102 | 215 | 11.2 | 13.5 | 7.4 | 20.5 | 47.4 |
| Yangtze | Anhe | AH | 5 | 3 | 2 | 3 | 1 | 14 | 35.7 | 21.4 | 14.3 | 21.4 | 7.1 |
| | Anren | AR | 8 | 14 | 3 | 3 | 5 | 33 | 24.2 | 42.4 | 9.1 | 9.1 | 15.2 |
| | Baitugang | BT | 1 | 3 | 1 | 6 | 0 | 11 | 9.1 | 27.3 | 9.1 | 54.5 | 0.0 |
| | Biyang | BY | 1 | 1 | 0 | 10 | 0 | 12 | 8.3 | 8.3 | 0.0 | 83.3 | 0.0 |
| | Chengcun | CC | 11 | 3 | 9 | 0 | 0 | 23 | 47.8 | 13.0 | 39.1 | 0.0 | 0.0 |
| | Dutou | DT | 6 | 8 | 1 | 8 | 0 | 23 | 26.1 | 34.8 | 4.3 | 34.8 | 0.0 |
| | Gaotan | GT | 4 | 5 | 4 | 6 | 4 | 23 | 17.4 | 21.7 | 17.4 | 26.1 | 17.4 |
| | Jiahe | JH | 6 | 6 | 1 | 0 | 0 | 13 | 46.2 | 46.2 | 7.7 | 0.0 | 0.0 |
| | Jiajiafang | JJ | 2 | 4 | 0 | 4 | 1 | 11 | 18.2 | 36.4 | 0.0 | 36.4 | 9.1 |
| | Jinping | JP | 3 | 2 | 6 | 2 | 4 | 17 | 17.6 | 11.8 | 35.3 | 11.8 | 23.5 |
| | Jitan | JT | 0 | 2 | 2 | 3 | 4 | 11 | 0.0 | 18.2 | 18.2 | 27.3 | 36.4 |
| | Juwan | JW | 4 | 3 | 0 | 8 | 1 | 16 | 25.0 | 18.8 | 0.0 | 50.0 | 6.3 |
| | Liangshuikou | LK | 24 | 6 | 6 | 26 | 3 | 65 | 36.9 | 9.2 | 9.2 | 40.0 | 4.6 |
| | Liqingdian | LQ | 0 | 6 | 2 | 14 | 7 | 29 | 0.0 | 20.7 | 6.9 | 48.3 | 24.1 |
| | Loudi | LD | 7 | 5 | 6 | 2 | 5 | 25 | 28.0 | 20.0 | 24.0 | 8.0 | 20.0 |
| | Miping | MP | 3 | 3 | 5 | 3 | 5 | 19 | 15.8 | 15.8 | 26.3 | 15.8 | 26.3 |
| | Pingshi | PS | 5 | 3 | 1 | 8 | 5 | 22 | 22.7 | 13.6 | 4.5 | 36.4 | 22.7 |
| | Shahebu | SH | 3 | 3 | 2 | 2 | 0 | 10 | 30.0 | 30.0 | 20.0 | 20.0 | 0.0 |
| | Shanggao | SG | 10 | 2 | 2 | 3 | 2 | 19 | 52.6 | 10.5 | 10.5 | 15.8 | 10.5 |
| | Shijie | SJ | 3 | 4 | 0 | 4 | 2 | 13 | 23.1 | 30.8 | 0.0 | 30.8 | 15.4 |
| | Shimenkan | SM | 16 | 25 | 2 | 5 | 2 | 50 | 32.0 | 50.0 | 4.0 | 10.0 | 4.0 |
| | Shuangfeng | SF | 9 | 8 | 7 | 8 | 1 | 33 | 27.3 | 24.2 | 21.2 | 24.2 | 3.0 |
| | Shuangjiangkou | SK | 8 | 3 | 12 | 1 | 0 | 24 | 33.3 | 12.5 | 50.0 | 4.2 | 0.0 |
| | Sifen | SI | 4 | 2 | 2 | 0 | 2 | 10 | 40.0 | 20.0 | 20.0 | 0.0 | 20.0 |
| | Tangdukou | TD | 10 | 19 | 1 | 2 | 1 | 33 | 30.3 | 57.6 | 3.0 | 6.1 | 3.0 |
| | Tanghe | TH | 0 | 3 | 1 | 5 | 9 | 18 | 0.0 | 16.7 | 5.6 | 27.8 | 50.0 |
| | Tonggu | TG | 5 | 2 | 0 | 0 | 10 | 17 | 29.4 | 11.8 | 0.0 | 0.0 | 58.8 |
| | Tongtang | TO | 14 | 6 | 5 | 2 | 1 | 28 | 50.0 | 21.4 | 17.9 | 7.1 | 3.6 |
| | Wuxigou | WX | 4 | 5 | 0 | 7 | 1 | 17 | 23.5 | 29.4 | 0.0 | 41.2 | 5.9 |
| | Xiawan | XW | 6 | 0 | 0 | 2 | 3 | 11 | 54.5 | 0.0 | 0.0 | 18.2 | 27.3 |
| | Xixia | XI | 1 | 1 | 3 | 5 | 6 | 16 | 6.3 | 6.3 | 18.8 | 31.3 | 37.5 |
| | Xupu | XP | 12 | 14 | 4 | 5 | 1 | 36 | 33.3 | 38.9 | 11.1 | 13.9 | 2.8 |
| | Yanling | YL | 18 | 4 | 4 | 7 | 0 | 33 | 54.5 | 12.1 | 12.1 | 21.2 | 0.0 |
| | Yanta | YA | 6 | 2 | 1 | 4 | 0 | 13 | 46.2 | 15.4 | 7.7 | 30.8 | 0.0 |
| | Yuanken | YK | 2 | 3 | 1 | 0 | 7 | 13 | 15.4 | 23.1 | 7.7 | 0.0 | 53.8 |
| | Yucun | YC | 12 | 0 | 18 | 3 | 1 | 34 | 35.3 | 0.0 | 52.9 | 8.8 | 2.9 |
| | Yuexi | YX | 14 | 4 | 11 | 5 | 3 | 37 | 37.8 | 10.8 | 29.7 | 13.5 | 8.1 |
| | Zhangdou | ZD | 4 | 3 | 0 | 5 | 0 | 12 | 33.3 | 25.0 | 0.0 | 41.7 | 0.0 |
| | Total | | 251 | 190 | 125 | 181 | 97 | 844 | 29.7 | 22.5 | 14.8 | 21.4 | 11.5 |
| Southeast | Anxi | AX | 1 | 3 | 4 | 6 | 0 | 14 | 7.1 | 21.4 | 28.6 | 42.9 | 0.0 |

| | | | | | | | | | | | | |
|---|---|---|---|---|---|---|---|---|---|---|---|---|
| | Longshan | LS | 1 | 3 | 16 | 3 | 0 | 23 | 4.3 | 13.0 | 69.6 | 13.0 | 0.0 |
| | Tunxi | TX | 5 | 3 | 1 | 1 | 3 | 13 | 38.5 | 23.1 | 7.7 | 7.7 | 23.1 |
| | Xufan | XF | 1 | 3 | 5 | 1 | 0 | 10 | 10.0 | 30.0 | 50.0 | 10.0 | 0.0 |
| | Zhaoan | ZA | 1 | 5 | 12 | 8 | 4 | 30 | 3.3 | 16.7 | 40.0 | 26.7 | 13.3 |
| | Total | | 9 | 17 | 38 | 19 | 7 | 90 | 10.0 | 18.9 | 42.2 | 21.1 | 7.8 |
| | Hezikou | HZ | 42 | 17 | 7 | 22 | 1 | 89 | 47.2 | 19.1 | 7.9 | 24.7 | 1.1 |
| Pearl | Huishui | HS | 3 | 3 | 0 | 4 | 0 | 10 | 30.0 | 30.0 | 0.0 | 40.0 | 0.0 |
| | Libo | LB | 5 | 0 | 0 | 6 | 0 | 11 | 45.5 | 0.0 | 0.0 | 54.5 | 0.0 |
| | Xiaogulu | XL | 2 | 24 | 0 | 0 | 4 | 30 | 6.7 | 80.0 | 0.0 | 0.0 | 13.3 |
| | Total | | 52 | 44 | 7 | 32 | 5 | 140 | 37.1 | 31.4 | 5.0 | 22.9 | 3.6 |
| | Total | | 347 | 306 | 195 | 375 | 223 | 1446 | 24.0 | 21.2 | 13.5 | 25.9 | 15.4 |

"

More specifically, in the results section, the comprehensive introductions of flood response characteristics of different classes (Section 4.2), and control mechanisms of meteorological and physio-geographical factors (Section 4.4) were given to avoid the repeated present the results in the tables and figures. Additionally, the discussions were strengthened in the discussion section, particularly for the comparison of our flood event classification with the existing studies.

The figures were redrawn following the comments of you and the second reviewer, including Figures 1, 4, 5, 6, 8 and 10.

Minor comments:

Line 40. You refer many times in the text to behavior characteristics what I consider response types. When we talk about behavior, you are trying to characterize the catchment dynamic which is intrinsic to each catchment. In other words, you try to characterize the low filter function that transform input to outputs. I would suggest changing the word behavior for response which is a more precise word for what you are analyzing.

**Response:** We replaced the word "behavior" with "response" in the whole manuscript.

Line 77. The expression "solid data foundation" is a biased description of your research.

**Response:** It was revised as "*provides the mechanism supports for predicting flood event classes*".

Line 94. This is not the right way to refer to information extracted from a webpage. Check the referring rules from the journal.

**Response:** The websites were removed from the manuscript because the detailed data sources were given in the section of Code/Data availability.

Line 109. How dense is the meteorological gauge network? How can we be sure that they are representative of the basin analyzed?

**Response:** The meteorological stations in the buffer zone with a radius of 100 km of individual catchment centers were selected. All the selected meteorological stations were added in Figure 1. The total number of meteorological stations was 466 and no less than eight stations were within or around individual catchments.

[Figure]

*Figure 1. Spatial distributions of all the selected flood events and their corresponding climate types*

Additionally, the relationships between flood events and meteorological factors were well captured by the catchment hydrological model (Zhang *et al*., 2024), which could well demonstrate the representatives of all the control factors.

Some revisions were given as follow.

"*All the meteorological stations in the buffer zone with a radius of 100 km of every catchment centers were selected. The station number was 466 in total and no less than eight stations for each catchment.*"

"*All these control factors well represented the meteorological and underlying surface conditions of individual catchments because all these flood events were captured satisfactorily by the catchment hydrological model developed using these factors (Zhang et al., 2024).*"

*References*

*Zhang, Y. Y., Zhang, Y. Q., Zhai, X., Xia, J., Tang, Q., Zhao, T., and Wang, W.: Predicting flood event class using a novel class membership function and hydrological modeling, Earth's Future, 12, e2023EF004081. https://doi.org/10.1029/2023EF004081,2024.*

Figure 1. The gauge distribution is strongly biased to Yangtze and Huai Rivers. How can you develop an analysis by basin with this low density in the other basins?

**Response:** The selections of hydrological stations and flood events were mainly based on the basin area, flood prone area, data availability and quality (i.e., no regulations of human activities), and so on.

The flood events in headstream catchments were selected, which were mainly in the Huaihe River Basin in the south–north climate zone of China, and the Yangtze, Southeast and Pearl River Basins in the Southern China. The flood events were more likely to occur in all these basins than those in the Songliao and Yellow River Basin in

the Northern China. Thus, the densities of flood events and gauges in the Huaihe River Basin and Southern China were much greater than those in the Northern China, i.e., $0.09–0.48\times10^{-4}$ station/km$^2$ and $3.09–7.96\times10^{-4}$ events/km$^2$ in the Huaihe River Basin and Southern China, $0.03–0.05\times10^{-4}$ station/km$^2$ and $0.42–1.36\times10^{-4}$ events/km$^2$ in the Northern China. Additionally, although the station densities in the Huaihe and Yangtze River Basins were greater than those of Southeast and Pearl River Basins, the flood event densities were approximately close, all of which were around $3.09–7.96\times10^{-4}$ events/km$^2$ (see the Table S1 in the Supplement).

The explanations were added in the manuscript as follows: *"The densities of flood events and gauges in the Huaihe River Basin and Southern China were much greater than those in the Songliao and Yellow River Basins in the Northern China because of the higher occurrences of flood events (Table S1 in the Supplement) (China Institute of Water Resources and Hydropower Research and Research Center on Flood and Drought Disaster Prevention and Reduction, the Ministry of Water Resources, 2021)."*

*"Table S1. Total numbers and densities of hydrological stations and flood events in different river basins*

| Basin | Area ($10^4km^2$) | Number | | Density | |
|---|---|---|---|---|---|
| | | Station | Flood event | Station ($10^{-4}$ station/km$^2$) | Event ($10^{-4}$ event/km$^2$) |
| Songliao River Basin | 124.92 | 4 | 53 | 0.03 | 0.42 |
| Yellow River Basin | 75.24 | 4 | 104 | 0.05 | 1.38 |
| Huaihe River Basin | 27.00 | 13 | 215 | 0.48 | 7.96 |
| Yangtze River Basin | 180.85 | 38 | 844 | 0.21 | 4.67 |
| Southeast River Basin | 24.02 | 5 | 90 | 0.21 | 3.75 |
| Pearl River Basin | 45.36 | 4 | 140 | 0.09 | 3.09 |

*"*

Additionally, the representatives of flood event classes would be investigated if more events were selected in future works. It was revised in the discussion section as follows.

*"However, several works should be paid attention for further improvements of our study……."* and *"The representatives of individual classes should be further investigated particularly in the basins with low densities of flood events……."*

Line 139. PCA is known to work well for linear factors. Did you check for non-linear relationships?

**Response:** We tested the independence and linear correlations among different flood response metrics using the ANOVA test and correlations test. The results showed that $T_{bgn}$ is independent from $R$, $RQ_r$, $RQ_d$ and $N_{pk}$; $Q_{pk}$ is independent from $T_{pk}$; and $N_{pk}$ is independent from $RQ_r$ and $RQ_d$. Expect these independent metrics, all the other metrics have linear correlations with each other. Therefore, non-linear relationships do not exist among the flood response metrics and the PCA can be used for the dimensionality reduction analysis.

The revisions were given as follows:

*"...involving the* `aov`, `cor` *and* `princomp` *functions in stats Package (version 4.1.3) for independence test, linear correlation test..."*

*"By the tests of independence and linear correlation for all the flood response metrics, $T_{bgn}$ is independent from R, $RQ_r$, $RQ_d$ and $N_{pk}$; $Q_{pk}$ is independent from $T_{pk}$; and $N_{pk}$ is independent from $RQ_r$ and $RQ_d$. Expect these independent metrics, all the other metrics have linear correlations with each other (Table S3 in the Supplement)."*

**"Table S3. Results of independence and linear correlation tests among different flood response metrics**

| Methods | | Correlation coefficient for the correlations test | | | | | | | | |
|---|---|---|---|---|---|---|---|---|---|---|
| | | R | $Q_{pk}$ | CV | $T_{bgn}$ | $T_{pk}$ | $T_{drn}$ | $RQ_r$ | $RQ_d$ | $N_{pk}$ |
| p-value for ANOVA test | R | | **0.68** | **0.14** | *0.00* | **0.06** | **0.14** | **0.26** | **0.34** | **0.34** |
| | $Q_{pk}$ | **0.00** | | **0.41** | *0.02* | *-0.03* | **-0.18** | **0.75** | **0.77** | **0.08** |
| | CV | **0.00** | **0.00** | | **0.06** | **-0.24** | **0.18** | **0.38** | **0.19** | **-0.21** |
| | $T_{bgn}$ | *0.93* | *0.45* | **0.02** | | **-0.12** | **0.07** | *0.04* | *0.04* | *-0.04* |
| | $T_{pk}$ | **0.02** | *0.19* | **0.00** | **0.00** | | **-0.14** | **-0.19** | **0.11** | **0.14** |
| | $T_{drn}$ | **0.00** | **0.00** | **0.00** | **0.01** | **0.00** | | **-0.19** | **-0.28** | **0.23** |
| | $RQ_r$ | **0.00** | **0.00** | **0.00** | *0.12* | **0.00** | **0.00** | | **0.68** | *-0.03* |
| | $RQ_d$ | **0.00** | **0.00** | **0.00** | *0.15* | **0.00** | **0.00** | **0.00** | | *0.02* |
| | $N_{pk}$ | **0.00** | **0.00** | **0.00** | *0.17* | **0.00** | **0.00** | *0.31* | *0.38* | |

Note: the bold value indicates that the test passes the 95% significance test, and the italic value indicates that the test does not pass the 95% significance test."

Line 163-168. You are presenting the same information as Table 2. You should summarize.

**Response:** These sentences were summarized as follows: *"In the meteorological category, 17 factors related to precipitation, potential evapotranspiration and aridity index are selected, including the amounts, intensities and timing factors during flood events, in the antecedent period and at annual scale."*

Line 173-178. You are presenting the same information as Table 2. You should summarize.

**Response:** These sentences were summarized as follows: *"For the physio-geographical factors, the 10 catchment attributes are selected, including catchment location, area, elevation and slope, river density and slope."* and *"Seven land cover factors are selected, including the area fractions of paddy, dryland, forest, grassland, water, urban and rural area to the total catchment, respectively for the seven land cover periods."*

Table 2. Factors are hard to visualize. Add a bullet for each one.

**Response:** It was revised accordingly which were given as follows.

*"Table 2. Meteorological and physio-geographical factors in our study*

| Factor categories | Factors | | Data sources | Flood event effects |
|---|---|---|---|---|
| Meteorology | Precipitation | • pcp_ant: cumulative amount in the antecedent seven days (mm);
• pcp_dur: total amount during the flood event (mm);
• pcp_av: mean amount during the flood event (mm hr$^{-1}$);
• pcp_max: maximum intensity during the flood event (mm hr$^{-1}$);
• pcp_max: maximum intensity during the flood event (mm hr$^{-1}$);
• pcp_Tbeg: precipitation timing;
• pcp_Tdur: precipitation duration (days);
• pcp_ann: annual mean amount (mm);
• pcp_year: amount in the year when the flood event happens (mm) | Hourly precipitation in hydrological yearbooks; daily precipitation at 466 meteorological stations | Flood yield process |
| | Potential evapotranspiration | • pet_ant: cumulative amount in the antecedent seven days (mm);
• pet_dur: total amount during the flood event (mm)
• pet_max: maximum intensity during the flood event (mm hr$^{-1}$)
• pet_ann: annual mean amount (mm);
• pet_year: amount in the year when the flood event happens (mm) | Daily maximum and minimum temperature at 466 meteorological stations | Flood yield process |
| | Aridity index | • SPEI_ant: mean value in the antecedent seven days;
• SPEI_dur: mean value during the flood event ;
• SPEI_ann: annual mean value; | Daily maximum and minimum temperature at 466 | Flood yield process |

| | | | | |
|---|---|---|---|---|
| | | • SPEI_year: mean value in the year when the flood event happens | meteorological stations | |
| | Locations | • Longitude: longitude of catchment center
• Latitude: latitude of catchment center | Global positioning system | Meteorological conditions |
| Physio-geography | Catchment attributes | • Slope: catchment slope (%);
• Area: catchment area (km²);
• Length: catchment slope length (km);
• Elevation: average elevation of catchment (m);
• MaxiElev: maximum elevation of catchment (m); | Digital elevation model (size: 30 m×30 m) | Flood yield and overland routing processes |
| | River attributes | • Rivden: river density (km/km²);
• RivSlope: river slope (%);
• Rwd: ratio of river width to depth (m/m); | Digital elevation model (size: 30 m×30 m) | Flood routing processes in river system |
| | Land covers | • Rpaddy: area fraction of paddy to catchment (%);
• Rdryland: area fraction of dryland to catchment (%);
• Rforest: area fraction of forest to catchment (%);
• Rgrass: area fraction of grass to catchment (%);
• Rwater: area fraction of water to catchment (%);
• Rurban: area fraction of urban to catchment (%);
• Rrural: area fraction of unused land to catchment (%) | Land covers in 1990, 1995, 2000, 2005, 2010 and 2015 (size: 30 m×30 m) | Flood yield and overland routing processes |

"

Line 193-196. These lines should be at the beginning of the paragraph with a more detailed explanation of the method used.

**Response:** This paragraph was revised following your comments and a more detailed explanation of the constrained rank analysis was added.

*"The constrained rank analysis is adopted to quantify the direct or combined effects of control factor categories on spatial and temporal variabilities of individual flood event classes for both the distributed and lumped analyses. The widely adopted methods of constrained rank analysis are the Redundancy Analysis (RDA) and the Canonical Correlation Analysis (CCA). The RAD is a linear model and the CCA is a unimodal model, both of which are the extended methods of principal component analysis combined with regression analysis. These methods have strong advantages to solve multiple linear regressions and interactions between dependent and independent variable matrixes which are transformed into a few independent composite factors (ter Braak, 1986; Legendre and Anderson, 1999), and are beneficial to quantify the effects of independent variable matrix on dependent variable matrix and to find the most important factors, which have been commonly used in testing the multispecies response to environmental variables in the biological or ecological sciences (Legendre and Anderson, 1999), effects of physio-geographical factors and human activities on diffuse nutrient losses or water quality (Zhang et al., 2016; Shi et al., 2017), and so on. The constrained proportion is the percentage of explained variance by independent variable matrix to the total variance of dependent variable matrix, which is usually*

*considered as the effect contribution of individual meteorological and physio-geographical factors or categories on total variabilities of flood event classes. If the contribution sum of individual factor effects is less than the entire contribution of all the factors, the interactive effects are among the factors and the difference between the summed and entire contributions is the combined contribution (Zhang et al., 2016). The selection is based on the first axis length. The CCA is proposed when the first axis length is greater than 4.0, while the RDA is proposed when the first axis length is less than 3.0. Otherwise, both CCA and RDA are proposed (ter Braak, 1986; Zhang et al., 2020)."*


Line 208. You should be more specific about how you got that. What are the values inside the table? Explain more.

**Response:** The method and results of principal component analysis were introduced specifically.

In the method section: *"The main flood response metrics in the individual PCAs were determined according to the load coefficient matrix. If the load coefficient is over 0.45, the corresponding flood response metric are considered to be highly correlated with the PCA."*

In the result section: *"By the principal component analysis, five independent PCAs are found with the total cumulative variance of 85.7%, which are greater than the threshold (80.0%) (Table 3). Thus, the first five PCAs are selected in our study. According to the load coefficient matrix, the first PCA is related with magnitude (R and $Q_{pk}$), variability (CV) and rates of changes ($RQ_r$ and $RQ_d$) with the load coefficients of 0.61, 0.97, 0.47, 0.84 and 0.84, respectively, and all of these metrics explain 33.3% of total variances of flood response metrics. The second PCA is related with R, CV, $T_{pk}$ and $N_{pk}$ with the load coefficients of 0.51, -0.47 and 0.56, respectively, and all of these metrics explain 17.0% of total variances. The third PCA is mainly related with $T_{drn}$ and $T_{pk}$ with the load coefficients of -0.48 and 0.48, respectively, and all of these metrics explain 16.0% of total variances. The fourth and fifth PCAs are mainly related with timings ($T_{bgn}$ and $T_{pk}$) of flood event and maximum flood peak with the load coefficients of 0.92 and 0.64, respectively. The explained variances are 10.8% and 8.6%, respectively."*

Line 209. Typo. What is the value 33.2 or 33.3%?

**Response:** The value is 33.3%, and it was corrected accordingly.

Line 210. What clustering methods are you referring here?

**Response:** The clustering methods are the the hierarchical and *k*-medoids methods. It was revised as follows: *"Compared with the classification performance of these two clustering methods (i.e., the hierarchical and k-medoids methods) among individual optimal cluster numbers……"*

Line 226-254. You are just describing the data that could be summarized on an appendix table.

**Response:** The revised sentences were given as follows: *"The value ranges of flood response metrics in different classes are presented in Figure 3 and Table S4 in the Supplement. For the magnitude*

*metrics, both total flood volume (R) and maximum flood peak ($Q_{pk}$) variations are the same among different classes. The metric values in Class 3 are the largest, followed by Classes 5, 2, 1 and 4. For the variability metrics (CV), the events are the most variable in Class 5, and are slightly variable in the other Classes with the mean CV being less than 1.0, i.e., 0.90±0.26 (Class 1), 0.87±0.25 (Class 2), 0.86±0.26 (Class 4) and 0.84±0.22 (Class 3). For the timing and duration metrics (i.e., $T_{bgn}$, $T_{drn}$ and $T_{pk}$), 73.2% of flood events in Class 1 occur before the wet season (i.e., January - May), and 58.5%, 67.7% and 57.0% of flood events in Classes 2, 3 and 5 occur in the earlier wet season (i.e., June - July), and 52.8% of flood events in Class 4 occur in the latter wet season (i.e., August - September). The mean duration ($T_{drn}$) is the longest in Class 5, followed by Classes 3 and 1. The mean $T_{drn}$ values in Classes 4 and 2 are the shortest, i.e., 85.73±39.97 h and 83.82±41.20 h. The timings of maximum flood peaks ($T_{pk}$) are usually the largest in Class 2 with the mean of 50.6%±10.3%, which means that the flood peaks mainly occur in the middle or late stages of flood events. The flood peaks usually occur in the early stage of flood events in the other classes (i.e., Classes 1, 2, 4 and 5). Particularly in Class 3, the mean $T_{pk}$ value is only 23.7%±13.6%.*

*For the rates of changes, $RQ_r$ in most classes are much greater than $RQ_d$ because the flood peaks usually occur in the early stage of flood events, except Class 2. The largest values of both $RQ_r$ and $RQ_d$ are in Class 3 because of the greatest flood peak. The smallest $RQ_r$ values are mainly in Classes 2 because of the late occurrences of flood peaks, while the smallest $RQ_d$ values are mainly in Class 5 because of the long durations of flood recession. For the flood peak number ($N_{pk}$), 71.2%, 69.9%, 76.5% and 77.1% of flood events has one flood peaks in Classes 1, 2, 4 and 5, respectively, and multiple flood peaks (i.e., two–four) exist in 94.4% of total flood events in Class 3, accounting for 33.8% (two peaks), 48.7% (three peaks) and 11.8% (four peaks), respectively. "*

The characteristic values of flood response metrics in different classes were provided in Table S4 of Supplement.

| Characteristic value | Class | $R(mm \cdot day^{-1})$ | $Q_{pk}(mm \cdot day^{-1})$ | CV | $T_{bgn}$ | $T_{pk}(\%)$ | $T_{drn}(h)$ | $RQ_r(h^{-1})$ | $RQ_d(h^{-1})$ | $N_{pk}$ |
|---|---|---|---|---|---|---|---|---|---|---|
| Average± Standard Deviation | 1 | 43.97±29.94 | 2.04±2.51 | 0.90±0.26 | 2.28±0.49 | 27.14±9.60 | 103.92±43.39 | 0.13±0.32 | 0.04±0.07 | 1.31±0.51 |
| | 2 | 45.81±34.01 | 2.21±2.52 | 0.87±0.25 | 3.06±0.69 | 50.64±10.28 | 83.82±41.20 | 0.08±0.14 | 0.08±0.12 | 1.32±0.50 |
| | 3 | 143.97±108.33 | 5.23±6.04 | 0.84±0.22 | 3.24±0.61 | 33.90±15.02 | 145.26±68.99 | 0.25±0.62 | 0.12±0.28 | 2.67±0.76 |
| | 4 | 33.31±26.64 | 1.69±2.11 | 0.86±0.26 | 3.85±0.51 | 26.11±9.09 | 85.73±39.97 | 0.14±0.30 | 0.04±0.08 | 1.24±0.43 |
| | 5 | 65.79±43.80 | 2.98±3.68 | 1.40±0.43 | 3.43±0.61 | 23.74±13.60 | 202.88±85.42 | 0.18±0.62 | 0.03±0.04 | 1.24±0.46 |
| Median | 1 | 35.63 | 1.17 | 0.89 | 2.30 | 27.27 | 97.01 | 0.05 | 0.02 | 1.00 |
| | 2 | 37.84 | 1.36 | 0.84 | 3.03 | 49.04 | 76.99 | 0.04 | 0.04 | 1.00 |
| | 3 | 115.53 | 3.09 | 0.82 | 3.21 | 32.09 | 139.01 | 0.07 | 0.03 | 3.00 |
| | 4 | 25.09 | 1.00 | 0.83 | 3.79 | 26.39 | 79.01 | 0.05 | 0.02 | 1.00 |
| | 5 | 57.11 | 1.92 | 1.32 | 3.42 | 21.26 | 190.99 | 0.04 | 0.01 | 1.00 |
| Maximum | 1 | 171.48 | 22.92 | 1.97 | 3.24 | 57.14 | 357.00 | 4.58 | 0.74 | 3.00 |
| | 2 | 194.87 | 19.84 | 1.81 | 4.65 | 86.96 | 256.99 | 1.24 | 1.06 | 3.00 |
| | 3 | 610.70 | 34.79 | 1.45 | 4.72 | 79.91 | 493.99 | 6.89 | 2.45 | 4.00 |
| | 4 | 174.43 | 21.02 | 2.12 | 5.25 | 55.67 | 241.01 | 3.50 | 0.91 | 3.00 |
| | 5 | 201.00 | 27.18 | 3.15 | 5.24 | 81.56 | 465.00 | 6.76 | 0.31 | 3.00 |
| Minimum | 1 | 3.22 | 0.13 | 0.33 | 1.05 | 4.17 | 25.01 | 0.00 | 0.00 | 1.00 |
| | 2 | 1.11 | 0.07 | 0.32 | 1.09 | 32.65 | 13.99 | 0.00 | 0.00 | 1.00 |
| | 3 | 7.79 | 0.14 | 0.32 | 1.07 | 4.47 | 19.99 | 0.00 | 0.00 | 1.00 |
| | 4 | 1.17 | 0.04 | 0.29 | 2.88 | 5.56 | 16.99 | 0.00 | 0.00 | 1.00 |
| | 5 | 1.54 | 0.07 | 0.65 | 1.57 | 1.61 | 25.01 | 0.00 | 0.00 | 1.00 |

"

Figure 4. I would try 2 columns. Left: Flood event distribution. Right: Frequency histogram. Currently, it is too small to watch some differences in the distributions.

**Response:** This figure was redrawn following your comments.

[Figure]

*Figure 4. Flood event distributions in the 95% confidence interval and their median, and their duration frequencies of Classes 1–5 (a–e)*

Line 268-283. You should add a discussion about your results. You are mainly describing information that could be in an appendix table.

**Response:** We reorganized this paragraph to clearly introduce the main spatial distributions of individual classes.

*"The spatial distributions of individual classes are showed in Figure 5 and Table S5 in the Supplement. The moderately fast flood event class (i.e., Class 1) is mainly in the Pearl and Yangtze River Basins, accounting for 37.1% (52/140) and 29.7% (251/844) of total events, respectively. Specifically, Class 1 is dominant in the Xiawan, Yanling and Songgao catchments in the Yangtze River Basin, and Hezikou catchment in the Pearl River Basin. The highly fast flood event class (i.e., Class 2) is mainly in the Pearl River Basin, accounting for 31.4% (44/140) of total events, particularly in the Xiaogulu catchment. The highly slow and multipeak flood event class (i.e., Class 3) is mainly in the Southeast River Basin, accounting for 42.2% (38/90) of total events, particularly in the Longshan catchment. The slightly fast flood event class (i.e., Class 4) is mainly in the Yellow and Songliao River Basins, accounting for 64.4% (67/104) and 60.4% (32/53) of total events, respectively. The most obvious catchments are Biyang in the Yangtze River Basin, Qiaotou and Luanchuan in the Yellow River Basin, Jingyu and Dongfeng in the Songliao River Basin. The moderately slow flood event class (i.e., Class 5) is mainly in the Huaihe River Basin, accounting for 47.4% (102/215) of total events, particularly in the Beimiaoji and Qilin catchments. Therefore, the Classes 1 to 3 are mainly in the Temperate without Dry Season climate region in southern China (Figure 1), the Class 4 is mainly in the Cold with Dry Winter climate region in northern China, and the Class 5 is mainly in the transition region between Temperate without Dry Season climate and Cold with Dry Winter climate."*

More discussions about the reasons of the spatial differences of individual classes were also provided in the discussion section.

*"Classes 1 and 2 are mainly in the southern China, particularly in the Pearl and Yangtze River Basins, which are controlled by the temperate climate without a dry season. Storms with high intensities and short durations before the wet season in the southern China are likely to cause flood events with great magnitudes and variabilities (Class 1) or fast flood events with a high single peak and short durations (Class 2) (Gao et al., 2018)".*

*"Class 3 is mainly in the Southeast River Basin controlled by the tropical cyclone climate. Severe storms with high intensities and durations are likely to cause high slow flood events with multiple peaks (Class 3) (Yin et al., 2010; Zhang et al., 2020)."*

*"Class 4 is mainly in the northern China controlled by the cold climate with dry winters. The heavy storms ahead of westerlies trough mainly occur in the latter wet season in this region, which usually have low intensities and short durations (Gao et al., 2018). Thus they are likely to cause the small fast flood events (Class 4),……".*

*"Class 5 is mainly in the south–north climate zone of China (i.e., Huaihe River Basin), which has the dual climate characteristics of both south and north monsoons. Storms characterized by a long period of continuous rainy meteorological with high frequency and low intensities (e.g., Meiyu rainfalls) in the earlier wet season are likely to cause moderate slow flood events with long durations (Gao et al., 2018; Sampe and Xie, 2010)."*

We also added a table in the supplement (Table S5) to show the class distributions and their percentages of all the selected catchments.

*"Table S5. Flood event number and their percentages of individual classes in all the selected catchments*

| Basins | Stations | Abbreviations | Flood event number of class | | | | | | Percentage(%) | | | | |
|---|---|---|---|---|---|---|---|---|---|---|---|---|---|
| | | | 1 | 2 | 3 | 4 | 5 | Total | 1 | 2 | 3 | 4 | 5 |
| Songliao | Dongfeng | DF | 0 | 3 | 1 | 9 | 1 | 14 | 0.0 | 21.4 | 7.1 | 64.3 | 7.1 |
| | Jingyu | JY | 0 | 3 | 1 | 9 | 0 | 13 | 0.0 | 23.1 | 7.7 | 69.2 | 0.0 |
| | Muling | ML | 0 | 0 | 2 | 7 | 3 | 12 | 0.0 | 0.0 | 16.7 | 58.3 | 25.0 |
| | Yitong | YT | 0 | 6 | 0 | 7 | 1 | 14 | 0.0 | 42.9 | 0.0 | 50.0 | 7.1 |
| | Total | | 0 | 12 | 4 | 32 | 5 | 53 | 0.0 | 22.6 | 7.5 | 60.4 | 9.4 |
| Yellow | Huating | HT | 0 | 2 | 0 | 7 | 2 | 11 | 0.0 | 18.2 | 0.0 | 63.6 | 18.2 |
| | Luanchuan | LC | 4 | 6 | 2 | 27 | 0 | 39 | 10.3 | 15.4 | 5.1 | 69.2 | 0.0 |
| | Qiaotou | QT | 0 | 4 | 1 | 17 | 0 | 22 | 0.0 | 18.2 | 4.5 | 77.3 | 0.0 |
| | Tantou | TT | 7 | 2 | 2 | 16 | 5 | 32 | 21.9 | 6.3 | 6.3 | 50.0 | 15.6 |
| | Total | | 11 | 14 | 5 | 67 | 7 | 104 | 10.6 | 13.5 | 4.8 | 64.4 | 6.7 |
| Huaihe | Beimiaoji | BM | 0 | 0 | 0 | 0 | 12 | 12 | 0.0 | 0.0 | 0.0 | 0.0 | 100.0 |
| | Dapoling | DP | 0 | 6 | 1 | 5 | 9 | 21 | 0.0 | 28.6 | 4.8 | 23.8 | 42.9 |
| | Huangnizhuang | HN | 1 | 0 | 1 | 4 | 4 | 10 | 10.0 | 0.0 | 10.0 | 40.0 | 40.0 |
| | Lixin | LX | 0 | 5 | 5 | 4 | 4 | 18 | 0.0 | 27.8 | 27.8 | 22.2 | 22.2 |
| | Luzhuang | LZ | 1 | 0 | 0 | 4 | 6 | 11 | 9.1 | 0.0 | 0.0 | 36.4 | 54.5 |
| | Peihe | PH | 5 | 0 | 1 | 5 | 7 | 18 | 27.8 | 0.0 | 5.6 | 27.8 | 38.9 |
| | Qilin | QL | 2 | 0 | 0 | 1 | 7 | 10 | 20.0 | 0.0 | 0.0 | 10.0 | 70.0 |
| | Xiagushan | XG | 3 | 3 | 1 | 3 | 9 | 19 | 15.8 | 15.8 | 5.3 | 15.8 | 47.4 |
| | Xinxian | XX | 3 | 3 | 2 | 2 | 14 | 24 | 12.5 | 12.5 | 8.3 | 8.3 | 58.3 |
| | Yangzhuang | YZ | 0 | 5 | 1 | 2 | 2 | 10 | 0.0 | 50.0 | 10.0 | 20.0 | 20.0 |
| | Zhongtang | ZT | 2 | 3 | 1 | 4 | 5 | 15 | 13.3 | 20.0 | 6.7 | 26.7 | 33.3 |
| | Zhuganpu | ZG | 4 | 2 | 1 | 2 | 17 | 26 | 15.4 | 7.7 | 3.8 | 7.7 | 65.4 |

| Region | Station | Abbr | | | | | | Total | | | | | |
|---|---|---|---|---|---|---|---|---|---|---|---|---|---|
| | Ziluoshan | ZL | 3 | 2 | 2 | 8 | 6 | 21 | 14.3 | 9.5 | 9.5 | 38.1 | 28.6 |
| | Total | | 24 | 29 | 16 | 44 | 102 | 215 | 11.2 | 13.5 | 7.4 | 20.5 | 47.4 |
| Yangtze | Anhe | AH | 5 | 3 | 2 | 3 | 1 | 14 | 35.7 | 21.4 | 14.3 | 21.4 | 7.1 |
| | Anren | AR | 8 | 14 | 3 | 3 | 5 | 33 | 24.2 | 42.4 | 9.1 | 9.1 | 15.2 |
| | Baitugang | BT | 1 | 3 | 1 | 6 | 0 | 11 | 9.1 | 27.3 | 9.1 | 54.5 | 0.0 |
| | Biyang | BY | 1 | 1 | 0 | 10 | 0 | 12 | 8.3 | 8.3 | 0.0 | 83.3 | 0.0 |
| | Chengcun | CC | 11 | 3 | 9 | 0 | 0 | 23 | 47.8 | 13.0 | 39.1 | 0.0 | 0.0 |
| | Dutou | DT | 6 | 8 | 1 | 8 | 0 | 23 | 26.1 | 34.8 | 4.3 | 34.8 | 0.0 |
| | Gaotan | GT | 4 | 5 | 4 | 6 | 4 | 23 | 17.4 | 21.7 | 17.4 | 26.1 | 17.4 |
| | Jiahe | JH | 6 | 6 | 1 | 0 | 0 | 13 | 46.2 | 46.2 | 7.7 | 0.0 | 0.0 |
| | Jiajiafang | JJ | 2 | 4 | 0 | 4 | 1 | 11 | 18.2 | 36.4 | 0.0 | 36.4 | 9.1 |
| | Jinping | JP | 3 | 2 | 6 | 2 | 4 | 17 | 17.6 | 11.8 | 35.3 | 11.8 | 23.5 |
| | Jitan | JT | 0 | 2 | 2 | 3 | 4 | 11 | 0.0 | 18.2 | 18.2 | 27.3 | 36.4 |
| | Juwan | JW | 4 | 3 | 0 | 8 | 1 | 16 | 25.0 | 18.8 | 0.0 | 50.0 | 6.3 |
| | Liangshuikou | LK | 24 | 6 | 6 | 26 | 3 | 65 | 36.9 | 9.2 | 9.2 | 40.0 | 4.6 |
| | Liqingdian | LQ | 0 | 6 | 2 | 14 | 7 | 29 | 0.0 | 20.7 | 6.9 | 48.3 | 24.1 |
| | Loudi | LD | 7 | 5 | 6 | 2 | 5 | 25 | 28.0 | 20.0 | 24.0 | 8.0 | 20.0 |
| | Miping | MP | 3 | 3 | 5 | 3 | 5 | 19 | 15.8 | 15.8 | 26.3 | 15.8 | 26.3 |
| | Pingshi | PS | 5 | 3 | 1 | 8 | 5 | 22 | 22.7 | 13.6 | 4.5 | 36.4 | 22.7 |
| | Shahebu | SH | 3 | 3 | 2 | 2 | 0 | 10 | 30.0 | 30.0 | 20.0 | 20.0 | 0.0 |
| | Shanggao | SG | 10 | 2 | 2 | 3 | 2 | 19 | 52.6 | 10.5 | 10.5 | 15.8 | 10.5 |
| | Shijie | SJ | 3 | 4 | 0 | 4 | 2 | 13 | 23.1 | 30.8 | 0.0 | 30.8 | 15.4 |
| | Shimenkan | SM | 16 | 25 | 2 | 5 | 2 | 50 | 32.0 | 50.0 | 4.0 | 10.0 | 4.0 |
| | Shuangfeng | SF | 9 | 8 | 7 | 8 | 1 | 33 | 27.3 | 24.2 | 21.2 | 24.2 | 3.0 |
| | Shuangjiangkou | SK | 8 | 3 | 12 | 1 | 0 | 24 | 33.3 | 12.5 | 50.0 | 4.2 | 0.0 |
| | Sifen | SI | 4 | 2 | 2 | 0 | 2 | 10 | 40.0 | 20.0 | 20.0 | 0.0 | 20.0 |
| | Tangdukou | TD | 10 | 19 | 1 | 2 | 1 | 33 | 30.3 | 57.6 | 3.0 | 6.1 | 3.0 |
| | Tanghe | TH | 0 | 3 | 1 | 5 | 9 | 18 | 0.0 | 16.7 | 5.6 | 27.8 | 50.0 |
| | Tonggu | TG | 5 | 2 | 0 | 0 | 10 | 17 | 29.4 | 11.8 | 0.0 | 0.0 | 58.8 |
| | Tongtang | TO | 14 | 6 | 5 | 2 | 1 | 28 | 50.0 | 21.4 | 17.9 | 7.1 | 3.6 |
| | Wuxigou | WX | 4 | 5 | 0 | 7 | 1 | 17 | 23.5 | 29.4 | 0.0 | 41.2 | 5.9 |
| | Xiawan | XW | 6 | 0 | 0 | 2 | 3 | 11 | 54.5 | 0.0 | 0.0 | 18.2 | 27.3 |
| | Xixia | XI | 1 | 1 | 3 | 5 | 6 | 16 | 6.3 | 6.3 | 18.8 | 31.3 | 37.5 |
| | Xupu | XP | 12 | 14 | 4 | 5 | 1 | 36 | 33.3 | 38.9 | 11.1 | 13.9 | 2.8 |
| | Yanling | YL | 18 | 4 | 4 | 7 | 0 | 33 | 54.5 | 12.1 | 12.1 | 21.2 | 0.0 |
| | Yanta | YA | 6 | 2 | 1 | 4 | 0 | 13 | 46.2 | 15.4 | 7.7 | 30.8 | 0.0 |
| | Yuanken | YK | 2 | 3 | 1 | 0 | 7 | 13 | 15.4 | 23.1 | 7.7 | 0.0 | 53.8 |
| | Yucun | YC | 12 | 0 | 18 | 3 | 1 | 34 | 35.3 | 0.0 | 52.9 | 8.8 | 2.9 |
| | Yuexi | YX | 14 | 4 | 11 | 5 | 3 | 37 | 37.8 | 10.8 | 29.7 | 13.5 | 8.1 |
| | Zhangdou | ZD | 4 | 3 | 0 | 5 | 0 | 12 | 33.3 | 25.0 | 0.0 | 41.7 | 0.0 |
| | Total | | 251 | 190 | 125 | 181 | 97 | 844 | 29.7 | 22.5 | 14.8 | 21.4 | 11.5 |
| Southeast | Anxi | AX | 1 | 3 | 4 | 6 | 0 | 14 | 7.1 | 21.4 | 28.6 | 42.9 | 0.0 |
| | Longshan | LS | 1 | 3 | 16 | 3 | 0 | 23 | 4.3 | 13.0 | 69.6 | 13.0 | 0.0 |
| | Tunxi | TX | 5 | 3 | 1 | 1 | 3 | 13 | 38.5 | 23.1 | 7.7 | 7.7 | 23.1 |
| | Xufan | XF | 1 | 3 | 5 | 1 | 0 | 10 | 10.0 | 30.0 | 50.0 | 10.0 | 0.0 |
| | Zhaoan | ZA | 1 | 5 | 12 | 8 | 4 | 30 | 3.3 | 16.7 | 40.0 | 26.7 | 13.3 |
| | Total | | 9 | 17 | 38 | 19 | 7 | 90 | 10.0 | 18.9 | 42.2 | 21.1 | 7.8 |
| Pearl | Hezikou | HZ | 42 | 17 | 7 | 22 | 1 | 89 | 47.2 | 19.1 | 7.9 | 24.7 | 1.1 |
| | Huishui | HS | 3 | 3 | 0 | 4 | 0 | 10 | 30.0 | 30.0 | 0.0 | 40.0 | 0.0 |
| | Libo | LB | 5 | 0 | 0 | 6 | 0 | 11 | 45.5 | 0.0 | 0.0 | 54.5 | 0.0 |
| | Xiaogulu | XL | 2 | 24 | 0 | 0 | 4 | 30 | 6.7 | 80.0 | 0.0 | 0.0 | 13.3 |
| | Total | | 52 | 44 | 7 | 32 | 5 | 140 | 37.1 | 31.4 | 5.0 | 22.9 | 3.6 |
| | Total | | 347 | 306 | 195 | 375 | 223 | 1446 | 24.0 | 21.2 | 13.5 | 25.9 | 15.4 |

"


Figure 5. This is too small. You could move this figure to the appendix and add a figure with a more informative visualization, maybe zoon in a small area. Maybe you should correlate with some of the PC factors in space, etc.

**Response:** This figure was redrawn and the area with high densities of stations were zoomed to present detailed distributions of flood event classes. We also drew the spatial distributions of load coefficients of all the principal components (PCA1−5) which were provided in the Supplement.

[Figure]

*Figure 5. Spatial variabilities of individual flood event classes in major river basins*

[Figure]

*Figure S1. Spatial distributions of load coefficients of all the principal components.*

Line 288. How can you talk about class per basin if some of them have a few gauges?

**Response:** It was revised to "*In the headstream of Songliao River Basin,.....*"

Line 292. Why does the class 5 increase over time?

**Response:** The increase in Class 5 was probably due to the increase in precipitation amount and duration caused by climate change. This sentence was revised as follows: "*In the headstream stations of Huaihe River Basin, the Class 5 gradually prevail with the annual mean percentage of 41.5±23.7% (n=102), particularly after 2007, whose percentage reaches 63.2±15.8% (n=79). The event numbers of both Classes 1 and 2 gradually decrease, accounting for 33.1±24.4% (n=11) and 8.7±7.1% (n=5) of annual flood events in the period of 1993-1999 and 2011-2015 for the Class 1, respectively, and 20.3±20.9% (n=9) and 2.7±1.3% (n=1) in the period of 1993-1999 and 2011-2015 for the Class 2, respectively. The explanations are that the total precipitation amount and duration probably increase due to the climate change (Dong et al, 2011; Jin et al., 2024).*".


Section 4.4.1. you mainly describe the same information presented in the figure 7. You should add an analysis or discussion about the implication of your findings.

**Response:** This section was revised as follows: *"According to the Monte Carlo permutation test between flood response matrix and control factor matrix (i.e., meteorological and land cover categories) in the individual catchments (Figures 7 and S2–5 in the Supplement), the factors only in the meteorological category are statistically significant for the temporal variabilities of flood events in all*

*the classes, particularly the precipitation factors (e.g., amount, intensity) and aridity index during the events. Taking the Class 1 as an example, the total and mean precipitations, and aridity index during the event ($r_{pcp\_dur}$=0.65–0.99, n=14; $r_{pcp\_av}$=0.70–0.97, n=7; $r_{SPEI\_dur}$=0.52–0.97, n=7) are the major control factors in 44.7% (17/38) of total catchments of the Yangtze River Basin, and Tunxi catchment of the Southeast River Basin and Hezikou catchment of the Pearl River Basin. The contributions of control factors are statistically significant only in the Liangshuikou and Hezikou catchments. In the Liangshuikou catchment, 96.3% of temporal differences are explained, in which the meteorological and land cover categories explain 92.5% and 3.8%, respectively. In the Hezikou catchment, 66.7% of temporal differences are explained, in which the meteorological category and the combined impact explain 49.4% and 17.3%, respectively.*

*In the Class 2, the significant control factors are in the catchments of Yangtze (18.4%, 7/38), Yellow (25%, 1/4) and Pearl (50%, 2/4) River Basins, particularly the total and mean precipitations, and aridity index during the event with the correlation coefficients of 0.61–0.99, 0.58–0.99 and 0.50–0.98, respectively. The contributions only in the Shimenkan, Tangdukou and Xiaogulu catchments are statistically significant with the total values of 90.7–96.8%. The contributions of meteorological category are the greatest with the values of 71.9–95.9%. In the Class 4, the significant control factors are in the catchments of Yellow (75%, 3/4), Songliao (50%, 2/4) and Pearl (50%, 2/4) River Basins, particularly the total precipitation during the event, and the aridity index in the corresponding year with the correlation coefficients of 0.53–1.00 and 0.45–0.93, respectively. The contributions only in the Liangshuikou and Hezikou catchments are statistically significant with the total values of 87.0–98.1%. The factors in the meteorological category also contribute the most considerably with the values of 76.8–82.1%. In the Classes 3 and 5, the contributions are not statistically significant in all the catchments because of the smaller numbers of flood events. However, several important control factors are also statistically significant in the catchments of Yangtze (26.3%, 10/38) and Southeast (40%, 2/5) River Basin for Class 3 (e.g., total and mean precipitations during the event with the correlation coefficients of 0.77–0.99 and 0.70–1.00, respectively), and Huaihe (61.5%, 8/13) and Yangtze (26.3%, 7/38) River Basin for Class 5 (e.g., the aridity index in the corresponding year and during the event, and the annual mean precipitation amount with the correlation coefficients of 0.62–0.86, 0.68–1.00 and 0.65–0.92, respectively)."*

Furthermore, more discussions were given for the control factors and their contributions in the discussion section: *" Similar results were reported in Kuentz et al. (2017), which are that the climatic variables (e.g., precipitation, temperature and aridity index) play the most important role for 75% of total flow signatures and catchment attributes (e.g., area, elevation, slope and river density) are more important for the flood flashiness."*

*"The contribution of meteorological category is the largest in the Class 2, particularly in the Tangdukou catchment of Yangtze River Basin because the flood events in this class usually show quick responses to the precipitation, while the contribution is the lowest in the Class 5 because the river density and river morphology play important roles in the flood storage capacity and routing time in the river system. "*

*"The contributions of catchment attribute category in the slow flood event classes (e.g., Classes 3 and 5) are usually larger than those in the fast flood event classes (e.g., Classes 1, 2 and 4) because the catchment attribute factors are significantly correlated with the flood response metrics in the Classes 3 and 5, particularly the catchment maximum elevation and river density. "*

Figures 7 and 8. Do you need a big figure only to show almost non-significance in the factors?

**Response:** Figure 7 were divided into five subfigures for individual classes. The figure for Class 1 was provided in the manuscript and the other figures for Classes 2–5 were provided in the Supplement.

We also used a table to present the effect contributions of control factor categories on the temporal variabilities of all the flood event classes. The table was given as follows.

**"Table 4. Effect contributions of control factor categories on the temporal variabilities of flood event classes**

| Classes | Stations | Meteorology | Land cover | Combination | All |
|---------|----------|-------------|------------|-------------|-----|
| Class1 | Hezikou | 49.4% | 0.0% | 17.3% | 66.7% |
| | Liangshuikou | 92.4% | 3.8% | 0.1% | 96.3% |
| | Shimenkan | 87.1% | 0.0% | 3.6% | 90.7% |
| Class2 | Tangdukou | 95.9% | 0.0% | 0.0% | 95.9% |
| | Xiaogulu | 71.9% | 0.0% | 24.9% | 96.8% |
| Class3 | - | - | - | - | - |
| Class4 | Hezikou | 82.1% | 0.0% | 16.0% | 98.1% |
| | Liangshuikou | 76.8% | 0.0% | 10.2% | 87.0% |
| Class5 | - | - | - | - | - |

"

[Figure]

*Figure 7. Significant control factors and their correlation coefficients for the temporal variabilities of flood event Class 1 in the individual catchments. The gray color means the control factor without statistical significance.*

*Note: Anhe, Anren, Chengcun, Jiahe, Liangshuikou, Loudi, Pingshi, Shanggao, Shimenkan, Shuangjiangkou, Tangdukou, Tongtang, Xiawan, Yanling, Yanta, Yucun and Yuexi catchments are from the Yangtze River Basin; Tunxi catchment is from Southeast River Basin; Hezikou catchment is from Pearl River Basin.*

[Figure]

*Figure S2. Significant control factors and their correlation coefficients for the temporal variabilities of flood event Class 2 in the individual catchments. The gray color means the control factor without statistical significance.*
*Note: Anren, Dutou, Jiahe, Loudi, Shimenkan, Shuangfeng and Tangdukou catchments are in the Yangtze River Basin; Luanchuan catchment is in the Yellow River Basin; Hezikou and Xiaogulu catchments are in the Pearl River Basin*

[Figure]

*Figure S3. Significant control factors and their correlation coefficients for the temporal variabilities of flood event Class 3 in the individual catchments. The gray color means the control factor without statistical significance.*

*Note: Chengcun, Jinping, Liangshuikou, Loudi, Miping, Shuangfeng, Shuangjiangkou, Tongtang, Yucun and Yuexi catchments are in the Yangtze River Basin; Longshan and Zhaoan catchments are in the Pearl River Basin*

[Figure]

*Figure S4. Significant control factors and their correlation coefficients for the temporal variabilities of flood event Class 4 in the individual catchments. The gray color means the control factor without statistical significance.*

*Note: Jingyu and Yitong catchments are in the Songliao River Basin; Luanchuan, Qiaotou and Tantou catchments are in the Yellow River Basin; Luzhuang and Ziluoshan catchments are in the Huaihe River Basin; Dutou, Liqingdian, Liangshuikou, Pingshi, Shuangfeng, Xupu, Yanling and Yuexi catchment are in the Yangtze River Basin; Zhaoan catchment is in the Southeast River Basin; Hezikou and Libo catchments are in the Pearl River Basin*

[Figure]

*Figure S5. Significant control factors and their correlation coefficients for the temporal variabilities of flood event Class 5 in the individual catchments. The gray color means the control factor without statistical significance.*

*Note: Beimiaoji, Huangnizhuang, Peihe, Qilin, Xiagushan, Xinxian, Zhongtang and Zhuganpu catchments are in the Huaihe River Basin; Anhe, Anren, Liqingdian, Miping, Tanghe, Tonggu and Xixia catchments are in the Yangtze River Basin.*

Figure 9. A rainbow color scale is not recommended because it is very difficult to recognize visually what value is higher than others.

**Response:** This figure was redrawn following your comments.

[Figure]

*Figure 8. Significant control factors and their correlation coefficients for the variabilities of individual flood event classes (i.e., Classes 1–5). The gray color means the control factor without statistical significance.*

Line 354-361. What about the high collinearity between meteorological factors? If you have many factors representing the same, the relative importance decreases. I would try to group them for more general characteristics because you have many factors in the range r=0.15-0.21.

**Response:** We selected the potential control factors of meteorology and physio-geography as many as possible to comprehensively detect the control mechanisms according to the existing studies (Ali *et al.*, 2012; Brunner *et al.*, 2018; Merz and Blöschl, 2003; Zhang *et al.*, 2022). Our adopted constrained rank analysis is the extended method of principal component analysis combined with regression analysis. It has strong advantages to solve multiple linear regressions and interactions between

dependent and independent variable matrixes which are transformed into a few independent composite factors (ter Braak, 1986; Legendre and Anderson, 1999), and is beneficial to quantify the effects of explanatory metrics on a response metrics and to find the most important factors. It has been commonly used in testing the multispecies response to environmental variables in the biological or ecological sciences (Legendre and Anderson, 1999), effects of physio-geographical factors and human activities on diffuse nutrient losses or water quality (Zhang et al., 2016; Shi et al., 2017), and so on. Therefore, although some factors have high collinearities, all the factors are transformed into a few independent composite factors firstly, and then multiple linear regressions and interactions between dependent and independent composite factors are detected.

The constrained rank analysis method is explained in more detail as follows: *"The widely adopted methods of constrained rank analysis are the Redundancy Analysis (RDA) and the Canonical Correlation Analysis (CCA). The RAD is a linear model and the CCA is a unimodal model, both of which are the extended methods of principal component analysis combined with regression analysis. These methods have strong advantages to solve multiple linear regressions and interactions between dependent and independent variable matrixes which are transformed into a few independent composite factors (ter Braak, 1986; Legendre and Anderson, 1999), and are beneficial to quantify the effects of explanatory metrics on a response metrics and to find the most important factors, which have been commonly used in testing the multispecies response to environmental variables in the biological or ecological sciences (Legendre and Anderson, 1999), effects of physio-geographical factors and human activities on diffuse nutrient losses or water quality (Zhang et al., 2016; Shi et al., 2017), and so on."*

**Response:** These paragraphs were revised following your constructive comments. The results were comprehensively summarized, and the analysis were presented. The revisions were given as follows:*" The significant control factors are mainly the meteorological factors in the antecedent seven days and during the flood events for the Class 2, the meteorological factors during the flood events and catchment elevation for the Class 3, the meteorological factors in the antecedent seven days, during the flood events and at the annual scale, and the catchment factors related to slope and river for the Classes 4 and 5, respectively. The specific factors are the precipitation and potential evapotranspiration in the antecedent seven days ($r_{pcp\_ant}$=0.15 and $r_{pet\_ant}$=0.14), precipitation and aridity index during the flood events ($r_{pcp\_dur}$=0.73, $r_{pcp\_av}$=0.44, $r_{pcp\_max}$=0.38, $r_{pcp\_Tbeg}$=0.19, $r_{pcp\_Tdur}$=0.24 and $r_{SPEI\_dur}$=0.32) for the Class 2, the precipitation and aridity index during the flood events ($r_{pcp\_dur}$=0.74, $r_{pcp\_av}$=0.38, $r_{pcp\_max}$=0.25, and $r_{SPEI\_dur}$=0.36) in the meteorological category, and the catchment center elevation ($r_{Elevation}$=0.19) and maximum elevation ($r_{MaxiElev}$=0.31) in the catchment attribute category for the Class 3, the precipitation and potential evapotranspiration in the antecedent*

[revised manuscript text omitted]

---

## Author Comment (AC2)

**RC2: 'Comment on hess-2024-126', Anonymous Referee #2, 09 Jun 2024**

Heterogeneities in meteorological and underlying surface conditions usually result in remarkable spatial and temporal variabilities of flood events. It is very beneficial to investigate comprehensive variation characteristics of flood events and their formation mechanisms by clustering massive homogeneous events into some representative classes. This manuscript made an interesting contribution to understand meteorological and physio-geographical controls of flood event variabilities at class scale across China. Over a thousand flood events were selected from most of river basins in China. The sizes of flood events, meteorological and physio-geographical factors were impressive, and the investigation was convincing because multiple statistical analysis methods were adopted, including the hierarchical and partitional clustering methods, constrained rank analysis and Monte Carlo permutation test. This topic fits well with the scope of HESS, and the study is original. I think that some moderate revisions are required for this manuscript before publication.

**Response:** Thank you very much for your careful review and constructive comments. We revised this manuscript substantially and provided point-by-point responses to all the comments and suggestions of reviewers accordingly.

Line 104, how to "assess" the potential meteorological and physio-geographical control factors of flood events?

**Response:** This sentence was revised to "*Meteorological, catchment and land cover data sources were collected together to calculate the potential meteorological and physio-geographical control factors and assess their contributions on the spatial and temporal variabilities of flood event classes.*"

Line 123, the $T_{bgn}$ is calculated using the circular variable. Please explain the reason.

**Response:** The circular variable is widely used to characterize the timing or seasonality of hydrological variables (e.g., flood and precipitation) (Fisher, 1993; Black and Werritty, 1997; Villarini, 2016; Hall and Blöschl, 2018). This method translates the calendar date into the polar coordinates on the circumference of a circle, which is beneficial to distinguish the seasonal pattern (Fisher, 1993; Dhakal et al., 2015). The explanation was given as follows:

*"$T_{bgn}$ is characterized using the circular statistical approach which translates the calendar date into the polar coordinates on the circumference of a circle, and is beneficial to distinguish the seasonal pattern (Fisher, 1993; Dhakal et al., 2015)."*

*References*

*Black, A.R., Werritty, A: Seasonality of flooding: a case study of North Britain, J. Hydrol.,195:1–25, https://doi.org/10.1016/S0022-1694(96)03264-7, 1997.*

*Dhakal, N., Jain, S., Gray, A. , Dandy, M., and Stancioff, E.: Nonstationarity in seasonality of extreme precipitation: a nonparametric circular statistical approach and its application, Water Resour. Res., 51(6), 4499-4515. https://doi.org/10.1002/2014WR016399, 2015.*

*Fisher, N.I.: Statistical Analysis of Circular Data. Cambridge University Press, Cambridge, UK, 1993.*

*Hall, J., and Blöschl, G.: Spatial patterns and characteristics of flood seasonality in Europe, Hydrol. Earth Syst. Sc., 22, 3883–3901, https://doi.org/10.5194/hess-22-3883-2018, 2018.*

*Villarini, G. On the seasonality of flooding across the continental United States, Adv. Water Resour., 87, 80-91, https://doi.org/10.1016/j.advwatres.2015.11.009, 2016.*

In the section of methods, many of flood behavior metrics or control factors were not independent. Why were they selected? Please clarify specifically.

**Response:** We selected the flood response metrics or potential control factors as many as possible to fully characterize flood events and to comprehensively detect the control mechanisms according to the existing studies (Ali *et al*., 2012; Brunner *et al*., 2018;

Merz and Blöschl, 2003; Zhang *et al*., 2022). All the correlated metrics or factors were transformed into a few independent composite metrics without losing the metric or factor information using the principal component analysis and the constrained rank analysis, respectively.

For the flood response metrics, the magnitude, variability, timing, duration, and rate of changes were widely-accepted as the main five components to characterize the entire flood events. Thus, eight related metrics were selected including total flood volume, maximum flood peak, coefficient of variation, timings of flood event and maximum flood peak, flood event duration, and rates of positive and negative changes, which covered all the main five components. Additionally, flood peak number is one of the most important metrics for flood control, which was also selected to characterize the flood events.

For the potential control factors of meteorology and physio-geography, precipitation and evapotranspiration related factors were selected including the amounts and intensities in the antecedent period and during the events, all of which mainly affect the flood yield processes. The catchment attributes were selected including position (longitude and latitude), elevation, catchment area, slope and its length, river density and slope, ratio of river width to depth, all of which mainly affect the flood yield and routing processes. The area percentages of main land covers were also adopted, which mainly affect the flood yield and overland routing processes.

The revisions were provided as follows: *"The magnitude, variability, timing, duration, and rate of changes are widely-accepted as the main five components to characterize the entire flood events (Poff et al., 2007) and thus…,nine metrics are used to fully characterize the response of flood events"*

*"The potential control factors are selected as many as possible to investigate the control mechanisms on the variability of flood event classes according to the existing studies and the total number is 34 meteorological, catchment and land cover factors in all the catchments. In the meteorological category, 17 factors related to precipitation, potential evapotranspiration and aridity index are selected, including the amounts, intensities and timing factors during flood events, in the antecedent period and at annual scale. …. All of these factors mainly affect the flood yield processes (Merz and Blöschl, 2003; Aristeidis et al., 2010; Zhang et al., 2022).*

*For the physio-geographical factors, the 10 catchment attributes are selected, including catchment location, area, elevation and slope, river density and slope. All these factors mainly affect the flood yield and routing processes (Ali et al., 2012; Kuentz et al., 2017). Seven land cover factors are selected, including the area fractions of paddy, dryland, forest, grassland, water, urban and rural area to the total catchment, respectively for the seven land cover periods. All of these factors mainly affect the flood yield and overland routing processes (Kuentz et al., 2017; Zhai et al., 2021)."*

*References*

*Ali, G., Tetzlaff, D., Soulsby, C., McDonnell, J. J., and Capell, R.: A comparison of similarity indices for catchment classification using a cross-regional dataset, Adv. Water Resour., 40, 11–22, https://doi.org/10.1016/j.advwatres.2012.01.008, 2012.*

*Aristeidis, G. K., Tsanis, I. K., and Daliakopoulos, I. N.: Seasonality of floods and their hydrometeorologic characteristics in the island of Crete, J. Hydrol., 394(1–2), 90-100, https://doi.org/10.1016/j.jhydrol.2010.04.025, 2010.*

*Brunner, M. I., Viviroli, D., Furrer, R., Seibert, J., and Favre, A. C.: Identification of flood reactivity regions via the functional clustering of hydrographs, Water Resour. Res., 54, 1852-1867, https://doi.org/10.1002/2017WR021650, 2018.*

*Kuentz, A., Arheimer, B., Hundecha, Y., and Wagener, T.: Understanding hydrologic variability across Europe through catchment classification, Hydrol. Earth Syst. Sc., 21, 2863–2879. https://doi.org/10.5194/hess-21-2863-2017, 2017.*

*Merz, R., and Blöschl, G.: A process typology of regional floods, Water Resour. Res., 39(12), 1340, https://doi.org/10.1029/2002WR001952, 2003.*

*Poff, N. L., Olden, J. D., Merritt, D., and Pepin, D.: Homogenization of regional river dynamics by dams and global biodiversity implications, P. Natl. Acad. Sci. USA, 104, 5732–5737, https://doi.org/10.1073/pnas.0609812104, 2007.*

*Zhai, X. Y., Guo, L., and Zhang, Y. Y.: Flash flood type identification and simulation based on flash flood behavior indices in China, Sci. China Earth Sci., 64(7), 1140–1154, https://doi.org/10.1007/s11430-*

*020-9727-1,2021.*

*Zhang, S.L., Zhou, L.M., Zhang, L., Yang, Y.T., Wei, Z.W., Zhou, S., Yang, D.W., Yang, X. F., Wu, X.C., Zhang, Y.Q., and Dai, Y.J.: Reconciling disagreement on global river flood changes in a warming climate, Nat. Clim. Change, 12, 1160–1167, https://doi.org/10.1038/s41558-022-01539-7, 2022.*

Lines 142-147, 22 criteria were used to assess the classification performance and determine the best number of clusters. I agreed that it would be a robust way to select an optimal class number. However, most of the criteria were given as an abbreviation. Could you please give a detailed explanation about these criteria including full names, equations and units in the supplementary material?

**Response:** All the criteria were explained clearly, which was provided in the Supplement.

*"Table S2. Criteria of classification performance assessment*

| ID | Criteria name | Abbreviation | Equation | Reference |
|----|---------------|--------------|----------|-----------|
| 1 | Krzanowski-Lai | KL | $KL(q) = \left\| \dfrac{DIEF_q}{DIEF_{q+1}} \right\|$ | Krzanowski and Lai 1988 |
| 2 | Calinski-Harabasz | CH | $CH(q) = \dfrac{trace(B_q)/(q-1)}{trace(W_q)/(n-q)}$ | Calinski and Harabasz 1974 |
| 3 | Hartigan | Hartigan | $Hartigan = \left( \dfrac{trace(W_q)}{trace(W_{q+1})} - 1 \right)(n-q-1)$ | Hartigan 1975 |
| 4 | Cubic Clustering Criterion | CCC | $CCC = \ln\left[ \dfrac{1-E(R^2)}{1-R^2} \right] \dfrac{\sqrt{\frac{nq}{2}}}{(0.001+E(R^2))^{1.2}}$ | Sarle 1983 |
| 5 | Scott | Scott | $Scott = n\log\dfrac{\det(T)}{\det(W_q)}$ | Scott and Symons 1971 |
| 6 | Marriot | Marriot | $Marriot = q^2\det(W_q)$ | Marriot 1971 |
| 7 | Trcovw | TrCovW | $Trcovw = trace(COV(W_q))$ | Milligan and Cooper 1985 |
| 8 | Tracew | TraceW | $Tracew = trace(W_q)$ | Milligan and Cooper 1985 |
| 9 | Friedman | Friedman | $Friedman = trace(W_q^{-1}B_q)$ | Friedman and Rubin 1967 |
| 10 | Silhouette | Silhouette | $Silhouette = \dfrac{\sum_{i=1}^{n}S(i)}{n}, Silhouette \in [-1,1]$ | Rousseeuw 1987 |
| 11 | Ratkowsky-Lance | Ratkowsky | $Ratkowsky = \dfrac{\bar{S}}{q^{1/2}}$ | Ratkowsky and Lance 1978 |
| 12 | Ball | Ball | $Ball = \dfrac{W_q}{q}$ | Ball and Hall 1965 |
| 13 | Ptbiserial | Ptbiserial | $Ptbiserial = \dfrac{[\bar{S}_b - \bar{S}_w][N_w N_b/N_t^2]^{1/2}}{s_d}$ | Milligan 1980, 1981 |
| 14 | Dunn | Dunn | $Dunn = \dfrac{\min\limits_{1\leq i<j\leq q}(C_i,C_j)}{\max\limits_{1\leq k\leq q} diam(C_k)}$ | Dunn 1974 |

| 15 | Rubin | Rubin | $\text{Rubin} = \dfrac{\det(T)}{\det(W_q)}$ | *Friedman and Rubin 1967* |
|----|-------|-------|------|------|
| 16 | *C-Index* | *Cindex* | $\text{Cindex} = \dfrac{S_w - S_{min}}{S_{max} - S_{min}}, S_{min} \neq S_{max}, \text{Cindex} \in (0,1)$ | *Hubert and Levin 1976* |
| 17 | *Davies-Bouldin* | *DB* | $DB(q) = \dfrac{1}{q}\sum_{k=1}^{q}\max_{k \neq l}(\dfrac{\delta_k + \delta_l}{d_{kl}})$ | *Davies and Bouldin 1979* |
| 18 | *Duda* | *Duda* | $\text{Duda} \geq 1 - \dfrac{2}{\pi p} - \sqrt{\dfrac{2(1 - \frac{8}{\pi^2 p})}{n_m p}} = critValue\_Duda$ | *Duda and Hart 1973* |
| 19 | *Pseudo t²* | *Pseudot2* | $\text{Pseudot2} = \dfrac{V_{kl}}{\frac{W_k + W_l}{n_k + n_k - 2}}$ | *Duda and Hart 1973* |
| 20 | *McClain-Rao* | *McClain* | $\text{McClain} = \dfrac{\bar{S}_w}{\bar{S}_b} = \dfrac{S_w / N_w}{S_b / N_b}$ | *McClain and Rao 1975* |
| 21 | *SD validity* | *SDindex* | $\text{SDindex}(q) = \alpha\text{Scat}(q) + \text{Dis}(q)$ | *Halkidi et al. 2000* |
| 22 | *SDbw validity* | *SDbw* | $\text{SDbw}(q) = \text{Scat}(q) + \text{Density.bw}(q)$ | *Halkidi and Vazirgiannis 2001* |

*Note: q is the number of clusters; n is the number of observations; p is the number of variables; $B_q$ is the between-group dispersion matrix for data clustered into q clusters; $W_q$ is the within-group dispersion matrix for data clustered into q clusters; $R^2$ is the coefficient of determination; T is the total sum of squares; $S_b$ is the sum of the between-cluster distances; $S_w$ is the sum of the within-cluster distances; $\bar{S}_b$ is the ratio of the $S_b$ and $N_b$; $\bar{S}_w$ is the ratio of the $S_w$ and $N_w$; $N_w$ is the total number of pairs of observations belonging to the same cluster; $N_b$ is the total number of pairs of observations belonging to different clusters; $N_t$ is the total number of pairs of observations in the data set; $S_{max}$ is the sum of the $N_w$ largest distances between all the pairs of points in the entire data set; $S_{min}$ is the sum of the $N_w$ smallest distances between all the pairs of points in the entire data set (there are $N_t$ such pairs); $S_d$ is the standard deviation of all distances; $\bar{S}$ is the average of the ratios of sum and total sum of squares between the clusters for each variable; i is the number ranges from 1 to n; j is the number ranges from 1 to p; k, l and m is the cluster number ranges from 1 to q; $C_i$; $C_j$ and $C_k$ are the different clusters; $n_k$, $n_l$ and $n_m$ are the number of objects in cluster $C_k$, $C_l$ and $C_m$, respectively; $W_k$, $W_l$ and $W_m$ are the squared errors of the different clusters; $V_{kl}$ equals $W_m$ minus $W_k$ and then minus $W_l$; $d_{kl}$ is the distance between centroids of clusters $C_k$ and $C_l$; $\delta_k$ and $\delta_l$ are the standard deviation of the distance of objects in cluster $C_k$ and $C_l$, respectively."*

*References*

*Ball, G. H., and Hall, J.: ISODATA: A Novel Method of Data Analysis and Pattern Classification, Stanford Research Institute, Menlo Park, NTIS No. AD 699616, 1965.*

*Calinski, T., and Harabasz, J.: A dendrite method for cluster analysis, Communications in Statistics-Theory and Methods, 3, 1-27, https://doi.org/10.1080/03610927408827101, 1974.*

*Davies, D. L., and Bouldin, D. W.: A cluster separation measure, IEEE Transactions on Pattern Analysis and Machine Intelligence, 1, 224-227, https://doi.org/10.1109/TPAMI.1979.4766909, 1979.*

Duda, R. O., and Hart, P. E.: *Pattern Classification and Scene Analysis*, John Wiley & Sons, New York, 1973.

Dunn, J. C.: Well-separated clusters and optimal fuzzy partitions, Journal of Cybernetics, 4, 95-104, https://doi.org/10.1080/01969727408546059, 1974.

Friedman, H. P., and Rubin, J.: On some invariant criteria for grouping data, Journal of the American Statistical Association, 62, 1159-1178, https://doi.org/10.1080/01621459.1967.10500923, 1967.

Friedman, H. X., and Rubin, J.: On some invariant criteria for grouping data, Journal of the American Statistical Association, 62, 1159-1178, https://doi.org/10.2307/2283767, 1967.

Halkidi, M., and Vazirgiannis, M.: Clustering validity assessment: Finding the optimal partitioning of a data set, in: Proceedings 2001 IEEE International Conference on Data Mining, San Jose CA, USA, 29 November-02 December 2001, 187-194, 2001.

Halkidi, M., Vazirgiannis, M., and Batistakis, I.: Quality scheme assessment in the clustering process, in: Principles of Data Mining and Knowledge Discovery: 4th European Conference, PKDD 2000 Lyon, France, 13-16 September 2000, 265-276, 2000.

Hartigan, J. A.: *Clustering Algorithms*, John Wiley & Sons, New York, ISBN 047135645X1975, 1975.

Hubert, L. J., and Levin, J. R.: A general statistical framework for assessing categorical clustering in free recall, Psychological Bulletin, 83, 1072-1080, https://doi.org/10.1037/0033-2909.83.6.1072, 1976.

Krzanowski, W., and Lai, Y.: A criterion for determining the number of groups in a data set using sum-of-squares clustering, Biometrics, 44, 23-34, https://doi.org/10.2307/2531893, 1988.

Marriott, F. H. C.: Practical problems in a method of cluster analysis, Biometrics, 27, 501-514, https://doi.org/10.2307/2528592, 1971.

McClain, J. O., and Rao, V. R.: Clustisz: A program to test for the quality of clustering of a set of objects, Journal of Marketing Research, 12, 456-460, https://doi.org/10.2307/3151097, 1975.

Milligan, G. W., and Cooper, M. C.: An examination of procedures for determining the number of clusters in a data set, Psychometrika, 50, 159-179, https://doi.org/10.1007/BF02294245, 1985.

Milligan, G. W.: A Monte Carlo study of thirty internal criterion measures for cluster analysis, Psychometrika, 46, 187-199, https://doi.org/10.1007/BF02293899, 1981.

Milligan, G. W.: An examination of the effect of six types of error perturbation on fifteen clustering algorithms, Psychometrika, 45, 325-342, https://doi.org/10.1007/BF02293907, 1980.

Ratkowsky, D. A., and Lance, G. N.: Criterion for determining the number of groups in a classification, Australian Computer Journal, 10, 115-117, 1978.

Rousseeuw, P. J.: Silhouettes: a graphical aid to the interpretation and validation of cluster analysis, Journal of Computational and Applied Mathematics, 20, 53-65, https://doi.org/10.1016/0377-0427(87)90125-7, 1987.

Sarle, W. S.: SAS Technical Report A-108, Cubic Clustering Criterion, SAS Institute Inc, Cary, NC, 1983.

Scott, A. J., and Symons, M. J.: Clustering methods based on likelihood ratio criteria, Biometrics, 27, 387-397, https://doi.org/10.2307/2529003, 1971.

Lines 285-297, the comparisons of flood events among different classes are largely based on percentages, but the flood event numbers at many stations were not the same. Please give the detailed introductions about the spatial and temporal distributions of flood event classes.

**Response:** This paragraph and Figure 6 were revised and the flood event numbers in all the classes and basins were added according to your comments. The revised paragraph was provided as follows:

*"According to the interannual distributions of individual classes (Figure 6), all the classes are evenly distributed, whose annual mean percentages are 24.0±5.9%, 21.2±6.4%, 13.5±7.7%, 25.9±6.2%, and 15.4±12.5%, respectively. However, the interannual distributions of individual classes are quite distinct at different stations, particularly in the Songliao River Basin. In the headstream stations of Songliao River Basin, the dominant class is Class 4 with the annual mean percentage of 26.1±38.3% (n=32) though flood events are missed in several years due to the dry period. In the headstream stations of Yellow River Basin, the Class 4 is also dominant across the whole period with the annual mean percentage of 58.1±33.9% (n=67), particularly in 1994-1996, 1999 and 2007. In the headstream stations of Huaihe River Basin, the Class 5 gradually prevail with the annual mean percentage of 41.5±23.7% (n=102), particularly after 2007, whose percentage reaches 63.2±15.8% (n=79). The event numbers of both Classes 1 and 2 gradually decrease, accounting for 33.1±24.4% (n=11) and 8.7±7.1% (n=5) of annual flood events in the period of 1993-1999 and 2011-2015 for the Class 1, respectively, and 20.3±20.9%*

*(n=9) and 2.7±1.3% (n=1) in the period of 1993-1999 and 2011-2015 for the Class 2, respectively. The explanations are that the total precipitation amount and duration probably increase due to the climate change (Dong et al, 2011; Jin et al., 2024). In the headstream stations of Yangtze River Basin, the Classes 1, 2 and 4 are dominant, accounting for 29.3±9.6% (n=251), 23.0±11.5% (n=197) and 21.1±7.0% (n=181) of annual mean flood events, respectively. Although the interannual changes of event numbers of Classes 1 (n=1–21), 2 (n=1–14) and 4 (n=1–16) are considerable, those of class percentages are relatively uniform except 2015. In the headstream stations of Southeast River Basin, the Class 3 gradually prevail after 2000 with the annual mean percentage of 46.2±32.5% (n=39). In the headstream stations of Pearl River Basin, the Class 1 is dominant with the annual mean percentage of 36.0±24.0% (n=52), but gradually shifts to Class 2 which accounts for 30.0±25.2% of annual mean flood events (n=40), particularly after 2008.*"

*References:*

*Dong, Q., Chen, X., and Chen, T.: 2011. Characteristics and changes of extreme precipitation in the Yellow-Huaihe and Yangtze-Huaihe Rivers Basins, China, J. Climate, 24(14), 3781-3795, https://doi.org/10.1175/2010JCLI3653.1, 2011.*

*Jin, H., Chen, X., and Adamowski, J. H. S.: Determination of duration, threshold and spatiotemporal distribution of extreme continuous precipitation in nine major river basins in China, Atmos Res, 300, 107217, https://doi.org/10.1016/j.atmosres.2023.107217, 2024.*

In Figure 1, the main river names should be replaced by the river basin names.

 **Response:** It was revised accordingly.

[Figure]

*Figure 1. Spatial distributions of all the selected flood events and their corresponding climate types*

In Figure 5, the legend "Flood classes" should be changed to "Flood event classes". Please remove shading from the stacked bars. That adds no information.

**Response:** It was revised accordingly.

[Figure]

*Figure 5. Spatial variabilities of individual flood event classes in major river basins*

What are the means of 21 in Figure 5 and 0.46 in Figure 8?

**Response:** The number in the figure means the measuring scale of the bar, which is the number of flood event classes at each station. Figure 5 was revised following the comments of Reviewer 1 and Figure 8 was changed to Table 4.

In Figure 6, I suggested that the flood event numbers could be given for every year in all the basins.

**Response:** It was revised accordingly.

[Figure]

*Figure 6. Interannual variabilities of individual flood event classes and their percentages in major river basins*

In Figure 7, it should be changed to a single column of the five cases. The coefficients should be "correlation coefficients".

**Response:** This figure was revised following the comments of you and Reviewer 1.

[Figure]

*Figure 7. Significant control factors and their correlation coefficients for the temporal variabilities of flood event class 1 in the individual catchments. The gray color means the control factor without statistical significance.*
*Note: Anhe, Anren, Chengcun, Jiahe, Liangshuikou, Loudi, Pingshi, Shanggao, Shimenkan, Shuangjiangkou, Tangdukou, Tongtang, Xiawan, Yanling, Yanta, Yucun and Yuexi catchments are from the Yangtze River Basin; Tunxi catchment is from Southeast River Basin; Hezikou catchment is from Pearl River Basin.*

---

## Author Response (AR1)

**Comments in black and our response in blue**

**RC1: 'Comment on hess-2024-126', Anonymous Referee #1, 09 Jun 2024**

The paper provides a comprehensive analysis of three primary classifications for a catchment: meteorological, attributes, and response. By correlating this information, the paper identifies characteristic classes of flood responses. The main findings show that meteorological data has a much greater impact on flood response compared to land cover and catchment attributes. However, certain catchment attributes were also found to be correlated with the response.

**Response:** Thank you very much for your careful review and constructive comments. We revised this manuscript substantially and provided point-by-point responses to all the comments and suggestions of reviewers accordingly.

Here are my main concerns about this paper:

1. The results don't contribute new knowledge about the streamflow-generating process. It's well known that streamflow is mainly controlled by factors such as precipitation, intensity, duration, and its distribution. A similar analysis using the rational method could yield the same results as presented in this paper.

**Response:** We appreciate your critical comments. In our study, the main motivations are to investigate some manageable flood event classes from massive events across China with statistical significance and to quantify the meteorological and physio-geographical controls of spatial and temporal variabilities of these flood event classes using the clustering method, constrained rank analysis and Monte Carlo permutation test. We agreed that this study did not contribute new mechanisms about the streamflow-generating process at event scale because the investigation was quite difficult from massive heterogenous flood events in space and time at large scale.

However, existing studies usually focused on impacts of changes in meteorological or underlying surface conditions on specific flood metrics (e.g., magnitude, peak and timings) using trend separation method, correlation testing, mathematical modelling, and so on (Berghuijs et al., 2016; Tarasova et al., 2018; Liu et al., 2020). All of these studies were implemented at event scale or in catchments with certain landscapes and climates, which were insufficient for the comprehensive flood change investigation and generalized results (Tarasova et al., 2019; Zhang et al., 2020). Therefore, we explored the control mechanisms of meteorological and physio-geographical factors on flood event variabilities at class scale across China. The primary meteorological and physiogeographical control factors were identified for different flood event classes clustered from over one thousand flood events, and their contributions of the class variabilities were quantified for individual classes. All of these analyses were implemented in more heterogeneous catchments with wider meteorological and physio-geographical conditions and flood events, and provided more comprehensive insights into meteorological and physio-geographical controls of variabilities of flood event classes in China.

To make the novelty and contributions of our studies clearer, we made several revisions. The manuscript was revised as follows: "Our study investigates comprehensive manageable flood event classes from 1446 unregulated flood events in 68 headstream catchments in China using the hierarchical and partitional clustering methods. Control mechanisms of meteorological and physiogeographical factors (e.g., meteorology, land cover and catchment attributes) on spatial and temporal variabilities of individual flood event classes are explored using constrained rank analysis and Monte Carlo permutation test." (see Lines 12–17 in the manuscript with track changes)

"Existing studies provide insights on impacts of changes in meteorological or underlying surface conditions on specific flood metrics (e.g., magnitude, peak and timings) and their changes using trend

separation method, correlation testing, mathematical modelling, and so on (Berghuijs et al., 2016; Tarasova et al., 2018; Liu et al., 2020; Wang et al., 2024). However, all of these studies are implemented at event scale or in catchments with certain landscapes and climates, which are insufficient for the comprehensive flood change investigation and generalized results (Tarasova et al., 2019; Zhang et al., 2020)." (see Lines 27–32 in the manuscript with track changes)

"Over one thousand unregulated flood events at 68 heterogeneous catchments with wider meteorological and physio-geographical conditions are selected for our study." (see Lines 91–93 in the manuscript with track changes)

"This study provides more comprehensive insights into meteorological and physio-geographical controls of variabilities of flood event classes at large scale, and provides the mechanism supports for predicting flood event classes." (see Lines 100–102 in the manuscript with track changes)

**References**

- Berghuijs, W. R., Woods R. A., Hutton C. J., and Sivapalan M.: Dominant flood generating mechanisms across the United States, Geophys. Res. Letters, 43, 4382–4390, http://doi.org/10.1002/2016GL068070, 2016.
- Liu, J.Y., Feng, S.Y., Gu, X.H., Zhang, Y.Q., Beck, H.E., Zhang, J.W., and Yan, S.: Global changes in floods and their drivers, J. Hydrol., 614, 128553. https://doi.org/10.1016/j.jhydrol.2022.128553, 2020
- Tarasova, L., Basso, S., Zink, M. and Merz, R.: Exploring controls on rainfall-runoff events: 1. Time series-based event separation and temporal dynamics of event runoff response in Germany, Water Resour. Res. 54, 7711–7732, https://doi.org/10.1029/2018WR022587, 2018.
- Wang, H., Liu, J.G., Klaar, M., Chen, A.F., Gudmundsson, L., and Holden, J.: Anthropogenic climate change has influenced global river flow seasonality, Science, 383(6686), 1009-1014, https://doi.org/10.1126/science.adi9501, 2024.

2. While the classification found in the paper may have value for local or basin analysis, most of the results cannot be applied to other regions or countries. The attempt to connect with other countries in the discussion is qualitative and not valid for comparison without quantitative analysis. What is considered high or low, fast or slow in one country could be entirely different in another.

**Response:** Thank you very much for your constructive comments.

For the applicability of our study, we provided an approach to investigate some manageable flood event classes from massive events at large scale and to quantify the meteorological and physio-geographical controls of spatial and temporal variabilities of flood event classes. The approach could also be applied easily to other regions or countries if a great number of flood events were collected. The main motivations and implications of this study were clarified as follows: *"This study provides more comprehensive insights into meteorological and physio-geographical controls of variabilities of flood event classes at large scale, and provides the mechanism supports for predicting flood event classes."* in the introduction section (see Lines 100–102 in the manuscript with track changes), and *"Our study provided an approach to investigate some manageable flood event classes from massive events at large scale and to quantify the meteorological and physio-geographical controls of spatial and temporal variabilities of flood event classes. The approach could be applied easily to other regions or countries if a great number of flood event classes. The approach could be applied easily to other regions or countries if a great number of flood event classes.*

For the comparability of our study, we agreed that the results were difficult to quantitatively compare with most existing studies because the adopted classification methods and boundaries of individual classes were usually different. The widely-adopted classification method categories were presented in the revision to explain the comparability of classification results. "According to the classification procedure, there are two widely-adopted approaches, namely the tree clustering methods (e.g., decision tree, regression tree, fuzzy tree and random forest) (Sikorska et al., 2015; Brunner et al., 2017) and the non-tree clustering methods

(e.g., single linkage, complete linkage, average linkage, centroid linkage, ward linkage, k-mean, kmedoids) (Zhang et al., 2020; Zhai et al., 2021). The tree clustering methods as the hard clustering methods, are implemented to binarily split all the flood events successively into smaller classes of similar flood events according to the thresholds of flood response metrics until obtaining final classes (Sikorska et al., 2015; Brunner et al., 2017). The classification results could be applicable to other basins and the flood response characteristics of different studies would be directly comparable if the same thresholds are adopted. However, these methods assume that the boundaries of flood response metrics in different classes are clear and the thresholds of flood response metrics should be predefined and should not overlap among different classes (Olden et al., 2012; Sikorska et al., 2015; Zhai et al., 2021). Additionally, the classification is very sensitive to the thresholds, whose small changes would cause different flood event classes (Olden et al., 2012; Sikorska et al., 2015). Therefore, it will be difficult to define the thresholds clearly to get robust classification performance. The non-tree clustering methods as the soft clustering methods, are implemented to directly split all the flood events according to different division rules of the comprehensive similarity measures of flood event shapes or metrics (Olden et al., 2012; Zhang et al., 2020). The class boundaries of flood response metrics are not clear, which are mainly based on sufficient of heterogeneous flood events (Sikorska et al., 2015). The flood response characteristics of individual classes were usually qualitatively described to distinguish the differences among classes (Olden et al., 2012; Tarasova et al., 2019; Zhang et al., 2020). Therefore, the classification results obtained from different flood event samples are still difficult to quantitatively compare even though the flood response characteristics or hydrographs in the certain class are similar (e.g., high or low, fast or slow floods) (Zhang et al., 2024). However, these methods were widely-used due to their ease of use (Olden et al., 2012; Tarasova et al., 2019; Zhang et al., 2020)." (see Lines 59-86 in the manuscript with track changes)

We also discussed the reliability of our classifications in China and tried to make quantitative comparisons with the existing studies of other regions. In our study, a total of 1446 unregulated flood events in 68 headstream catchments were selected for classification. All the catchments were mainly spread across the floodprone areas and in all the monsoon controlled climate types of China, except tropical climate in the islands (i.e., A). The selected flood events were sufficient to represent the flood response characteristics of headstream catchments in main river basins of China. Thus, our classification results and the control mechanisms of variability of flood event classes would be applied in other regions with similar climate types. The revisions were given as follows: "thus the region in the monsoon controlled climate types is usually considered as the flood-prone area of China (China Institute of Water Resources and Hydropower Research and Research Center on Flood and Drought Disaster Prevention and Reduction, the Ministry of Water Resources, 2021)." and "Sixty-eight headstream stations spread across the flood-prone areas were selected with catchment areas ranging from 21 km2 to 4830 km2, which were in all the monsoon controlled climate types of China, except tropical climate in the islands (i.e., A)." in the section of study area and data sources (see Lines 111–114 in the manuscript with track changes), and "All the selected flood events were sufficient to represent the flood response characteristics of headstream catchments in main river basins of China. Thus, our classification results and the control mechanisms of variability of flood event classes would be applied in other regions with similar climate types." in the discussion section (see Lines 646–649 in the manuscript with track changes).

The values of some critical metrics of individual classes were also quantitatively compared with those of existing studies in the discussion section. The revisions were given as follows: "The specific values and boundaries of flood response metrics of individual classes were difficult to quantitatively compare with most existing studies because the adopted classification methods were usually different. However, the flood event classes with similar hydrographs or response mechanisms were also found in the existing studies. ..... The flood response characteristics in these two classes are similar to the flash floods and short-rain floods in Austria (Merz and Blöschl 2003), and fast events in Switzerland (Brunner et al., 2018) and China (Zhai et al., 2021)." (see Lines 573–581 in the manuscript with track changes)

"The flood response characteristics are similar to the high unit peak flood in the west coast of the USA (Saharia et al., 2017) because both the response characteristics were mainly controlled by subtropical or tropical storms near the ocean in the Cf climate type. They are also similar to the slow events in China (Zhai et al., 2021) because the rates of positive changes are 0.01-0.94 h-1 in our study,

and  $0.04-1.78 h^{-1}$  in China (Zhai et al., 2021), and the rates of negative changes are  $0.01-0.33 h^{-1}$  in our study and  $0.02-0.25 h^{-1}$  in China (Zhai et al., 2021)." (see Lines 584–588 in the manuscript with track changes)

"The similar flood events are also reported, e.g., the low flashiness floods with the mean flood peak magnitude of  $0.20-0.25 \text{ m}^3/\text{s/km}^2$  and the mean coefficients of variation of approximate 0.90 in the northern part of central–eastern Europe (Kuentz et al., 2017), which is also controlled by the similar climate type (i.e., Df)." (see Lines 592–595 in the manuscript with track changes)

"The flood response characteristics are similar to the intermediate flood events in China (Zhai et al., 2021). For example, the coefficients of variation are 0.65-3.15 in our study and 0.78-3.07 in China (Zhai et al., 2021). The rates of positive and negative changes are 0.02-8.00 h-1 and 0.01-0.64 h-1 in our study, respectively, while those reported in Zhai et al. (2021) were 0.36-4.90 h-1 and 0.09-0.46 h-1 in China, respectively." (see Lines 598–603 in the manuscript with track changes)

**References**

- Brunner, M. I., Viviroli, D., Furrer, R., Seibert, J., and Favre, A. C.: Identification of flood reactivity regions via the functional clustering of hydrographs, Water Resour. Res., 54, 1852-1867, https://doi.org/10.1002/2017WR021650, 2018.
- China Institute of Water Resources and Hydropower Research, and Research Center on Flood and Drought Disaster Prevention and Reduction, the Ministry of Water Resources. Atlas of Flash Flood Disasters in China. Sinomap Press, 2021. (in Chinese)
- Kuentz, A., Arheimer, B., Hundecha, Y., and Wagener, T.: Understanding hydrologic variability across Europe through catchment classification, Hydrol. Earth Syst. Sc., 21, 2863–2879. https://doi.org/10.5194/hess-21-2863-2017, 2017.
- Merz, R., and Blöschl, G.: A process typology of regional floods, Water Resour. Res., 39(12), 1340, https://doi.org/10.1029/2002WR001952, 2003.
- Olden, J. D., Kennard, M. J., and Pusey, B. J.: A framework for hydrologic classification with a review of methodologies and applications in ecohydrology, Ecohydrology, 5, 503–518, https://doi.org/10.1002/eco.251, 2012.
- Saharia, M., Kirstetter, P. E., Vergara, H., Gourley, J. J., and Hong, Y.: Characterization of floods in the United States, J. Hydrol., 548, 524-535, https://doi.org/10.1016/j.jhydrol.2017.03.010, 2017.

- Sikorska, A. E., Viviroli, D. and Seibert, J.: Flood-type classification in mountainous catchments using crisp and fuzzy decision trees, Water Resour. Res., 51, 7959–7976, https://doi.org/10.1002/2015WR017326, 2015.
- Tarasova, L., Merz, R., Kiss, A., Basso, S., Günter, B., Merz, B., Viglione, A., Plötner, S. Guse, B., Schumann, A., Fischer, S., Ahrens, B., Anwar, F., Bárdossy, A., Bühler, P., Haberlandt, U., Kreibich, H., Krug, A., Lun, D., Müller-Thomy, H., Pidoto, R., Primo, C., Seidel. J., Vorogushyn, S., Wietzke, L.: Causative classification of river flood events, WIRES Water, 6(4), e1353, https://doi.org/10.1002/wat2.1353, 2019.
- Zhai, X. Y., Guo, L., and Zhang, Y. Y.: Flash flood type identification and simulation based on flash flood behavior indices in China, Sci. China Earth Sci., 64(7), 1140–1154, https://doi.org/10.1007/s11430-020-9727-1,2021.
- Zhang, Y. Y., Zhang, Y. Q., Zhai, X., Xia, J., Tang, Q., Zhao, T., and Wang, W.: Predicting flood event class using a novel class membership function and hydrological modeling, Earth's Future, 12, e2023EF004081. https://doi.org/10.1029/2023EF004081,2024.

3. Throughout the paper, the authors mainly describe numerical findings that could be presented in a table. I believe that the value of research lies in the analysis, discussion, and implications of the findings. Additionally, many of the figures contain irrelevant information that doesn't help highlight the findings.

**Response:** Your suggestion has been adopted. We summarized our results in a higherlevel way and moved some detailed information into the supplementary tables (Tables S4 and S5). The examples were as follows:

"Table S4. Average, standard deviation, median, maximum and minimum of flood response metrics in different classes

| Characteristic
value | Class | $R(mm \cdot day^{-1})$ | $Q_{pk}(mm \cdot day^{-1})$ | CV        | T bgn | $T_{pk}(%)$ | $T_{drn}(h)$                | $RQ_r(h^{-1})$ | $RQ_d(h^{-1})$  | $N_{pk}$  |
|-------------------------|-------|------------------------|-----------------------------|-----------|------------------|-------------|-----------------------------|----------------|-----------------|-----------|
|                         | 1     | 43.97±29.94            | 2.04±2.51                   | 0.90±0.26 | 2.28±0.49        | 27.14±9.60  | 103.92±43.39                | 0.13±0.32      | 0.04±0.07       | 1.31±0.51 |
| $Average \pm$           | 2     | 45.81±34.01            | 2.21±2.52                   | 0.87±0.25 | 3.06±0.69        | 50.64±10.28 | 83.82±41.20                 | 0.08±0.14      | 0.08±0.12       | 1.32±0.50 |
| Standard                | 3     | 143.97±108.33          | 5.23±6.04                   | 0.84±0.22 | 3.24±0.61        | 33.90±15.02 | 145.26±68.99                | 0.25±0.62      | 0.12±0.28       | 2.67±0.76 |
| Deviation               | 4     | 33.31±26.64            | 1.69±2.11                   | 0.86±0.26 | 3.85±0.51        | 26.11±9.09  | 85.73 ± 39.97 | 0.14±0.30      | $0.04{\pm}0.08$ | 1.24±0.43 |
|                         | 5     | 65.79±43.80            | 2.98±3.68                   | 1.40±0.43 | 3.43±0.61        | 23.74±13.60 | 202.88±85.42                | 0.18±0.62      | 0.03±0.04       | 1.24±0.46 |
|                         | 1     | 35.63                  | 1.17                        | 0.89      | 2.30             | 27.27       | 97.01                       | 0.05           | 0.02            | 1.00      |
| Madian                  | 2     | 37.84                  | 1.36                        | 0.84      | 3.03             | 49.04       | 76.99                       | 0.04           | 0.04            | 1.00      |
| mealan                  | 3     | 115.53                 | 3.09                        | 0.82      | 3.21             | 32.09       | 139.01                      | 0.07           | 0.03            | 3.00      |
|                         | 4     | 25.09                  | 1.00                        | 0.83      | 3.79             | 26.39       | 79.01                       | 0.05           | 0.02            | 1.00      |

|         | 5 | 57.11  | 1.92  | 1.32 | 3.42 | 21.26        | 190.99        | 0.04 | 0.01 | 1.00 |
|---------|---|--------|-------|------|------|--------------|---------------|------|------|------|
|         | 1 | 171.48 | 22.92 | 1.97 | 3.24 | 57.14        | 357.00        | 4.58 | 0.74 | 3.00 |
|         | 2 | 194.87 | 19.84 | 1.81 | 4.65 | 86.96        | 256.99        | 1.24 | 1.06 | 3.00 |
| Maximum | 3 | 610.70 | 34.79 | 1.45 | 4.72 | 79.91 | 493.99 | 6.89 | 2.45 | 4.00 |
|         | 4 | 174.43 | 21.02 | 2.12 | 5.25 | 55.67        | 241.01        | 3.50 | 0.91 | 3.00 |
|         | 5 | 201.00 | 27.18 | 3.15 | 5.24 | 81.56        | 465.00        | 6.76 | 0.31 | 3.00 |
|         | 1 | 3.22   | 0.13  | 0.33 | 1.05 | 4.17         | 25.01         | 0.00 | 0.00 | 1.00 |
|         | 2 | 1.11   | 0.07  | 0.32 | 1.09 | 32.65        | 13.99         | 0.00 | 0.00 | 1.00 |
| Minimum | 3 | 7.79   | 0.14  | 0.32 | 1.07 | 4.47         | 19.99         | 0.00 | 0.00 | 1.00 |
|         | 4 | 1.17   | 0.04  | 0.29 | 2.88 | 5.56         | 16.99         | 0.00 | 0.00 | 1.00 |
|         | 5 | 1.54   | 0.07  | 0.65 | 1.57 | 1.61         | 25.01         | 0.00 | 0.00 | 1.00 |

| р.:       | g:                   | A11 * .*      |     | Flood  | l event nu | mber of c | lass |          |             | Pe                       | ercentage(  | ( %) |              |
|-----------|----------------------|---------------|-----|--------|------------|-----------|------|----------|-------------|--------------------------|-------------|-----------------|--------------|
| Basins    | Stations             | Abbreviations | 1   | 2      | 3          | 4         | 5    | Total    | 1           | 2                        | 3           | 4               | 5            |
|           | Dongfeng             | DF            | 0   | 3      | 1          | 9         | 1    | 14       | 0.0         | 21.4                     | 7.1         | 64.3            | 7.1          |
|           | Jingyu               | JY            | 0   | 3      | 1          | 9         | 0    | 13       | 0.0         | 23.1                     | 7.7         | 69.2            | 0.0          |
| Songliao  | Muling               | ML            | 0   | 0      | 2          | 7         | 3    | 12       | 0.0         | 0.0                      | 16.7        | 58. 3    | 25.0         |
| 0         | Yitong               | YT            | 0   | 6      | 0          | 7         | 1    | 14       | 0.0         | 42.9                     | 0.0         | 50.0            | 7.1          |
|           | Tot                  | tal           | 0   | 12     | 4          | 32        | 5    | 53       | 0.0         | 22.6                     | 7.5         | 60.4            | 9.4          |
|           | Huating              | HT            | 0   | 2      | 0          | 7         | 2    | 11       | 0.0         | 18.2                     | 0.0         | 63.6            | 18.2         |
|           | Luanchuan            | LC            | 4   | 6      | 2          | 27        | 0    | 39       | 10.3        | 15.4                     | 5.1         | 69.2            | 0.0          |
| Yellow    | Oiaotou              | OT            | 0   | 4      | 1          | 17        | 0    | 22       | 0.0         | 18.2                     | 4.5         | 77.3            | 0.0          |
|           | Tantou               | $\tilde{T}T$  | 7   | 2      | 2          | 16        | 5    | 32       | 21.9        | 6.3                      | 6.3         | 50.0            | 15.6         |
|           | Tor                  | tal           | 11  | 14     | 5          | 67        | 7    | 104      | 10.6        | 13.5                     | 4.8         | 64.4            | 6.7          |
|           | Beimiaoii            | BM            | 0   | 0      | 0          | 0         | 12   | 12       | 0.0         | 0.0                      | 0.0         | 0.0             | 100.0        |
|           | Danoling             | DP            | Ő   | 6      | 1          | 5         | 9    | 21       | 0.0         | 28.6                     | 4.8         | 23.8            | 42.9         |
|           | Huangnizhuang        | HN            | Ĩ   | Ő      | 1          | 4         | 4    | 10       | 10.0        | 0.0                      | 10.0        | 40.0            | 40.0         |
|           | Lixin                | LX            | 0   | 5      | 5          | 4         | 4    | 18       | 0.0         | 27.8                     | 27.8        | 22.2            | 22.2         |
|           | Luzhuano             | IZ            | 1   | 0      | 0          | 4         | 6    | 11       | 91          | 0.0                      | 0.0         | 36.4            | 54.5         |
|           | Peihe                | PH            | 5   | ő      | 1          | 5         | 7    | 18       | 27.8        | 0.0                      | 5.6         | 27.8            | 38.9         |
|           | Oilin                |               | 2   | ő      | 0          | 1         | 7    | 10       | 20.0        | 0.0                      | 0.0         | 10.0            | 70.0         |
| Huaihe    | Xiaoushan            | XG            | 3   | 3      | 1          | 3         | ģ    | 19       | 15.8        | 15.8                     | 53          | 15.8            | 474          |
|           | Xinxian              | XX            | 3   | 3      | 2          | 2         | 14   | 24       | 12.5        | 12.5                     | 83          | 83              | 58.3         |
|           | Yanozhuano           | YZ            | 0   | 5      | 1          | 2         | 2    | 10       | 0.0         | 50.0                     | 10.0        | 20.0            | 20.0         |
|           | Thongtang            | 7 7    | 2   | 3      | 1          | 4         | 5    | 15       | 133         | 20.0                     | 67          | 26.7            | 333          |
|           | Zhunghang            | 7G            | 4   | 2      | 1          | 2         | 17   | 26       | 15.5        | 20.0                     | 3.8         | 20.7            | 65.4         |
|           | Ziluoshan            | 20
7I      | 3   | 2      | 2          | 8         | 6    | 20       | 14.3        | 9.5                      | 9.5         | 38.1            | 28.6         |
|           | Zituoshun
Toi     | tal           | 24  | 20     | 16         | 44        | 102  | 215      | 11.2        | 13.5                     | 74          | 20.5            | 20.0
47.4 |
|           | Anho                 | ΔΗ            | 5   | 3      | 2          | 3         | 102  | 14       | 35.7        | 21.4                     | 1/1 3       | 20.5            | 71           |
|           | Anron                | AR            | 8   | 14     | 2          | 3         | 5    | 33       | 24.2        | 21. <del>4</del>
12.1 | 0.1         | 01              | 15.2         |
|           | Raituaana            | AK
BT      | 1   | 14     | 1          | 6         | 5    | 55
11 | 24.2        | 27.3                     | 9.1         | 9.1
54.5     | 15.2         |
|           | Biyana               | DI
RV      | 1   | 5      | 0          | 10        | 0    | 12       | 9.1
8 3  | 27.5                     | 9.1         | 82.2            | 0.0          |
|           | Changeum             |               | 1   | 2      | 0          | 10        | 0    | 23       | 0.J
17.8 | 13.0                     | 30.1        | 0.0             | 0.0          |
|           | Dutou                |               | 6   | 5
8 | 9          | 8         | 0    | 23       | 47.0        | 13.0
34.8             | 39.1
1 3 | 21.8            | 0.0          |
|           | Caotan               |               | 1   | 5      | 1          | 6         | 1    | 23       | 20.1        | 217                      | 4.5         | 26.1            | 174          |
|           | Gaolan               |               | 4   | 5      | 4          | 0         | 4    | 23       | 17.4        | 21.7                     | 17.4        | 20.1            | 17.4         |
|           | Jiane
L'alla Cana |               | 0   | 0      |            | 0         | 0    | 15       | 40.2        | 40.2                     | 1.1         | 0.0             | 0.0          |
|           | Jiajiajang           | JJ
ID      | 2   | 4      | 0          | 4         | 1    | 11       | 18.2        | 30.4                     | 0.0         | 30.4            | 9.1          |
|           | Jinping              | JP
IT      | 5   | 2      | 0          | 2         | 4    | 1/       | 17.0        | 11.8                     | 33.3        | 11.8            | 23.5         |
|           | Jitan                | JI            | 0   | 2      | 2          | 3         | 4    |          | 0.0         | 18.2                     | 18.2        | 27.3            | 30.4         |
|           | Juwan                | JW            | 4   | 3      | 0          | 8         | 1    | 10       | 25.0        | 18.8                     | 0.0         | 50.0            | 0.3          |
|           | Liangshuikou         |               | 24  | 0      | 0          | 20        | 3    | 05       | 30.9        | 9.2                      | 9.2         | 40.0            | 4.0          |
|           | Liqingdian           | LQ            | 0   | 0      | 2          | 14        | /    | 29       | 0.0         | 20.7                     | 0.9         | 48.3            | 24.1         |
|           | Loudi                | LD            | 7   | 5      | 6          | 2         | 5    | 25       | 28.0        | 20.0                     | 24.0        | 8.0             | 20.0         |
|           | Miping               | MP            | 3   | 3      | 5          | 3         | 5    | 19       | 15.8        | 15.8                     | 26.3        | 15.8            | 26.3         |
|           | Pingshi              | PS            | 5   | 3      | 1          | 8         | 5    | 22       | 22.7        | 13.6                     | 4.5         | 36.4            | 22.7         |
|           | Shahebu              | SH            | 3   | 3      | 2          | 2         | 0    | 10       | 30.0        | 30.0                     | 20.0        | 20.0            | 0.0          |
|           | Shanggao             | SG            | 10  | 2      | 2          | 3         | 2    | 19       | 52.6        | 10.5                     | 10.5        | 15.8            | 10.5         |
| Yangtze   | Shijie               | SJ            | 3   | 4      | 0          | 4         | 2    | 13       | 23.1        | 30.8                     | 0.0         | 30.8            | 15.4         |
|           | Shimenkan            | SM            | 16  | 25     | 2          | 5         | 2    | 50       | 32.0        | 50.0                     | 4.0         | 10.0            | 4.0          |
|           | Shuangfeng           | SF            | 9   | 8      | 7          | 8         | 1    | 33       | 27.3        | 24.2                     | 21.2        | 24.2            | 3.0          |
|           | Shuangjiangkou       | SK            | 8   | 3      | 12         | 1         | 0    | 24       | 33.3        | 12.5                     | 50.0        | 4.2             | 0.0          |
|           | Sifen                | SI            | 4   | 2      | 2          | 0         | 2    | 10       | 40.0        | 20.0                     | 20.0        | 0.0             | 20.0         |
|           | Tangdukou            | TD            | 10  | 19     | 1          | 2         | 1    | 33       | 30.3        | 57.6                     | 3.0         | 6.1             | 3.0          |
|           | Tanghe               | TH            | 0   | 3      | 1          | 5         | 9    | 18       | 0.0         | 16.7                     | 5.6         | 27.8            | 50.0         |
|           | Tonggu               | TG            | 5   | 2      | 0          | 0         | 10   | 17       | 29.4        | 11.8                     | 0.0         | 0.0             | 58.8         |
|           | Tongtang             | ТО            | 14  | 6      | 5          | 2         | 1    | 28       | 50.0        | 21.4                     | 17.9        | 7.1             | 3.6          |
|           | Wuxigou              | WX            | 4   | 5      | 0          | 7         | 1    | 17       | 23.5        | 29.4                     | 0.0         | 41.2            | 5.9          |
|           | Xiawan               | XW            | 6   | 0      | 0          | 2         | 3    | 11       | 54.5        | 0.0                      | 0.0         | 18.2            | 27.3         |
|           | Xixia                | XI            | 1   | 1      | 3          | 5         | 6    | 16       | 6.3         | 6.3                      | 18.8        | 31.3            | 37.5         |
|           | Хири                 | XP            | 12  | 14     | 4          | 5         | 1    | 36       | 33.3        | 38.9                     | 11.1        | 13.9            | 2.8          |
|           | Yanling              | YL            | 18  | 4      | 4          | 7         | 0    | 33       | 54.5        | 12.1                     | 12.1        | 21.2            | 0.0          |
|           | Yanta                | YA            | 6   | 2      | 1          | 4         | 0    | 13       | 46.2        | 15.4                     | 7.7         | 30.8            | 0.0          |
|           | Yuanken              | YK            | 2   | 3      | 1          | 0         | 7    | 13       | 15.4        | 23.1                     | 7.7         | 0.0             | 53.8         |
|           | Yucun                | YC            | 12  | 0      | 18         | 3         | 1    | 34       | 35.3        | 0.0                      | 52.9        | 8.8             | 2.9          |
|           | Yuexi                | YX            | 14  | 4      | 11         | 5         | 3    | 37       | 37.8        | 10.8                     | 29.7        | 13.5            | 8.1          |
|           | Zhangdou             | ZD            | 4   | 3      | 0          | 5         | 0    | 12       | 33.3        | 25.0                     | 0.0         | 41.7            | 0.0          |
|           | Tot                  | tal           | 251 | 190    | 125        | 181       | 97   | 844      | 29.7        | 22.5                     | 14.8        | 21.4            | 11.5         |
| Southeast | Anxi                 | AX            | 1   | 3      | 4          | 6         | 0    | 14       | 7.1         | 21.4                     | 28.6        | 42.9            | 0.0          |

 Table S5. Flood event number and their percentages of individual classes in all the selected catchments

|       | Longshan | LS    | 1   | 3   | 16  | 3   | 0   | 23   | 4.3  | 13.0 | 69.6 | 13.0 | 0.0  |
|-------|----------|-------|-----|-----|-----|-----|-----|------|------|------|------|------|------|
|       | Tunxi    | TX    | 5   | 3   | 1   | 1   | 3   | 13   | 38.5 | 23.1 | 7.7  | 7.7  | 23.1 |
|       | Xufan    | XF    | 1   | 3   | 5   | 1   | 0   | 10   | 10.0 | 30.0 | 50.0 | 10.0 | 0.0  |
|       | Zhaoan   | ZA    | 1   | 5   | 12  | 8   | 4   | 30   | 3.3  | 16.7 | 40.0 | 26.7 | 13.3 |
|       |          | Total | 9   | 17  | 38  | 19  | 7   | 90   | 10.0 | 18.9 | 42.2 | 21.1 | 7.8  |
|       | Hezikou  | HZ    | 42  | 17  | 7   | 22  | 1   | 89   | 47.2 | 19.1 | 7.9  | 24.7 | 1.1  |
|       | Huishui  | HS    | 3   | 3   | 0   | 4   | 0   | 10   | 30.0 | 30.0 | 0.0  | 40.0 | 0.0  |
| Pearl | Libo     | LB    | 5   | 0   | 0   | 6   | 0   | 11   | 45.5 | 0.0  | 0.0  | 54.5 | 0.0  |
|       | Xiaogulu | XL    | 2   | 24  | 0   | 0   | 4   | 30   | 6.7  | 80.0 | 0.0  | 0.0  | 13.3 |
|       | _        | Total | 52  | 44  | 7   | 32  | 5   | 140  | 37.1 | 31.4 | 5.0  | 22.9 | 3.6  |
|       | Tota     | l     | 347 | 306 | 195 | 375 | 223 | 1446 | 24.0 | 21.2 | 13.5 | 25.9 | 15.4 |
|       | >>       |       |     |     |     |     |     |      |      |      |      |      |      |

More specifically, in the results section, the comprehensive introductions of flood response characteristics of different classes (Section 4.2, see Lines 303–328 in the manuscript with track changes), and control mechanisms of meteorological and physio-geographical factors (Section 4.4, see Lines 395–437 and 473–536 in the manuscript with track changes) were given to avoid the repeated present the results in the tables and figures. Additionally, the discussions were strengthened in the discussion section, particularly for the comparison of our flood event classification with the existing studies (see Lines 573–603 in the manuscript with track changes).

The figures were redrawn following the comments of you and the second reviewer, including Figures 1, 4, 5, 6, 7, 9 and 11.

**Minor comments:**

Line 40. You refer many times in the text to behavior characteristics what I consider response types. When we talk about behavior, you are trying to characterize the catchment dynamic which is intrinsic to each catchment. In other words, you try to characterize the low filter function that transform input to outputs. I would suggest changing the word behavior for response which is a more precise word for what you are analyzing.

**Response:** We replaced the word "behavior" with "response" in the whole manuscript.

Line 77. The expression "solid data foundation" is a biased description of your research. **Response:** It was revised as "provides the mechanism supports for predicting flood event classes" (see Line 102 in the manuscript with track changes).

Line 94. This is not the right way to refer to information extracted from a webpage. Check the referring rules from the journal.

**Response:** The websites were removed from the manuscript because the detailed data sources were given in the section of Code/Data availability.

Line 109. How dense is the meteorological gauge network? How can we be sure that they are representative of the basin analyzed?

**Response:** The meteorological stations in the buffer zone with a radius of 100 km of individual catchment centers were selected. All the selected meteorological stations were added in Figure 1. The total number of meteorological stations was 466 and no less than eight stations were within or around individual catchments.

---

## Author Response (AR2)

**Public justification (visible to the public if the article is accepted and published)**:

Dear authors,

There are still some comments about this version. Please have a look and revise accordingly.

Regards,

Handling editor

**Response:** Thank you very much for the careful review and constructive comments of handling editor and all the reviewers. We revised this manuscript substantially and provided the point-by-point responses to all the comments and suggestions of reviewers accordingly. All the revisions were highlighted using track changes and blue words in the manuscript.

**Report #2**

Mayor comments

First, there is room for improvement in the redaction of the paper. Specially, each time the authors describe figures. They should focus on commenting on the findings from these figures more than mentioning the numbers present in the figure. These numbers can easily be added as an appendix or supplement information because the important part is the interpretation and analysis of the figures.

**Response:** Thank you very much for your constructive comments. The Results section was revised following your comments to clearly present the main findings and their interpretations of figures (see Lines 274–290, 307, 312, 407–429, 468–474, 480–516 and 535 in the manuscript with track changes). The introductions of numbers showed in figures were moved to the appendix or supplement (see Lines 684–695 in the manuscript with track changes and Text S1 in the supplement).

Second, I would recommend checking the interpretation of the influence of different factors in the overall response class. In some cases, it does not sound reasonable that factors with zero influence (0.0%), when analyzed locally, would have a high influence when they are considered combined. It could be just the deterioration of the most important factor when it is combined with very bad factors.

**Response:** We were very sorry that the unclear introduction of constrained rank analysis method which resulted in the misunderstandings of the individual and

interactive contributions of control factor categories.

The constrained rank analysis was widely adopted to quantify the direct and interactive effects of multiple explanatory matrix on a response matrix. In our study, the individual contributions of meteorological, land cover and catchment categories were calculated by the partial rank analysis. This analysis was implemented by involving a certain control factor category as the independent matrix and the effects of other control factor categories were held constant. The percentage of constrained variance to the total variance of dependent variable matrix was considered as the individual contribution of involved control factor category on total variabilities of flood event classes. Furthermore, the entire rank analysis was also implemented by involving all the control factors as the independent variable matrix, and the variance percentage explained by independent variable matrix was considered as the entire contribution of all the control factors or categories.

If the sum of all the individual contributions was less than the entire contribution of all the factors, the interactive effects existed among the control factors and the difference between the summed and entire contributions was the interactive contribution (see the figure below). Therefore, the total contribution of a certain control factor category was its individual contribution plus the interactive contribution, which you mentioned. The individual and interactive contributions are not comparable. In the manuscript, the combined contribution was revised to the interactive contribution, and the entire, individual and interactive contributions of every control factor categories were presented, and the total contribution of every control factor categories were not presented.

The method introduction was revised as follows: *"Additionally, because of multiple control factor categories considered, two constrained rank analyses are implemented, namely entire and partial analyses. The entire analysis is implemented by involving all the control factors as the independent variable matrix, and the variance percentage explained by independent variable matrix to the total variance of dependent variable matrix is considered as the entire contribution of all the control factors or categories on total variabilities of flood event classes. The partial analyses of individual control factor categories are also implemented by involving a certain control factor category as the independent matrix and the effects of other control factor categories are held constant. The percentage of constrained variance is considered as the individual contribution of involved control factor category. The meteorological, land cover and catchment categories are adopted for the analysis individually, and their individual contributions are determined. If the sum of all the individual contributions is less than the entire contribution of all the factors, the interactive effects exist among*

*the control factors and the difference between the summed and entire contributions is the interactive*

*contribution (Legendre and Anderson, 1999; Zhang et al., 2016).*" (see Lines 243–259 in the

manuscript with track changes)

[Figure]

Figure. Entire contribution of all the control factors by the entire rank analysis,
individual and interactive contributions of factor categories by the partial rank
analysis

Minor comments.

Line 48. That is not assumed, decades of hydrological studies prove that. In fact, your
results show the same.

**Response:** The sentence was revised to "*The deductive approach mainly focuses on the similarity
of environmental factors which control flood events,......*" (see Line 48 in the manuscript with
track changes).

Line 62. What do you mean by the hard clustering method? You should describe this
classification of soft and hard before you describe the methods.

**Response:** The hard and soft clustering methods were referred from Olden et al. (2012).
A hard clustering method assumes that the flood events can be divided into non-
overlapping clusters with well-defined boundaries of all the clusters, while a soft
clustering method assumes that the flood events can belong to different clusters
simultaneously with a certain degree of membership, whose boundaries were vague.
All of these explanations were given in the introductions of tree and non-tree clustering
methods. Thus, the hard and soft clustering methods were removed from the manuscript

to avoid the repeat introductions. (see Lines 62 and 70 in the manuscript with track changes)

The sentence was also revised to *"The class boundaries of flood response metrics are vague, and the flood event classes are mainly based on the class membership degree deduced from sufficient of heterogeneous flood events"* (see Lines 72–74 in the manuscript with track changes).

*Reference*

*Olden, J. D., Kennard, M. J., and Pusey, B. J.: A framework for hydrologic classification with a review of methodologies and applications in ecohydrology, Ecohydrology, 5, 503–518, https://doi.org/10.1002/eco.251, 2012.*

Line 77. This is subjective. Currently, there are many libraries and tools available, so it is hard to mention that one is easier than the other.

**Response:** This sentence was removed accordingly (see Lines 78–79 in the manuscript with track changes).

Line 122. How much greater? How does that affect your results?

**Response:** This sentence was revised to *"The densities of flood events and gauges in the Southern China (i.e., Huaihe, Yangtze, Southeast and Pearl River Basins) were 1.25–11.01 times and 2.94–9.15 times greater than those in the Northern China (i.e., Songliao and Yellow River Basins) because of the higher occurrences of flood events"* (see Lines 125–128 in the manuscript with track changes).

The densities of flood events and gauges did not affect the results because all the results were specified in the individual headstream catchments, and the major river basins just showed the geographic locations of these catchments, which were revised as follows: *"There were 53 events at four stations, 104 events at four stations, 215 events at 13 stations, 844 events at 38 stations, 90 events at five stations, and 140 events at four stations in the upper tributaries of the Songliao River Basin (i.e., Songhua and Wusuli Rivers), Yellow River Basin (i.e., Huangshui, Jinghe and Yiluo Rivers), Huaihe River Basin (i.e., Northern and Southern tributaries), Yangtze River Basin (i.e., Hanjiang, Wujiang, Dongtinglake, Poyanglake, and lower Yangtze River), Southeast River Basin (i.e., Qiantang and Jinjiang Rivers) and Pearl River Basin (i.e., Beijing, Xijiang and Dongjiang Rivers), respectively."* (see Lines 118–125 in the manuscript with track changes).

Line 135- 138. This paragraph is not clear. Rewrite it.

**Response:** This sentence was revised to *"The geographic information system (GIS) data contained the digital elevation model, and the land cover data series in six periods (i.e., 1990, 1995, 2000, 2005, 2010 and 2015) whose spatial resolution is 30 m×30 m. The GIS data were downloaded*

*from the Data Center of Resources and Environmental Science, Chinese Academy of Sciences, and were adopted to extract catchment attributes and area percentages of individual land cover types.*" (see Lines 140–145 in the manuscript with track changes).

Line 138-140. That a hydrological model predicts well doesn't mean your inputs are right. The parameters in the model can correct problems or biases in the inputs. Moreover, you should mention how good was the model.

**Response:** We agreed with you about the predictions of hydrological model. The data sources of control factors and interpolation methods were introduced clearly to show their reliability.

"*The daily meteorological variables were interpolated to the catchment by the inverse distance weighting method, which is one of commonly-used meteorological interpolation methods (Ahrens, 2006; Tan et al., 2021).*" (see Lines 138–140 in the manuscript with track changes).

"*All these data sources for control factor calculations had been widely used to represent the meteorological and underlying surface conditions in China for hydrometeorological change detection and causal analysis, hydrological modelling, and so on (Zhang et al., 2020; Du et al., 2022; Zhang et al., 2024).*" (see Lines 145–149 in the manuscript with track changes).

*Reference*

*Ahrens, B.: Distance in spatial interpolation of daily rain gauge data, Hydrol. Earth Syst. Sci., 10, 197–208, https://doi.org/10.5194/hess-10-197-2006, 2006.*

*Du, Y., Wang, D., Zhu, J., Lin, Z. and Zhong, Y.: Intercomparison of multiple high-resolution precipitation products over China: Climatology and extremes, Atmos. Res., 278, 106342, https://doi.org/10.1016/j.atmosres.2022.106342, 2022.*

*Tan, J., Xie, X., Zuo, J., Xing, X., Liu, B., Xia, Q., and Zhang, Y.: Coupling random forest and inverse distance weighting to generate climate surfaces of precipitation and temperature with multiple-covariates, J. Hydrol., 598, 126270, https://doi.org/10.1016/j.jhydrol.2021.126270, 2021.*

*Zhang, Y., Ren, Y., Ren, G. and Wang, G.: Precipitation trends over mainland China from 1961–2016 after removal of measurement biases, J. Geophys. Res.: Atmos., 125(11), e2019JD031728, https://doi.org/10.1029/2019JD031728, 2020.*

Line 150-153. You are mentioning the same information that is already in the table.

**Response:** This sentence was revised to "*Therefore, nine metrics are used to fully characterize the response of flood events (Table 1).*" (see Lines 159–162 in the manuscript with track changes).

Line 166-167. I think you do not need to mention what a dimension reduction is.

**Response:** This sentence was deleted accordingly, and the next sentence was revised to "*principal component analysis is used to transform the high dimensional metrics into a few principal components (PCA) based on the orthogonal transform.*" (see Lines 176–179 in the manuscript with track changes).

Line 183-188. You can send this information to the appendix.

**Response:** It was revised accordingly.

*"Appendix A:*

*All the multivariable statistical analyses are implemented using R software (version 3.1.1) (R Development Core Team, 2010), involving the `aov`, `cor` and `princomp` functions in stats Package (version 4.1.3) for independence test, linear correlation test and principal component analysis, respectively (Mardia et al., 1979), the `hcluster` function in amap Package (version 0.8-18) for hierarchical cluster analysis (Antoine and Sylvain, 2006), the `clara` function in cluster Package (version 2.1.3) for k-medoids cluster analysis (Kaufman and Rousseeuw, 1990), the `NbClust` function in NbClust Package (version 3.0.1) for the optimal class number determination and classification performance assessment (Charrad et al., 2014). The Monte Carlo permutation test are implemented using the `envfit`, `decorana`, `rda`, `cca`, `permutest` functions in the vegan Package (version 2.5-7) of R software (version 3.1.1) (ter Braak, 1986; R Development Core Team, 2010).*" (see Lines 675–683 in the manuscript with track changes).

Line 199. SPEI is a drought index. You did not use an aridity index. Please change the term.

**Response:** The aridity index was replaced by the drought index in the whole manuscript (see Lines 22, 209, 225, 213, 412, 458, 495, 500, 523, 526, 529, 539, 603, 605, 606, 626 and 667 in the manuscript with track changes).

Line 237. How can you conclude about catchment factors if they are not dynamic?

**Response:** The catchment factors are excluded for the effect analysis of control factors on variability of flood event classes in the individual catchments (i.e., distributed analysis) because they are not dynamic. However, they are included for the effect analysis in the entire regions because they are different among individual catchments (i.e., lumped analysis).

We were sorry about the misunderstanding, and revised these sentences to "*All the meteorological and physio-geographical factors are included for the lumped analysis, while the catchment attributes are excluded for the distributed analysis because they are not dynamic in the individual catchments.*" (see Lines 264–266 in the manuscript with track changes).

Line 247. You have to reference the table you are describing. Moreover, Try to add a hydrological meaning to the component, such as you did with PC 4 and 5.

**Response:** The paragraph and table 3 were revised to remove the repeated information of table and add the hydrological meaning of individual components catchment factors.

*"By the principal component analysis, five independent PCAs are found with the total cumulative variance of 85.7%, all of which are selected in our study (Table 3). The first PCA is related with magnitude, variability and rates of changes with the explained variances of 33.3%. The second PCA is related with magnitude, variability and peak number with the explained variances of 17.0%. The third–fifth PCAs are mainly related with flood event duration, beginning time of flood event and flood peak timing with the explained variances of 16.0%, 10.8% and 8.6%, respectively.*

*Table 3. Loads coefficients of flood response metrics in the selected PCAs and their explained variances*

| Components | Variances (%) | Main hydrological metrics and their coefficients | Hydrological meanings |
|---|---|---|---|
| PCA1 | 33.3 | $Q_{pk}$ (0.97), R (0.61), $RQ_r$ (0.84) and $RQ_d$ (0.84) | Flood magnitude and rates of changes |
| PCA2 | 17.0 | R (0.51), CV (-0.47), $T_{pk}$ (0.56) and $N_{pk}$ (0.77) | Flood magnitude, variability and peak number |
| PCA3 | 16.0 | $T_{drn}$ (0.84) | Flood event duration |
| PCA4 | 10.8 | $T_{bgn}$ (0.92) | Beginning time of flood event |
| PCA5 | 8.6 | $T_{pk}$ (0.64) | Flood peak timing |

*"* (see Lines 274–290 in the manuscript with track changes).

Line 258-266. This information should go in the appendix or supplement information because it only supports the selection of the number of clusters.

**Response:** The selection of cluster number is an important content of this study. We preferred to remove this paragraph into the appendix.

*"Furthermore, the optimal classification of all the 1446 flood events are determined by comparing the classification performance between the hierarchical and k-medoids clustering methods. The five clusters using the k-medoids clustering method are optimum for further analysis in our study (Figure B1 in the Appendix B)."* (see Lines 285–287 in the manuscript with track changes).

*"Appendix B:*

*The optimal classification method and cluster number are determined by comparing the classification performance between the hierarchical and k-medoids clustering methods among individual cluster numbers. Figure B1 shows that the optimal criteria number is the largest when the cluster number is five (i.e., 22.7% of total) for the k-medoids clustering method. The optimal criteria are the CCC, TrCovW, Silhouette, Ratkowsky and PtBiserial with the values of -2.98, $1.39 \times 10^{15}$, $4.12 \times 10^6$, 0.20, 0.29 and 0.39, respectively. Therefore, the five clusters using the k-medoids clustering method are optimum for further analysis in our study. The flood event numbers in the individual classes are 347, 306, 195,*

*375 and 223, accounting for 24.0%, 21.2%, 13.5%, 25.9% and 15.4% of total events, respectively."* (see Lines 684–695 in the manuscript with track changes).

Line 272. What do you mean by "variations are the same among the different classes"? Class 3 is statistically different than others.

**Response:** It means that "*the distributions of both total flood volume (R) and maximum flood peak (Qpk) are the same among different classes. That is to say, the metric values are the largest in Class 3, followed by Classes 5, 2, 1 and 4.*" (see Lines 304–306 in the manuscript with track changes).

Line 274. You do not need this level of description in a paragraph.

**Response:** The value ranges of individual classes were removed from the manuscript because all of these values had been presented in the table (see Lines 307 and 312 in the manuscript with track changes).

Line 291-297. I am not sure if the description you mention in this paragraph comes only from Fig. 3 or if you really need Fig 4 to conclude that. If Fig 4 is only supporting this description, my suggestion is to move it to the appendix.

**Response:** Thank you very much for your comments. The description of Classes 1–5 in this paragraph was based on Figures 3 and 4. Figure 4 showed the hydrographs of individual flood event classes and their duration frequencies, which were beneficial to support the descriptions of individual flood event classes. Thus, we preferred to keep this figure in this section, and the explanation of Figure 4 was also given to explain the description.

"*According to the metric distributions (Figure 2), and hydrographs and duration frequencies (Figure 3) of individual flood event classes, we can conclude that……*" (see Lines 325–326 in the manuscript with track changes).

Line 310. You should be more specific when you mention basins because your data does not cover the entire basin (e.g. upper, lower, headwater basin, etc).

**Response:** The locations of Classes 1–5 were specified to the tributaries of the main river basins. The revisions were given as follows:

"*The spatial distributions of individual classes are showed in Figures 4 and S1, and Table S5 in the Supplement. The moderately fast flood event class (i.e., Class 1) is mainly in the upper Dongjiang River of the Pearl River Basin, Poyanglake and Dongtinglake tributaries of Yangtze River Basin, accounting*

*for 37.1% (52/140) and 29.7% (251/844) of total events in the main river basins, respectively. Specifically, Class 1 is dominant at the Yanling (54.5%, 18/33) and Tongtang (50.0%, 14/28) stations in the Dongtinglake tributaries, the Shanggao (52.6%, 10/19) station in the Poyanglake tributaries, and the Hezikou (47.2%, 42/89) station in the Dongjiang River. The highly fast flood event class (i.e., Class 2) is mainly in the upper Beijing River of the Pearl River Basin, and Dongtinglake tributaries of Yangtze River Basin, accounting for 31.4% (44/140) and 22.5% (190/844) of total events in the main river basins, respectively. Class 2 is particularly dominant at the Xiaogulu (80.0%, 24/30) station in the Beijiang River, and the Tangdukou (57.6%, 19/33) station in the Dongtinglake tributaries. The highly slow and multipeak flood event class (i.e., Class 3) is mainly in the upper Jinjiang, Qiantang and Minjiang Rivers in the Southeast River Basin, accounting for 42.2% (38/90) of total events, particularly at the Longshan (69.6%, 16/23) station in the Jinjiang River. The slightly fast flood event class (i.e., Class 4) is mainly in the upper Huangshui, Jinghe and Yiluo Rivers of the Yellow River Basin, and upper Songhua and Wusuli Rivers of the Songliao River Basins, accounting for 64.4% (67/104) and 60.4% (32/53) of total events in the main river basins, respectively. This class is dominant at the Qiaotou (77.3%, 17/22) station in the Huangshui River, the Huating (63.6%, 7/11) station in the Jinghe River and the Luanchuan (69.2%, 27/39) station in the Yiluo River, the Jingyu (69.2%, 9/13) and Dongfeng (64.3%, 9/14) stations in the Songhua River, and the Muling (58.3%, 7/12) station in the Wusuli River. The moderately slow flood event class (i.e., Class 5) is mainly in the southern tributaries of Huaihe River Basin, accounting for 47.4% (102/215) of total events, particularly at the Beimiaoji (100%, 12/12) and Qilin (70.0%, 7/10) stations.*" (see Lines 344–364 in the manuscript with track changes)

Line 317. What do you mean by obvious?

**Response:** This sentence was revised to "*This class is dominant at……*". (see Line 358 in the manuscript with track changes)

Figure 5. If you are going to talk about average behavior by basin, Figure 5 should show a pie chart on top of each basin. From the gauge description in the current figure is almost impossible to follow your conclusions. Another option is to have a pie chart on the left panel and the gauge locations on the right.

**Response:** Thank you very much for your suggestion. This figure was revised carefully as follow (see Line 370 in the manuscript with track changes).

[Figure]

*Figure 4. Spatial variabilities of individual flood event classes at headstream stations of major river basins*

Line 326. When you talk about a subpanel of your figure, you should mention the subpanel in the text.

**Response:** The subpanels were mentioned accordingly (see Lines 371, 374, 377, 380, 388 and 396 in the manuscript with track changes).

Line 328. Be more specific about what part of the basin you are describing (same as Line 310).

**Response:** The interannual distributions of individual classes were presented specifically at station scales, which were given as follows.

"*However, the interannual distributions of individual classes are quite distinct at different stations, particularly in the upper Songhua and Wusuli Rivers of Songliao River Basin. At the headstream stations of Songliao River Basin (Figure 5b), the Class 4 is dominant with the annual mean percentage of 26.1±38.3% (n=32) though flood events are missed in several years due to the dry period. The dominance of Class 4 is the most considerable in 1996, 1998, 2002 and 2009 at the Muling station in the upper Wusuli River. At the headstream stations of Yellow River Basin (Figure 5c), the Class 4 is also dominant across the whole period with the annual mean percentage of 58.1±33.9% (n=67), particularly in 1994¬– 1996, 1999 and 2007. The dominance of Class 4 is the most considerable in 1993–1995 and 2001–2004 at the Huating station in the upper Jinghe River. At the headstream stations of Huaihe River Basin (Figure 5d), the Class 5 gradually prevail with the annual mean percentage of 41.5±23.7% (n=102), particularly after 2007, whose percentage reaches 63.2±15.8% (n=79). The dominance of Class 5 is the*

*most considerable in 2007-2014 at the Beimiaoji station in the southern tributaries. The event numbers of both Classes 1 and 2 gradually decrease, accounting for 33.1±24.4% (n=11) and 8.7±7.1% (n=5) of annual flood events in the period of 1993–1999 and 2011–2015 for the Class 1, respectively, and 20.3±20.9% (n=9) and 2.7±1.3% (n=1) in the period of 1993-1999 and 2011-2015 for the Class 2, respectively. The decrease in Classes 1 and 2 are remarkable at the Peihe station in the southern tributaries and the Ziluoshan station in the northern tributaries, respectively. The explanations are that the total precipitation amount and duration probably increase due to the climate change (Dong et al., 2011; Jin et al., 2024). At the headstream stations of Yangtze River Basin (Figure 5e), the Classes 1, 2 and 4 are dominant, accounting for 29.3±9.6% (n=251), 23.0±11.5% (n=197) and 21.1±7.0% (n=181) of annual mean flood events, respectively. Although the interannual changes of event numbers of Classes 1 (n=1–21), 2 (n=1–14) and 4 (n=1–16) are considerable, those of class percentages are relatively uniform except 2015. The class dominance is the most considerable in 1993, 1995–1997 and 1998 at the Yanling station in the Dongtinglake tributaries for Class 1, in 1993, 1994 and 1997 at the Dutou station in the Poyanglake tributaries for Class 2, in 1998, 2000, 2001, 2004, 2005, 2007, and 2010–2013 at the Biyang station in the tributaries of Hanjiang River for Class 4, respectively. At the headstream stations of Southeast River Basin (Figure 5f), the Class 3 gradually prevail after 2000 with the annual mean percentage of 46.2±32.5% (n=39), which is remarkable at the Longshan station in the upper Jinjiang River. At the headstream stations of Pearl River Basin (Figure 5g), the Class 1 is dominant with the annual mean percentage of 36.0±24.0% (n=52), but gradually shifts to Class 2 which accounts for 30.0±25.2% of annual mean flood events (n=40), particularly after 2008. The class dominance is the most considerable from 1993 to 2007 at the Hezikou station in the upper Dongjiang River for Class 1, and in 1993, 1994, 1996, 2005, 2006, and 2009–2011 at the Xiaogulu station in the upper Beijiang River for Class 2, respectively.*" (see Lines 372–400 in the manuscript with track changes)

Line 356. You should mention Table 4.

**Response:** The table was added accordingly (see Line 415 in the manuscript with track changes).

Line 362-376. Move to an appendix the description of the other classes. You should comment more than describe.

**Response:** The control factors and their contributions were presented in the Supplement (Text S1 and Figures S2–5 in the Supplement), and some explanations of the major control factors were also given in this paragraph. This paragraph was shortened as follows: *"According to the Monte Carlo permutation test between flood response matrix and control factor matrix in the individual catchments of Class 1, the total and mean precipitations, and drought index during the event ($rpcp\_dur$=0.65–0.99, n=14; $rpcp\_av$=0.70–0.97, n=7; $rSPEI\_dur$=0.52–0.97, n=7) are the major control factors in 44.7% (17/38), 20% (1/5) and 25% (1/4) of total catchments of the Yangtze, Southeast and Pearl River Basins, respectively (Figure 6 and Table*

*4). The contributions of control factors are statistically significant only in the Liangshuikou catchment of the Yangtze River Basin and Hezikou catchment of the Pearl River Basin. In the Liangshuikou catchment, 96.3% of temporal differences are explained, in which the meteorological and land cover categories explain 92.5% and 3.8%, respectively. In the Hezikou catchment, 66.7% of temporal differences are explained, in which the meteorological category and the interactive impact explain 49.4% and 17.3%, respectively. The major control factors and their contributions for the Classes 2–5 are also presented in Text S1 and Figures S2–5 of the Supplement. For all the classes, only the factors in the meteorological category are statistically significant, particularly the precipitation amount and intensity, and drought index during the events. The most control factors with statistical significances are in Class 1, followed by Classes 4, 5, 3 and 2. These control factors for individual classes are detected mainly in the catchments of Yangtze (Class 1), Yellow and Pearl (Class 4), Huaihe (Class 5), Southeast (Class 3) and Pearl (Class 2) River Basins, respectively. The explanations are that the precipitation amount and potential evapotranspiration during the event usually show remarkable differences among different events, which directly determine the spatial and temporal heterogeneities of flood generation process, and consequently flood event hydrograph, but the land covers usually show slow changes in the headstream catchments due to slight disturbances of human activities and climate changes."* (see Lines 407–429 in the manuscript with track changes)

Line 392. Add in parentheses the name that appears in the figure for the grassland.

Figure 8. Add separation between land cover and catchment

**Response:** It was revised accordingly.

*"……particularly the precipitation amount and intensity (i.e., pcp_ant, pcp_dur, pcp_max, pcp_av, pcp_Tbeg, and pcp_Tdur), and the drought index during the events (SPEI_dur) with the correlation coefficients of 0.33¬–0.74, 0.20–0.38 and 0.29–0.41, respectively. The significant factor number in the catchment attribute category is less, which are mainly the mean catchment length (Length), river density (Rivden) and ratio of river width to depth (RivSlope) with the correlation coefficients of 0.18–0.32, 0.15–0.24 and 0.21–0.30, respectively. In the land cover category, only the grassland area ratio (Rgrass)……"* (see Lines 457–462 and 465 in the manuscript with track changes)

[Figure]

Line 398-403. You are just describing the number of the figure. I think you should delete that or move to apprendix.

**Response:** The sentences were shortened, and the numbers of the figures were deleted.

"*In the Class 1, the significant control factors are the precipitation, potential evapotranspiration and drought index in the antecedent seven days (i.e., pcp_ant, pet_ant and SPEI_ant) and during the events (i.e., pcp_dur, pcp_av, pcp_max, pcp_Tbeg, pet_dur, pet_max and SPEI_dur), and the potential evapotranspiration at the annual scale (i.e., pet_ann and pet_year) in the meteorological category, the area (Area), mean length (Length), maximum elevation (MaxiElev), river density (Rivden) and slope (RivSlope) and ratio of river width to depth (Rwd) in the catchment attribute category, and the grassland area ratio (Rglass) in the land cover category.*" (see Lines 468–474 in the manuscript with track changes)

Line 411-424. You are just describing the number of the figure. I think you should delete that or move to apprendix.

**Response:** The sentences were shortened, and the numbers of the figures were deleted.

"*The significant control factors of Class 2 are mainly in the meteorological factor category, including precipitation and potential evapotranspiration in the antecedent seven days (i.e., pcp_ant and pet_ant), precipitation and drought index during the flood events (i.e., pcp_dur, pcp_av, pcp_max, pcp_Tbeg, pcp_Tdur and SPEI_dur). In the Class 3, the significant control factors are mainly the precipitation and drought index during the flood events (i.e., pcp_dur, pcp_av, pcp_max and SPEI_dur) and catchment elevation (i.e., Elevation and MaxiElev). In the Classes 4 and 5, most of the meteorological and catchment factors are significant. The specific factors are the precipitation and potential*

*evapotranspiration in the antecedent seven days and during the events (i.e., pcp_ant, pcp_dur, pcp_av, pcp_max, pcp_Tbeg, pcp_Tdur, pet_ant, pet_dur and pet_max), drought index during the events (i.e., SPEI_dur) and precipitation at the annual scale (i.e., pcp_year) for the meteorological factor category, and the catchment area (Area), mean length (Length), river density (Rivden) and ratio of river width to depth (Rwd) in the catchment attribute category for the Class 4, and precipitation factors (i.e., pcp_ant, pcp_dur, pcp_av, pcp_max, pcp_Tbeg and pcp_year), drought index during the events and at the annual scale (i.e., SPEI_dur  and SPEI_year) for the meteorological factor category, and the catchment mean length (Length), river density (Rivden) and ratio of river width to depth (Rwd) in the catchment attribute category for the Class 5.*" (see Lines 480–504 in the manuscript with track changes)

Line 429. I am not convinced about the combined impact. The clearer situation is class 2. Catchment and land factors have zero importance by themselves. However, the combined effect is 23%. How can we be sure that the combined effect comes by the synergy of the three factors, if the combined effect is not higher than the meteorological effect by itself? Maybe the combined effect is just the effect of mixing a good factor with two awful factors, and for this reason, it is lower than the meteorological factor. Probably you should sum this combined factor only if it is higher than one of the factors.

**Response:** Thank you very much for your comments and we were very sorry that the unclear introduction of constrained rank analysis method resulted in the misunderstanding of the effect contributions of individual factor categories and their interactive contributions. We explained the methods in more details and revised the demonstration styles of all the contributions of Figure 8.

The constrained rank analysis was widely adopted to quantify the direct and interactive effects of multiple explanatory matrix on a response matrix. In our study, the individual contributions of meteorological, land cover and catchment categories were calculated by the partial rank analysis. This analysis was implemented by involving a certain control factor category as the independent matrix and the effects of other control factor categories were held constant. The percentage of constrained variance to the total variance of dependent variable matrix was considered as the individual contribution of involved control factor category on total variabilities of flood event classes. Furthermore, the entire rank analysis was also implemented by involving all the control factors as the independent variable matrix, and the variance percentage explained by independent variable matrix was considered as the entire contribution of all the control factors or categories.

If the sum of all the individual contributions was less than the entire contribution of

all the factors, the interactive effects existed among the control factors and the difference between the summed and entire contributions was the interactive contribution (see the figure below). Therefore, the total contribution of a certain control factor category was its individual contribution plus the interactive contribution, which you mentioned. The individual and interactive contributions were not comparable. In the manuscript, the combined contribution was revised to the interactive contribution. The entire, individual and interactive contributions of every control factor categories were presented, and the total contribution of every control factor categories were not presented.

[Figure]

Figure. Entire contribution of all the control factors by the entire rank analysis, individual and interactive contributions of factor categories by the partial rank analysis

The method introduction was revised as follows: *"Additionally, because of multiple control factor categories considered, two constrained rank analyses are implemented, namely entire and partial analyses. The entire analysis is implemented by involving all the control factors as the independent variable matrix, and the variance percentage explained by independent variable matrix to the total variance of dependent variable matrix is considered as the entire contribution of all the control factors or categories on total variabilities of flood event classes. The partial analyses of individual control factor categories are also implemented by involving a certain control factor category as the independent matrix and the effects of other control factor categories are held constant. The percentage of constrained variance is considered as the individual contribution of involved control factor category. The meteorological, land cover and catchment categories are adopted for the analysis individually, and their individual contributions are determined. If the sum of all the individual contributions is less than the entire contribution of all the factors, the interactive effects exist among the control factors and the difference between the summed and entire contributions is the interactive*

*contribution (Legendre and Anderson, 1999; Zhang et al., 2016)."* (see Lines 243–259 in the manuscript with track changes)

Furthermore, Figure 8 was revised to present the entire, individual and interactive contributions clearly and some data error was also revised as follows.

*"For the entire contributions of all the control factors or categories, 73.3%, 85.4%, 65.9% and 65.7% of total spatial and temporal variabilities of flood events are significantly explained in the Classes 2–5, respectively (Figure 8b–e). For the individual contributions, the meteorological factor category explains the largest variabilities (i.e., 36.5%–50.5%), followed by the catchment attribute category (i.e., 5.1%–6.1%), and the land cover category explains the least variabilities, i.e., 0.0–2.4%. The interactive impacts of all the control factor categories also explain 17.5%–33.0% of total variabilities, particularly in the Class 3."* (see Lines 504–516 and 535 in the manuscript with track changes)

[Figure]

*Figure 8. Entire, individual and interactive contributions of control factor categories on the spatial and temporal variabilities of flood event classes 1–5 (a–e)*

Line 581. I don't think that you can claim that your analysis is representative of the entire country. Be specific.

**Response:** It was revised to "……*at some headstream stations of China*" (see Line 669 in the manuscript with track changes).

---

## Author Response (AR3)

**Public justification (visible to the public if the article is accepted and published):**

Dear authors,

There are still some minor comments. Please have a look and revise accordingly.

Regards,

Handling editor

**Response:** Thank you very much for the careful review and constructive comments of handling editor and all the reviewers. We revised this manuscript substantially and provided the point-by-point responses to all the comments and suggestions of reviewers accordingly. All the revisions were highlighted using track changes and blue words in the manuscript.

**Report #2**

The author made significant revisions to this manuscript version, which enhanced and clarified nearly all of my concerns. Therefore, I recommend approving the manuscript with only minor comments.

**Response:** Thank you very much for your careful review and constructive comments. We revised this manuscript substantially and provided the point-by-point responses to all the comments and suggestions of reviewers accordingly. All the revisions were highlighted using track changes and blue words in the manuscript.

Minor comments (line numbers are from the track changes version):

Line 118. Hard transition. Incorporate some like "As summary per basin, there are 53 events…."

**Response:** It was revised accordingly (see Line 117 in the manuscript with track changes).

Line 129-130. I do not think you need to add the reference to something that is in the supplementary information. This reference should be in the supplementary information.

**Response:** This reference was deleted accordingly (see Lines 126-128 in the manuscript with track changes).

Line 137. Does it mean that catchments larger than 200 km use only the information

around the centroid?

**Response:** The selected catchment area ranged from 21 km$^2$ to 4830 km$^2$, which were much smaller than the area of buffer zone, i.e., $3.14\times100^2=3.14\times10^4$ km$^2$. Therefore, many stations out of the catchments were also selected for the meteorological interpolation (see Figure 1).

Line 213. You have the definition of aridity here. SPEI is different. However, Table 2 mentions SPEI, so at this moment, I do not know which one you are using. You have to be consistent.

**Response:** We were very sorry about the inconsistency. In our study, we used the aridity index (ADI). The SPEI and drought index in the whole manuscript were replaced by ADI and aridity index, respectively (see Lines 22, 191,195, 357, 358, 366, 387, 397-399, 408-411, 414, 417, 418, 432, 434, 437, 447, 511, 513, 514, 534, 575, and Table 2, Figures 6-8 in the manuscript with track changes, and Figures S2-S6 in the Supplement).

Line 223-223. You do not need to mention the functions and tools you used.

**Response:** We removed the functions and tools in the manuscript (see Lines 201 and 202 in the manuscript with track changes).

Figure 4. The idea of my previous comment was to simplify the figure and focus on the important message: "spatial variability". However, the authors added additional information about the subbasin. I would try to minimize the use of solid colors for the pie chart (you just need the contour of the polygon). Delete the name of each station. Use a 2D pie chart (maybe make them bigger). If you consider the information of the subbasin relevant, you could add the name to the figure.

**Response:** The tributary subbasins added in the figure would be useful to present the spatial distributions of flood events among different subbasins. The figure was revised following your suggestions (see Figure 4 in the manuscript with track changes ).

[Figure]

Line 459. Grammar: "The significant factor number in the catchment attribute category is low, despite that the more relevant are the mean catchment length (Length), river density (Rivden), and the ratio of river width to depth (RivSlope)."

**Response:** It was revised accordingly (see Lines 388-390 in the manuscript with track changes).

---

## Author Response (AR4)

1. Figures 1, 4 and S1 may contain a territory that is disputed according to the United Nations. If and when the manuscript is accepted for final revised publication, you will be asked to choose one of the following options: (a) you could remove the disputed territory from the map and submit new figure files, or (b) we could add a statement that some figures contain disputed territories.

Response: Figures 1, 4 and S1 were revised and the national boundary and undetermined national boundary were provided.

2. For the next revision please use the initials instead of the full names of the authors for the section "Author`s contribution".

Response: The initials of all the author names were given in the section of "author's contribution".